# Integrated validation of assimilating satellite derived observations over France using a hydrological model

D. Fairbairn[1], A. L. Barbu[1,2], A. Napoly[1], C. Albergel[1], J.-F. Mahfouf[1], and J.-C. Calvet[1]

[1]CNRM, UMR 3589 (Météo-France, CNRS), Toulouse, France
[2]Now at Observatoire Midi-Pyrénées, Toulouse, France

*Correspondence to:* J.-C. Calvet (jean-christophe.calvet@meteo.fr)

**Abstract.** This study validates the impact of assimilating surface soil moisture (SSM) and leaf area index (LAI) observations into a land surface model using the SAFRAN-ISBA-MODCOU (SIM) hydrological suite. SIM consists of three stages: (1) An atmospheric reanalysis (SAFRAN) over France, which forces (2) the 3-layer ISBA land surface model, which then provides drainage and runoff inputs to (3) the MODCOU hydrogeological model. The drainage and runoff outputs from ISBA are validated by comparing the simulated river discharge from MODCOU with over 500 river-gauge observations over France and with a subset of stations with low-anthropogenic influence, during several years. This study makes use of the A-gs version of ISBA that allows for leaf-scale physiological processes. The atmospheric forcing for the ISBA-A-gs model underestimates direct short-wave and long-wave radiation by approximately $5\%$ averaged over France. The ISBA-A-gs model also significantly underestimates the grassland LAI compared with satellite retrievals during winter dormancy. These differences result in an underestimation (overestimation) of evapotranspiration (drainage and runoff). The excess runoff flowing into the rivers and aquifers contributes to an overestimation of the SIM river discharge. We attempted to resolve these problems by performing the following experiments: (i) a correction of the minimum LAI model parameter for grasslands, (ii) a bias-correction of the model radiative forcing, (iii) the assimilation of LAI observations and (iv) the assimilation of SSM and LAI observations. The data assimilation for (iii) and (iv) was done with a simplified extended Kalman filter (SEKF), which uses finite differences in the observation operator Jacobians to relate the observations to the model variables. Experiments (i) and (ii) improved the median SIM Nash scores by about $9\%$ and $18\%$ respectively. Experiment (iii) reduced the LAI phase errors in ISBA-A-gs but had little impact on the discharge Nash efficiency of SIM. In contrast, experiment (iv) resulted in spurious increases in drainage and runoff, which degraded the median discharge Nash efficiency by about $7\%$. The poor performance of the SEKF is an artifact of the observation operator Jacobians. These Jacobians are dampened when the soil is saturated and when the vegetation is dormant, which leads to positive biases in drainage/runoff and insufficient corrections to the LAI minimum, respectively. The SSM observations are assimilated into a shallow surface layer (0-1 cm depth) that is highly sensitive to rainfall events. The planned assimilation of SSM into a slightly deeper layer (1-5 cm depth) with a multiple-layer diffusion model and the use of an ensemble Kalman filter algorithm that accounts for rainfall uncertainty should alleviate these problems. The results also highlight the important role that vegetation plays on the soil moisture fluxes. It is recommended that a spatially variable LAI minimum parameter be introduced into ISBA-A-gs based on the lowest LAI values derived from satellite observations.

# 1 Introduction

Soil moisture influences the flow of water to rivers and aquifers on weekly to monthly timescales, which makes it an important factor in hydrological models. In the last two decades there have been considerable advances in soil moisture data assimilation (DA) using remotely sensed near-surface soil moisture (Houser et al., 1998; Crow and Wood, 2003; Reichle and Koster, 2005; Drusch and Viterbo, 2007; Draper et al., 2012; de Rosnay et al., 2013). The estimation of global-scale soil moisture states has benefitted considerably from a huge expansion of the satellite coverage, namely the Advanced Scatterometer (ASCAT) instrument on board the METOP satellites (Wagner et al., 2007), the Soil Moisture and Ocean Salinity (SMOS) Mission (Kerr et al., 2001) and the Soil Moisture Active Passive (SMAP) Mission (Entekhabi et al., 2010), amongst others. However, these instruments can only indirectly observe the top 1-3 cm of soil moisture and are subject to retrieval errors. There are also spatial and temporal gaps in the observation network. The vegetation influences the soil moisture state through evapotranspiration and the vegetation coverage can be estimated by the leaf area index (LAI). This is a dimensionless quantity that represents the one-sided green leaf area per unit ground surface area (Gibelin et al., 2006). The LAI can be derived from satellite measurements in the visible range. However, over France it is available from polar-orbiting satellites at a relatively low temporal frequency (on average every 10 days) compared with soil moisture satellite observations (about every 3 days) due to cloud cover. The aim of DA methods is to combine these observations with a short model forecast from the previous analysis (the background state) to provide an improved estimate of the state of the system (the analysis). DA methods are necessary to account for the errors in the observations and the model, and to spread the information through space and time.

Many studies have investigated the assimilation of surface soil moisture (SSM) and streamflow observations into hydrological models in order to improve streamflow predictions and hydrological parameters (Aubert et al., 2003; Moradkhani et al., 2005; Clark et al., 2008; Thirel et al., 2010; Moradkhani et al., 2012). In large-scale streamflow assimilation, a DA method is typically chosen that can take into account lateral background-error covariances and flow-dependence. These features are beneficial because streamflow has important horizontal interactions. For example, Thirel et al. (2010) used the Best Linear Unbiased Estimate (BLUE) method to assimilate streamflow observations into the MODCOU hydrogeological model over France, which they used to update soil moisture in the ISBA land surface model (LSM).

LSMs concern water and energy fluxes between the soil and atmosphere. Unlike hydrological models, layer-based LSMs such as the ISBA model are typically pointwise (there is no horizontal interaction between the gridpoints), since this greatly reduces the computational expense. It is also common to use a DA method with 1D Kalman filtering (where observations are used to update colocated gridpoints) as opposed to 2D Kalman filtering (where observations are used to update colocated gridpoints and neighbouring gridpoints). Moreover, a study by Gruber et al. (2015) over the contiguous US found that there was no advantage of 2D Kalman filtering over 1D Kalman filtering when assimilating ASCAT SSM data into a soil moisture model.

In large-scale land surface DA, it is common to assimilate satellite derived SSM observations and screen-level temperature and humidity observations into a LSM, in order to improve soil moisture and screen-level variables. Typically the root-zone soil moisture (WG2) (1-3 m deep) is of more interest than SSM as it has a much larger water capacity and a long memory

(from weeks to months). Land surface DA is often performed using a 1D ensemble Kalman filter (EnKF) or a simplified extended Kalman filter (SEKF). The SEKF simplifies the EKF by assuming that the errors in the background state are fixed and uncorrelated between gridpoints. It uses finite differences for computing Jacobians necessary to extract information from the observations to the prognostic variables.

There has been increasing interest in ensemble DA for LSMs over the last two decades (Reichle et al., 2002, 2008; Zhou et al., 2006; Muñoz Sabater et al., 2007; Draper et al., 2012; Carrera et al., 2015), partly because these methods can estimate the "errors of the day" in the background-error covariance. The operational EnKF at Environment Canada is also motivated by coupling land surface DA with ensemble weather forecasting (Carrera et al., 2015). However, the correct representation of the "errors of the day" is challenging in land surface DA. A large proportion of the errors in LSMs come from the model and

the atmospheric forcing, rather than the initial conditions. Furthermore, the integrating nature and the nonlinear interactions in LSMs cause the short-term errors to dissipate over time, including random errors in the precipitation (Maggioni et al., 2011). Indeed, experiments assimilating in situ SSM observations with the ISBA 3-layer model have demonstrated that the EnKF and the SEKF produce a WG2 analysis with comparable accuracy and both methods improve on the model simulation (Muñoz Sabater et al., 2007; Fairbairn et al., 2015).

Due to its efficacy, simplicity and low computational cost, the SEKF is the preferred method at several meteorological operational centres for analyzing soil moisture and screen level variables. Hess (2001) developed a simplified 2D-Var (theoretically equivalent to an SEKF) scheme for the assimilation of screen-level temperature and humidity at the German Weather service (DWD). An SEKF has been developed for research purposes to assimilate ASCAT satellite derived soil moisture at Météo-France (Draper et al., 2009; Mahfouf, 2010) and the UK Met Office (Candy et al., 2012), amongst other variables. The

European Centre for Medium Range Weather Forecasts (ECMWF) assimilate screen-level temperature and humidity operationally with an SEKF (de Rosnay et al., 2013) and more recently assimilate ASCAT derived SSM observations (ECMWF, 2016).

In our study, we use an SEKF to assimilate LAI and SSM observations to update LAI and WG2 in the ISBA LSM within the SAFRAN-ISBA-MODCOU (SIM) hydrological suite. This study makes use of the A-gs version of ISBA that allows for

leaf-scale physiological processes. SIM is operational at Météo-France and its streamflow and soil moisture outputs are used as a tool by the French National flood alert services (Thirel et al., 2010). SIM consists of three stages: (1) An atmospheric reanalysis (SAFRAN) over France, which forces (2) the ISBA-A-gs land surface model, which then provides drainage and runoff inputs to (3) the MODCOU distributed hydrogeological model. The drainage and runoff outputs from ISBA-A-gs are validated by comparing the simulated streamflow from MODCOU with observations. Our study is different to the hydrological

studies mentioned earlier because the LSM and the DA are independent of the hydrogeological model. This study is relevant to the land surface DA community because several operational centres assimilate SSM observations using an SEKF to update WG2. Many studies have demonstrated that the force-restore dynamics of the ISBA 3-layer model can effectively simulate soil moisture and propagate the increments downwards from the surface to the root-zone (Muñoz Sabater et al., 2007; Draper et al., 2009; Mahfouf et al., 2009). An integrated validation using SIM has demonstrated that the ISBA 3-layer model can skillfully

simulate drainage and runoff fluxes over France (Habets et al., 2008). The dynamic vegetation model in ISBA-A-gs is also

capable of modelling seasonal changes in LAI (Jarlan et al., 2008; Brut et al., 2009; Barbu et al., 2011, 2014). But relatively few studies have assessed the SEKF performance using an integrated validation of the soil moisture fluxes. To our knowledge, this is the first article to demonstrate this type of validatation for LAI assimilation. Furthermore, the validation is robust because it is performed using more than 500 river gauges over France during several years.

This work is partly motivated by the study of Draper et al. (2011), which validated the assimilation of SSM into ISBA-A-gs using an identical twin experiment with SIM. Although the SEKF corrected a dry bias in WG2 that resulted from the precipitation forcing, they acknowledged that this may have been related to a bias in the SEKF rather than the assimilation accurately responding to the precipitation errors. Studies by Szczypta et al. (2011) and Le Moigne (2002) have found underestimations of about $5\%$ by SAFRAN in the direct short-wave and long-wave radiative fluxes respectively, averaged over France. In addition

to these problems with radiative forcing, we demonstrate in this study that the LSM significantly underestimates LAI for grasslands in winter (compared with satellite retrievals). The specification of the LAI minimum in the model is important because it prevents vegetation mortality and allows the regrowth of vegetation in the spring period (Gibelin et al., 2006). We use SIM as a tool to validate potential solutions to these deficiencies, based on four experiments:

    i. Correcting the model under-estimated LAI minimum parameter;

ii. Bias-correcting the SAFRAN radiative forcing;

    iii. Assimilating only LAI observations with an SEKF;

    iv. Assimilating SSM and LAI observations with an SEKF.

    Since Draper et al. (2011) already investigated the impact of assimilating SSM on SIM, it was not necessary to perform an experiment with the assimilation of SSM only. We validate the performance of these experiments using observations from

more than 500 river gauges over France during the period July 2007 to August 2014. We include an additional validation using a subset of 67 stations with low-anthropogenic influence because the MODCOU hydrogeological model only accounts for natural features.

    It should be kept in mind that LSMs are not chaotic but accumulations of errors from the model and the atmospheric forcing are significant. Studies have shown that DA can effectively reduce errors in the LSM state variables that are caused by these

deficiencies, such as precipitation errors (Albergel et al., 2010a) and phase errors in the LAI evolution (Barbu et al., 2014). In this study the SSM observations are rescaled such that the mean and standard deviation match the model climatology. Therefore, the assimilation of the SSM observations is designed to correct short term errors in the model state rather than systematic errors. The LAI observations are not rescaled, so the assimilation of LAI can impact the systematic model and forcing deficiencies, as well as the short term errors. We repeat experiments (iii) and (iv) after correcting the LAI minimum

and the radiative forcing in order to explore other potential impacts of the DA on SIM.

    The paper is structured as follows. The methods and materials are given in Sect. 2, which includes a description of the LSM, the assimilated observations, the DA methods, the experimental setup and the SIM validation. The results are presented in Sect. 3, including the impact of the model simulations and DA on the model state variables and the river discharge. A

discussion in Sect. 4 considers potential solutions to the problems encountered in this study. Finally, the conclusion is given in Sect. 5.

## 2 Methods and materials

### 2.1 ISBA-A-gs land surface model

In our study, the ISBA-A-gs LSM was forced by the atmospheric variables provided by the "Système d'Analyse Fournissant des Renseignements à la Neige" (SAFRAN). The SAFRAN forcing is derived from a meso-scale analysis system and is assumed to be homogeneous over 615 specified climate zones. The forcing is interpolated from these zones to the Lambert projected grid with a horizontal resolution of 8 km (Durand et al., 1993). The delayed cut-off version of SAFRAN was employed, which uses information from an additional 3000 climatological observing stations (which report one-monthly) over France (Quitana-Ségui et al., 2008; Vidal et al., 2010) after the real-time cut-off, which makes it more accurate. The atmospheric variables include precipitation, wind, incoming short-wave and long-wave radiation, relative humidity and air temperature with an hourly temporal sampling.

Version 8.0 of SURFEX was used in the experiments, which contains the "Interactions between Soil, Biosphere and Atmosphere" (ISBA) LSM (Noilhan and Mahfouf, 1996). The model uses the same horizontal grid resolution as SAFRAN of 8 km. The ISBA-A-gs version was used, which allows for the influence of leaf-scale physiological proceseses, including photosynthesis (Calvet et al., 1998). Each grid cell is split into twelve land cover types (so called "patches"). Soil and vegetation paramaters are derived from the ECOCLIMAP database (Faroux et al., 2013). The nitrogen dilution version (referred to as "NIT" hereafter) of ISBA-A-gs was applied, which dynamically simulates the LAI evolution (Gibelin et al., 2006). The NIT version allows for the effects of atmospheric conditions on the LAI, including the carbon dioxide concentrations.

The three-layer version of ISBA was adopted for this study (Boone et al., 1999). This includes the WG1 layer with depth 0-1cm. The WG2 layer includes WG1 and is 1-3 m deep, with the depth depending on the patch type. A recharge zone exists below WG2. The model water transfers are governed by the force-restore method of Deardorff (1977). The surface and root-zone layers are forced by the atmospheric variables and restored towards an equilibrium value. The drainage and runoff outputs from ISBA-A-gs drive the MODCOU hydrogeological model. The gravitational drainage is proportional to the water amount exceeding the field capacity (the effective limit where gravitational drainage ceases) (Mahfouf and Noilhan, 1996). It is driven by the hydraulic conductivity of the soil, which depends on the clay content. A small residual drainage below the field capacity was introduced by Habets et al. (2008) to account for unresolved aquifers. Runoff occurs when the soil moisture exceeds the saturation value.

### 2.2 Assimilated observations

The SSM observations were retrieved from ASCAT radar observations, which observe at 5.255 GHz (C-band) and a resolution of approximately 25 km. The radar is on board EUMETSAT's Meteorological Operational (MetOP) satellite. The assimilation

of ASCAT data was chosen because it was available throughout the analysis period. The original ASCAT values were converted into the surface degree of saturation (SDS, with values between 0 and 1) using a change detection technique, which was developed at the Vienna University of Technology (Tu-Wien) and is detailed in Wagner et al. (1999); Bartalis et al. (2007). The historically lowest and highest backscatter coefficients are assigned values for dry and saturated soils respectively. The Copernicus Global Land Service then calculates the surface water index (SWI) by applying a recursive exponential filter to these SDS values (Albergel et al., 2008) using a time-scale that may vary between 1 and 100 days. The SWI represents the soil wetness over the soil profile and also has values between 0 (dry) and 1 (saturated). The longer the time-scale of the exponential filter, the deeper the representative soil profile. The SWI-001 version 2.0 product was used in this study, which has a one day timescale and represents the SWI for a depth up to 5 cm. We then interpolated the SWI-001 data to the 8 km resolution model grid. As in Draper et al. (2011) an additional screening step was performed to remove observations with an altitude greater than 1500m, frozen regions and areas with an urban fraction greater than 15%.

We applied a linear rescaling to the SWI-001 data, which scales them such that the mean and standard deviations match the WG1 layer climatology (Calvet and Noilhan, 2000; Scipal et al., 2008). This is a linear approximation of the cumulative distribution matching technique, which uses higher order moments (Reichle et al., 2004; Drusch et al., 2005). As in Barbu et al. (2014), we applied a seasonal rescaling using a 3-month moving average over the experiment period (2007-2014). The rescaling is designed to remove biases between the model and the observations and in the process the SWI-001 data are converted into the same units as the model, expressed in volumetric soil moisture ($m^3/m^3$). The rescaled SSM observations were assimilated into the WG1 model layer. The observations were assumed to occur at the same time as the analysis at 09UTC and had a temporal frequency of about 3 days. This was a reasonable assumption since the satellite overpass is at 09:30 UTC.

The GEOV1 LAI product is part of the European Copernicus GEOLAND 2 project. The LAI observations were retrieved from the SPOT-VGT (August 2007 to June 2014) and PROBA-V (June 2014-July 2014) satellite data. The retrieval methodology is discussed by Baret et al. (2013). Following Barbu et al. (2014), the 1 km resolution observations were interpolated to the 8 km model gridpoints, provided that observations are present for at least 32 of the observation gridpoints (just over half the maximum amount). The observations were assumed to occur at 09UTC with a temporal frequency of 10 days. This assumption was reasonable given that LAI evolves slowly.

## 2.3 Data assimilation

The SEKF simplifies the extended Kalman filter (EKF, (Jazwinski, 1970)) by using a fixed estimate of the background-error variances at the start of the window and by assuming the covariances are equal to zero (Mahfouf et al., 2009). We used the same SEKF formulation as Barbu et al. (2014) for the assimilation of SSM and LAI observations over France. The prognostic variables are LAI and WG2. The background state ($x^b$) at time $t_i$ is a model propagation of the previous analysis ($x^a(t_{i-1})$) to the end of the 24 hour assimilation window:

$$x^b(t_i) = M_{i-1}(x^a(t_{i-1})),$$ (1)

where $M$ is the (nonlinear) ISBA-A-gs model. The observation was assimilated at the analysis time, at 09UTC, at the end of the 24-hour assimilation window. The analysis was calculated from the generic Kalman filter equation:

$$\boldsymbol{x}^{\mathrm{a}}(t_i) = \boldsymbol{x}^{\mathrm{b}}(t_i) + \mathbf{K}_i(\boldsymbol{y}_i^{\mathrm{o}} - \boldsymbol{y}_i), \tag{2}$$

where $\boldsymbol{y}^{\mathrm{o}}$ is the assimilated observation and $\boldsymbol{y}_i = H(\boldsymbol{x}^{\mathrm{b}}(t_i))$ is the model predicted value of the observation at the analysis time. The Kalman gain is defined as:

$$\mathbf{K}_i = \mathbf{B}_i\mathbf{H}_i^{\mathrm{T}}(\mathbf{H}_i\mathbf{B}_i\mathbf{H}_i^{\mathrm{T}} + \mathbf{R}_i)^{-1}, \tag{3}$$

where $\mathbf{H}$ is the Jacobian matrix of the linearized observation operator, $\mathbf{B}$ is the background-error covariance matrix and $\mathbf{R}$ is the observation-error covariance matrix. The observation operator Jacobians were calculated using finite differences for observation $k$ and model variable $l$:

$$\mathbf{H}_i^{kl} = \frac{H_i^k(M_{i-1}(\boldsymbol{x}(t_{i-1}) + \Delta x_{i-1}^l)) - H_i^k(M_{i-1}(\boldsymbol{x}(t_{i-1})))}{\Delta x_{i-1}^l}, \tag{4}$$

where $\Delta x^l$ is a model perturbation applied to model variable $l$. The WG2 and LAI perturbations were set to $1.0 \times 10^{-4} \times$WG2 and $1.0 \times 10^{-3} \times$LAI respectively. These were within the range of acceptable perturbation sizes based on the experiments of Draper et al. (2009) and Rüdiger et al. (2010). Equation (4) requires an additional model simulation for each prognostic variable. The linear assumptions in deriving the Jacobians are generally acceptable for these perturbation sizes. However, occasionally the linear assumptions can break down, especially during dry periods in summer (Draper et al., 2009; Fairbairn et al., 2015). For this reason we set an upper bound on the Jacobians of 1.0. It is worth mentioning that in situations where the model and atmospheric forcing errors are not properly taken into account the SEKF analysis will be suboptimal even if the Jacobian calculation is accurate enough. The Jacobian matrix derived from Eq. (4) is defined as follows:

$$\mathbf{H} = \begin{pmatrix} \frac{\partial \mathrm{WG1}}{\partial \mathrm{WG2}} & \frac{\partial \mathrm{WG1}}{\partial \mathrm{LAI}} \\ \frac{\partial \mathrm{LAI}}{\partial \mathrm{WG2}} & \frac{\partial \mathrm{LAI}}{\partial \mathrm{LAI}} \end{pmatrix}. \tag{5}$$

When assimilating just LAI, only the $\frac{\partial \mathrm{LAI}}{\partial \mathrm{WG2}}$ and $\frac{\partial \mathrm{LAI}}{\partial \mathrm{LAI}}$ terms are included. The $\frac{\partial \mathrm{WG1}}{\partial \mathrm{LAI}}$ is generally small, since the LAI does not substantially influence the surface layer (Barbu et al., 2014). The $\frac{\partial \mathrm{WG1}}{\partial \mathrm{WG2}}$ Jacobian couples WG1 with WG2 (Draper et al., 2009). The $\frac{\partial \mathrm{LAI}}{\partial \mathrm{WG2}}$ couples LAI with WG2 (Barbu et al., 2014). The $\frac{\partial \mathrm{LAI}}{\partial \mathrm{LAI}}$ Jacobian was studied by Rüdiger et al. (2010) and has a strong seasonal dependence. As we will demonstrate in Sect. 3.3, the examination of these Jacobians is essential in order to understand the performance of the SEKF.

SURFEX is implemented using the mosaic approach of Koster and Suarez (1992), where each model grid-box is split into 12 vegetation patches. The SEKF analysis is calculated independently for each patch, with the same observation used for all the patches in the grid-box. The analysis for the gridpoint is simply a linear aggregation of the analyses over the 12 patches, which are weighted according to their patch fractions (see Barbu et al. (2014) for further details).

Following Draper et al. (2011), the WG2 background-error standard deviation was set to 0.2($\mathrm{w}_{fc}$-$\mathrm{w}_{wilt}$), where $\mathrm{w}_{fc}$ is the field capacity and $\mathrm{w}_{wilt}$ is the wilting point. The scaling by ($\mathrm{w}_{fc}$-$\mathrm{w}_{wilt}$) assumes that there is linear relationship between the

soil moisture errors and the dynamic range, which depends on the clay content of the soil (Mahfouf et al., 2009). The SSM observation error standard deviation was set to $0.65(w_{fc}-w_{wilt})$, which is about $0.055$ m$^3$/m$^3$ averaged over France. This value is slightly larger than the median ASCAT-derived SDS error of $0.05$ m$^3$/m$^3$ estimated by Draper et al. (2011) because it also approximates the oversampling issue i.e. the same ASCAT observation covers several gridpoints. This value is comparable with observation errors expected for remotely sensed SSM observations (de Jeu et al., 2008; Draper et al., 2013). As in Barbu et al. (2011) the LAI background and observation error standard deviations were proportional to the LAI values themselves and a value of $0.2 \times$LAI was used for LAI values greater than 2 m$^2$/m$^2$. For LAI values below 2 m$^2$/m$^2$ the LAI errors were fixed at $0.4$ m$^2$/m$^2$. Both the background-error and observation-error covariance matrices of the SEKF are diagonal (zero covariances between layers), but implicit background-error covariances are derived from the **H** matrix at the analysis time. The SEKF is a pointwise method i.e. it cannot take into account horizontal covariances between gridpoints.

## 2.4 Experimental setup

The main experiments in this study are summarised in Table 1. The SIM river discharge was compared with the observations from 546 stations over France. Firstly the baseline experiment (NIT) was performed, which shows the impact of the biased radiative forcing and the under-estimated LAI minimum on the SIM river discharge. Thereafter, various potential solutions to these deficiencies were investigated, as set out in the introduction: (i) NIT$_m$, which was equivalent to NIT but with an elevated LAI minimum of 1.2 m$^2$/m$^2$ for grasslands (as opposed to 0.3 m$^2$/m$^2$ with NIT), (ii) NIT$_{bc}$, which used both the elevated LAI minimum of 1.2 m$^2$/m$^2$ and the bias-corrected radiative forcing (+5% for direct long-wave and short-wave over France), (iii) LDAS1, which used the SEKF to assimilate LAI only with the NIT model and (iv) LDAS2, which assimilated both LAI and SSM observations with the NIT model. We chose an augmented grassland LAI minimum value of 1.2 m$^2$/m$^2$ for NIT$_m$ because over 99% of points with a high percentage of grassland (the grassland patch fraction exceeding 70%) had an observed average annual LAI minimum above this value. Szczypta et al. (2011) and Le Moigne (2002) demonstrated that the direct short-wave and long-wave radiative forcing respectively are underestimated by approximately 5% averaged over France. We followed Decharme et al. (2013) in bias-correcting the direct radiative forcing by +5% for NIT$_{bc}$.

Three additional experiments in Table 1 explored whether SSM observation outliers, the under-estimated LAI minimum or the radiative forcing bias might impact the performance of the DA. The LDAS2$_{QC}$ was equivalent to LDAS2 but with a strict quality control of the SSM observations to remove any abnormal outliers due to instrument noise. The outliers were removed by rejecting observations outside the 90% confidence interval of the model (as in Eq. (1) and (2) of Albergel et al. (2010b)) after the observations had been rescaled. The LDAS1$_{bc}$ and LDAS2$_{bc}$ experiments were equivalent to LDAS1 and LDAS2 respectively, except they used the NIT$_{bc}$ model. The SSM observations for LDAS2$_{bc}$ were rescaled such that the standard deviation and mean matched those of NIT$_{bc}$.

The MODCOU hydrogeological model does not account for anthropogenic water management. However, there are many parts of France where anthropogenic water management stongly influences streamflow observations, including the reservoir operations, for hydropower, irrigation, drinking water, flood and low-flow alleviation and recreation purposes. We used the reference networks of Giuntoli et al. (2012, 2013) to extract a subset of 67 river gauges with low-anthropogenic influence from

the original 546 stations, valid for both low and high flows. We compared the results for these 67 stations with the 546 stations in order to determine if the results were affected by the ability of SIM (with or without DA) to simulate anthropogenically influenced streamflow.

## 2.5 Performance diagnostics

### 2.5.1 System validation

A system validation was performed by comparing the LAI and WG1 states with the LAI and SSM observations respectively for all the simulations and data assimilation experiments. Note that this was not an independent validation of the performance of the system, for which we would have needed independent observations. The rationale was to check the effectiveness of the SEKF i.e. to see if it improved the fit between the model simulations and the observations. The fit to the observations was determined by the root mean square difference (RMSD), the correlation coefficient (CC) and the bias. These checks were important because the effectiveness of the SEKF had an important impact on the drainage and runoff fluxes.

### 2.5.2 Validation using SIM

The SIM hydrological model was used as a tool to validate the drainage and runoff from ISBA-A-gs. A complete description and validation of SIM can be found in Habets et al. (2008). The first two stages of SIM are the SAFRAN atmospheric forcing and the ISBA-A-gs LSM, which were introduced in Section 2.1. The runoff and drainage from ISBA-A-gs are fed into the MODCOU hydrogeological model (Ledoux et al., 1989), which computes the daily evolution of aquifer storages and three-hour river flow forecasts. More than 900 river gauges are simulated with areas ranging from 240 km$^2$ to 112,000 km$^2$. The temporal and spatial evolution of two aquifers in the Rhone and Seine Basins are simulated using a diffusivity equation. The interaction between the rivers and aquifers is modelled and the soil water is routed to the rivers using an isochronism algorithm. The influence of human activity, such as dams and irrigation, is not accounted for by MODCOU. The simulated river discharge from SIM was compared with the observations from 546 river gauges that had data during the period of evaluation (2007-2014). These observations are available from the French hydrographical database (http://www.hydro.eaufrance.fr/, last accessed March 2016). We also analyzed the results for the subset of 67 stations with low anthropogenic influence from the original 546 stations. The fit of the average daily river discharge from MODCOU (measured in m$^3$/m$^3$s$^{-1}$) to the observations was measured using the Nash efficiency score (Nash and Sutcliffe, 1970). The Nash efficiency can range from $-\infty$ to 1, with 1 corresponding to a perfect match of the model to the observed data and a negative value implying that the model performs worse than a constant model with a value equal to the average of all the observations. Following Habets et al. (2008) we considered an efficiency of 0.6 to be a good score and 0.5 to be a reasonable score. The median Nash scores were calculated for all the stations. The median is a more appropriate metric than the mean as it is less sensitive to extreme outliers and is a better indicator for highly skewed distributions (Moriasi et al., 2007). These issues were present in this study due to some stations being heavily affected by anthropogenic water management or unresolved aquifers, despite most stations being well simulated. The validation period extended from August 2007 to July 2014, with the hydrological year running from August to July.

The SIM domain consists of 9892 gridpoints, of which 8602 are based in France. The remaining 1290 points are based in mountainous regions bordering the French mainland, including most of Switzerland (see Habets et al. (2008) for details). The LSM does not model horizontal exchanges, but MODCOU takes into account horizontal streamflow. Therefore it is important to include these external points in SIM because they impact the streamflow over France, particularly in the Rhone basin in the southeast. However, we only applied the SEKF over the 8602 points in the LDAS France domain. Fig. 1 shows a flowchart of SIM and how LDAS France was connected with ISBA-A-gs in SIM. Fig. 2 shows the river network used by MODCOU and the 546 stations used to validate the discharge. A map of the subset of 67 stations with low anthropogenic influence can be found in Figure S1.1 of the supplement.

## 3 Results

### 3.1 Impact of model and forcing bias-corrections on SIM

To begin with we examine the influence of the different model simulations (NIT, $NIT_m$ and $NIT_{bc}$) on the LAI evolution for the four dominant vegetation patches. We can then link the hydrological performance to each simulation. Over France, the four dominant vegetation patches are grasslands (32%), C3 croplands (24%), deciduous forests (20%) and coniferous forests (12%). Fig. 3 shows the monthly averaged LAI model simulations and observations for the gridpoints that contain at least 50% of the dominant vegetation types. The 50% threshold was used because no points contain more than 70% of deciduous forests, while over 1000 gridpoints contain at least 50% of any vegetation type. Table 2 shows the average LAI scores over France (RMSD, CC and bias) for each of the model simulations.

Firstly we examine the LAI performance for the NIT simulation, which dynamically estimates the LAI evolution. Fig. 3 shows that the NIT simulation is close to the observations for the deciduous forests (Fig. 3(a)). However, the growth and senescence phases are delayed for the simulated C3 crops and grasslands (Fig. 3(c) and (d)) compared with the observations. Furthermore, the grassland LAI is significantly understimated by NIT in winter. It is clear in Fig. 3 that imposing this higher minimum LAI value ($NIT_m$) significantly increases the LAI for grasslands in winter and improves the fit to observations. This is reflected by better scores for $NIT_m$, reducing (increasing) the RMSD (CC) by about 4% compared with NIT. Fig. 4 shows the average annual LAI minimum over France for the original simulation (NIT), the new simulation ($NIT_m$) and the GEOV1 data. Fig. 4 emphasizes that the LAI minimum is underestimated (compared to the GEOV1 data) over much of France for NIT. By increasing the grassland LAI minimum from 0.3 m$^2$/m$^2$ to 1.2 m$^2$/m$^2$ the model agrees much better with the data over most regions. Finally, the benefit of the bias-correction ($NIT_{bc}$) on LAI is also demonstrated in Fig. 3. The bias-correction has little impact on the LAI of the deciduous and coniferous forest patch types. However, it does reduce the phase errors for both the C3 crops and grassland patches. This results in significantly better LAI scores, reducing (increasing) the RMSD (CC) by about 10% compared with $NIT_m$.

The WG1 scores for the various simulations are given in Table 3. Recall that the SSM observations are linearly rescaled such that their mean and standard deviation match the NIT model simulation of WG1, which removes any bias already present.

Changing the model simulation has little impact on the scores, which suggests that the LAI evolution and the radiative forcing have a relatively small influence on the moisture content of the surface layer.

Next, the Nash efficiency scores for the different model simulations are displayed in Fig. 5 (a), showing the percentage of gauging stations at efficiency scores between 0 and 1.0. For the NIT simulation, about 26% of the stations have a score above
0.6 (a good score), 42% of the stations have a score above 0.5 (a reasonable score) and 79% of the stations have a positive Nash score. These scores are significantly improved by increasing the LAI minimum and by bias-correcting the radiative forcing. For the $NIT_m$ ($NIT_{bc}$) simulation about 31% (42%) of stations reach a score of at least 0.6, 48% (58%) of stations reach a score of 0.5 or higher and 80% (83%) of the stations have a positive score. Table 4 shows the median Nash scores for each simulation. The median Nash scores for NIT are increased by about 9% for $NIT_m$ and further increased by 18% for $NIT_{bc}$.
The median discharge ratio between the modelled ($Q_s$) and observed ($Q_o$) discharge is also shown for each simulation. A value that is greater (smaller) than 1.0 indicates a positive (negative) bias in the model. NIT has a median discharge ratio of 1.19, which indicates that the simulated streamflow is over-estimated by about 20%. This is reduced to 1.15 by applying the LAI minimum and further reduced to 1.02 by applying the bias-correction. Therefore it appears that the bias in the discharge ratio has an important impact on the Nash score, with larger biases corresponding to smaller Nash scores. This is clarified when
comparing the annual median Nash scores (Fig. 6(a)) with the annual median discharge ratios in (Fig. 6(b)). It seems that the size of the bias in the discharge ratio is negatively correlated with the Nash score, which would explain why $NIT_{bc}$ performs so well. Fig. 6(c) and (d) show the average annual temperature and rainfall respectively. There does not appear to be a strong correlation between either the temperature or rainfall and the Nash score.

The Nash efficiency for NIT for each station over France is shown in Fig. 2. The river discharge is well simulated over
most areas, but the southeast and northern regions have generally negative scores (shown in black). In southeast France this is related to a large number of dams in the alps, which are not simulated by MODCOU. In northern France, this is linked to a large aquifer that is also not taken into account by MODCOU (see Habets et al. (2008) for details). There are a small number of stations with negative scores elsewhere, which could also be related to anthropogenic water management. The maps show similar patterns for the other simulations (not shown). The vast majority of stations ($> 80\%$) for $NIT_{bc}$ are improved relative
to NIT, including most of the stations with negative scores. A scatter plot of the Nash efficiency scores of NIT and $NIT_{bc}$ for all the stations can be found in Figure S1.2(a) of the supplement.

Finally, we investigate the influence of the model simulations on the evapotranspiration, drainage and runoff fluxes in order to explain the differences in the SIM discharge. Figures 7(a-e) show the average monthly LAI, WG2, evapotranspiration, drainage and runoff respectively, averaged over France. The $NIT_m$ simulation has a greater average LAI in winter than NIT because
the NIT LAI minimum is under-estimated. The effect of a higher LAI minimum is to enhance evapotranspiration in winter and spring, which reduces the soil moisture and therefore diminishes the drainage and runoff. The consequence of increased radiative forcing in $NIT_{bc}$ is to further increase evapotranspiration and lower WG2 during much of the year. This substantially reduces drainage and runoff, especially from October to June. These effects are emphasized in Fig. 8(a), which shows the difference between the sum of drainage and runoff for the different simulations compared with NIT. The reduced drainage

and runoff feeding into the MODCOU hydrogeological model results in less river discharge, which explains the reduced river discharge bias and superior Nash scores for $NIT_m$ and $NIT_{bc}$ relative to NIT in Table 4.

## 3.2   Impact of DA on SIM

The performance of the DA runs on the LAI and WG1 scores are shown in Tables 2 and 3 respectively. LDAS1 significantly
improves the fit of the simulated LAI to the LAI observations compared to NIT. We investigate the influence of DA on the drainage and runoff fluxes in Fig. 7(f-j), which is equivalent to Fig. 7(a-e) except that LDAS1 and LDAS2 are compared with NIT. Fig. 7(g) demonstrates that the assimilation of LAI reduces the LAI phase errors in NIT, indicating that the SEKF is working effectively during much of the year. However, the LAI assimilation with the SEKF does not address the problem of the underestimated LAI in winter, unlike $NIT_m$ in Fig. 7(b). Fig. 8(b) shows the differences between the combined drainage
and runoff fluxes between NIT and the DA methods. The LAI assimilation has a relatively small influence on the drainage and runoff fluxes in Fig. 8(b) compared to $NIT_m$ in Fig. 8(a). The small positive correction of LAI in spring slightly increases (reduces) evapotranspiration (drainage and runoff) which is cancelled out by the opposite effect in autumn. Overall, LDAS1 does not significantly modify the discharge ratio or the Nash scores.

The LDAS2 experiment slightly improves the WG1 scores relative to NIT (Table 3). The median Nash discharge scores
are degraded by about 7% for LDAS2 compared to NIT (Fig. 5(b) and Table 4) and the positive bias in the discharge ratio is increased by about 2% (Table 4). The reason for this is that LDAS2 has a higher average WG2 relative to NIT (Fig. 7(f)), which translates to augmented drainage and runoff for LDAS2. This is emphasized by comparing the combined drainage and runoff for LDAS2 relative to NIT in Fig. 8(b). The extra water in the rivers exacerbates the Nash discharge bias already present in NIT, resulting in degraded Nash efficiency scores. The LDAS2 scores are degraded for about 70% of the stations relative to
the NIT simulation and a scatter plot of the scores for all the stations can be found in Figure S1.2(b) of the supplement.

The indifferent impact of LDAS1 and the destabilizing influence of LDAS2 on the soil moisture fluxes is explained in the following section by examining the observation operator Jacobians.

## 3.3   Examining the SEKF Jacobians

The SEKF observation operator Jacobians are governed by the physics of the model. Their examination is important in order
to understand the SEKF performance. The LAI increments for LDAS1 are mainly driven by the $\frac{\partial LAI}{\partial LAI}$ Jacobian. The behaviour of the $\frac{\partial LAI}{\partial LAI}$ Jacobian values for ISBA-A-gs was investigated by Rüdiger et al. (2010). Their behaviour can be split into three distinct types, which depend on the atmospheric conditions. The type "O" Jacobian is strictly equal to zero and occurs mainly in the winter when the vegetation is dormant. In this case the LAI will instantaneously return to its default model minimum. The type "A" Jacobian represents a fraction between zero and one and is correlated with the LAI value itself. It occurs during
periods of vegetation growth i.e. predominantly in spring. The type "B" Jacobian is equal to 1.0 and takes place during periods of low vegetation growth or high mortality, which occurs mainly in autumn. The grassland Jacobians are plotted for LDAS1 in Fig. 9 for a particular point in southwest France (43.35° N, 1.30° E). Also plotted in the same graph are the LAI values themselves, with the minimum indicated by the red line. Indeed, the type O Jacobians tend to occur in winter, during which time

the LAI instantaneously returns to its minimum value of 0.3 m²/m². The type A and B Jacobians tend to occur in spring and autumn respectively. These findings are in agreement with Fig. 4 of Rüdiger et al. (2010). The LAI performance for LDAS1 can now be explained by these Jacobian values. Figure 7(g) shows that during the winter the lowest LAI values are barely corrected by LDAS1 because, as shown in Fig. 9, the LAI is frequently forced back to its minimum value (type O Jacobians).

During the spring there is a small correction (type A Jacobians) and during the autumn there is a much larger correction (type B Jacobians). Hence the LDAS1 is able to correct the LAI phase errors to some extent, but LDAS1 is unable to correct the LAI minimum in winter. Since most of the drainage and runoff is present in winter and spring, the assimilation of LAI has little influence on SIM. Therefore it is much more effective to correct the LAI minimum parameter for grasslands directly than to correct the minimum using DA.

The $\frac{\partial \text{LAI}}{\partial \text{WG2}}$ Jacobian has generally positive values, since an increase in water content in the soil generally enhances photosynthesis and plant growth (not shown). However, this term is close to zero from about November to March while the vegetation is dormant. Therefore it does not significantly influence the LAI minimum in winter.

   The WG2 analysis increments for LDAS2 are largely driven by the $\frac{\partial \text{WG1}}{\partial \text{WG2}}$ Jacobian. A scatter plot of these Jacobian values against the WG1 variable is shown in Fig. 10 for the same point as Fig. 9 in Southwest France. The density of the points is

derived from the kernel density estimation of Scott (1992). There are two dense regions when WG1 is equal to 0.15 and 0.30 m³/m³, which occur because WG1 is a thin layer, and therefore most of the time it is either dry or close to saturation. The WG1 and $\frac{\partial \text{WG1}}{\partial \text{WG2}}$ values are negatively correlated, with larger values of WG1 corresponding to smaller values of $\frac{\partial \text{WG1}}{\partial \text{WG2}}$. This implies that when rain is detected in the model but not in the SSM observations, the analysis increment will be smaller than when the rain is missed by the model but detected by the observations. Indeed, the average WG2 analysis increment for a positive

innovation is $0.7 \times 10^{-3}$ m³/m³, while the average increment for a negative innovation is $-0.5 \times 10^{-3}$ m³/m³. This imbalance in the analysis increments leads to a net uptake of water in WG2, which induces the positive bias in the SIM river discharge. This problem was already highlighted by Draper et al. (2011). The Jacobians exhibited similar patterns of behaviour for other vegetation types than grasslands and across other points in France, albeit with different magnitudes (not shown).

### 3.4   Additional experiments

Additional experiments were performed to examine whether the poor performance of the SEKF was related to other factors than the Jacobians, namely the quality control of the observations, the underestimated LAI minimum or the bias in the atmospheric forcing. It is evident in Tables 2 to 4 that applying the additional quality control of the SSM observations (LDAS2$_{QC}$) does not significantly modify the LAI, WG1 or Nash discharge scores compared to LDAS2, despite removing about 10% of the SSM observations. Figure 11(a) shows only small differences in the Nash efficiency percentages between LDAS2 and LDAS2$_{QC}$.

As expected, the LDAS1$_{bc}$ and LDAS2$_{bc}$ experiments improved on the LAI scores of LDAS1 and LDAS2 (Tables 2), but did not improve on the WG1 scores in Table 3. These changes are a similar order of magnitude to the improvement of NIT$_{bc}$ over NIT. In terms of discharge Nash efficiency scores, LDAS1$_{bc}$ performed similarly to NIT$_{bc}$ and LDAS2 performed significantly worse than NIT$_{bc}$ (Table 4). The Nash efficiency percentages are shown in Fig. 11(b). The comparison between LDAS1$_{bc}$ and LDAS2$_{bc}$ with NIT$_{bc}$ in Fig. 11(b) is analogous to the comparison between LDAS1 and LDAS2 with NIT in Fig. 5(b).

The scores for the subset of 67 stations with low anthropogenic influence are also shown in Table 4. The scores for this subset are improved relative to the 546 stations in Table 4, as expected. In particular, the percentage of stations with good scores (Nash efficiency > 0.6) is increased significantly. For the interested reader, scatter plots of the Nash scores for the 67 stations are shown in Figure S1.3 in the supplement. The discharge bias is also slightly smaller for the stations with low anthropogenic influence relative to the 546 stations. This suggests that a small part of the positive bias in the discharge ratio of the NIT simulation for the 546 stations could be attributed to abstractions not being accounted for, such as drinking water or irrigation. However, most of the discharge bias in the NIT simulation is still present with the 67 stations with low anthropogenic influence. Moreover, the relative performances of the experiments are very similar. Therefore, the conclusions of the experiments are not affected by the ability of SIM (with or without DA) to simulate anthropogenically influenced streamflow. These results confirm that the inability of the SEKF to improve the soil moisture fluxes is an artifact of the SEKF Jacobians.

## 4 Discussion

Previous work by Muñoz Sabater et al. (2007) and Fairbairn et al. (2015) clearly demonstrated that the assimilation of SSM observations with an SEKF can improve WG2 with the 3-layer ISBA-A-gs model. Barbu et al. (2014) also demonstrated that the assimilation of LAI reduces phase errors in the modelled LAI evolution. However, in this work we showed that the SEKF has little influence on the drainage and runoff fluxes when assimilating LAI observations (LDAS1 experiment). Furthermore, the SEKF actually degrades the fluxes when assimilating SSM and LAI observations (LDAS2 experiment). The differences in these findings are not suprising because the nonlinear interactions in LSMs can cause the assimilation of one state variable to destabilize other soil moisture processes (Walker and Houser, 2005). The poor results for LDAS1 and LDAS2 can be explained by model errors, atmospheric forcing errors and model nonlinearities near the soil moisture wilting point and field capacity thresholds, none of which are captured by the SEKF observation operator Jacobians.

Firstly, we examine the poor performance of LDAS1, which is linked to the the inability of LDAS1 to correct the lowest LAI values during winter dormancy. The LAI Jacobian ($\frac{\partial \text{LAI}}{\partial \text{LAI}}$) was frequently equal to zero during winter and therefore the LAI returned instantaneously to its incorrect minimum value after the analysis update. These Jacobian values are physically sensible, since the vegetation is dependent on the atmospheric conditions and is often dormant during the winter period. The problem is related to the lack of a model error term in the SEKF. The lowest LAI values could be corrected further with a full EKF and a model error term, but it would be complicated to parameterize the model-error covariance matrix because the LAI minimum is linked to several factors concerning the atmospheric conditions and the vegetation type. Moreover, LAI is only assimilated every 10 days so the model LAI would drift back to its underestimated minimum value between cycles. An effective and far simpler alternative was demonstrated in the experiments, which is to impose a higher LAI minimum parameter in the model. It should also be recognised that spatial variability in the LAI minimum is not currently taken into account. Figure 4 demonstrates that there is significant heterogeneity in the LAI minimum over France. Therefore work is underway to incorporate a spatially variable LAI minimum parameter into the model, based on satellite observations.

Next we examine the poor performance of LDAS2. It is important to point out that it is physically sensible for WG1 to decouple from WG2 during precipitation events. The precipitation forcing leads to a saturation of the surface layer and subsequently WG1 becomes less dependent on WG2. The degradation of drainage and runoff is caused by limitations in the SEKF and the land surface model. Firstly, as recognised by Draper et al. (2011), a significant problem is that the SEKF is not designed to capture the uncertainty in the model and the precipitation forcing, which should increase during precipitation events and therefore compensate for the smaller Jacobians. The SAFRAN precipitation forcing performs well for a mesoscale analysis and has a higher spatial resolution than global satellite products such as ERA-interim (Quitana-Ségui et al., 2008; Vidal et al., 2010). However, by design the precipitation is assumed to be homogeneous over 615 specified climate zones. Errors are therefore introduced from the spatial heterogeneity of the precipitation, particularly in mountainous regions (Quitana-Ségui et al., 2008). This problem could more easily be addressed with an EnKF than an SEKF because an EnKF can stochastically represent model and precipitation errors (Maggioni et al., 2012; Carrera et al., 2015). Secondly, the assimilation is performed in a very shallow surface layer (0-1 cm depth). This is not sensible because WG1 is highly sensitive to the atmospheric forcing during the 24 hour assimilation window and the influence from WG2 (from capillarity rises) is drowned out during a rainfall event. This problem could be alleviated by using a multi-layer diffusion model (ISBA-DIF, (Decharme et al., 2011)), which would allow the assimilation of SSM observations into a slightly deeper layer (1-5 cm depth). This is possible because the SWI-001 data used in this study represent a layer up to 5 cm depth. This deeper layer would be less sensitive to the precipitation forcing during a 24 hour assimilation window than the 1 cm deep surface layer. Parrens et al. (2014) evaluated the assimilation of SSM with an SEKF for the 11-layer version of ISBA-DIF for a single grassland site in southwest France. The WG2 performance was enhanced by assimilating soil moisture observations into a slightly deeper layer. Thirdly, the 3-layer ISBA model has strong nonlinearities near the soil moisture thresholds, some of which are unrealistic features of the model. During dry conditions in summer the SEKF $\frac{\partial \text{WG1}}{\partial \text{WG2}}$ Jacobian can be excessive. This is linked to a rapid increase in transpiration when water is added to WG2 following dry conditions (Draper et al., 2009; Fairbairn et al., 2015). This in turn increases WG2, leading to greater drainage/runoff, which would have exacerbated the drainage and runoff bias in our study. The origin of this nonlinearity is partly related to an unrealistic feature of the surface energy balance. One single surface temperature is used to represent the vegetation and the surface layer, which causes the transpiration to increase too quickly after water is added to WG2 (Draper et al., 2009; Mahfouf, 2014). This problem could be relieved to some extent by introducing the new version of ISBA with a multiple energy balance. Additionally, Decharme et al. (2011) found that the drainage and runoff of the 3-layer ISBA model is much more abrupt than with the ISBA-DIF multi-layer model because water transfers in the root-zone are more gradual with multiple layers. Therefore, we expect the problems with the SEKF Jacobians to be less severe with ISBA-DIF and the multiple energy balance version.

There are DA methods such as particle filters that are designed to handle model nonlinearities directly. Moradkhani et al. (2012) demonstrated that good results on a hydrological model could be achieved with a particle filter with about 200 members. However, this it is substantially more computationally expensive than an EnKF, which typically requires about 20 members to overcome sampling error problems for LSMs (Maggioni et al., 2012; Carrera et al., 2015; Fairbairn et al., 2015). Moreover,

particle filters are much less effective when there are significant model and forcing errors because the non-Gaussian pdf they generate is no longer accurate.

Finally, in order to be consistent with Draper et al. (2011) and to demonstrate known deficiencies in the radiative forcing, we used the original version of SIM in our study. The original version of SIM has recently been upgraded to incorporate a direct bias-correction of the underestimated radiative forcing, the multi-layer ISBA-DIF land surface model and the introduction of a sub-grid scale hydrogeological model specifically for mountainous regions. The bias-correction of the radiative forcing is not homogeneous as in our experiments, but varies depending on the altitude and the cloud cover. A comparison of the original and new versions of SIM is expected to be published shortly (Patrick Le Moigne, personal communication). Furthermore, in our experiments we did not perform the DA for the regions in the SIM domain outside of France (shown in Fig. 2). Instead these regions used the model drainage and runoff. This would not have influenced the WG1 and LAI scores over France because SURFEX does not model horizontal exchanges. However, the MODCOU river discharge in the Rhone basin in southeast France is partly influenced by mountain rivers in Switzerland (Habets et al., 2008). Given that the modelled discharge in this region is generally unreliable because of the numerous dams, we would not expect the assimilation of data over Switzerland to substantially change the results in this study. In future applications we intend to use the new version of SIM with an extension of LDAS to the full SIM domain.

## 5  Conclusions

This study assessed the impact of assimilating surface soil moisture (SSM) and leaf area index (LAI) observations into the ISBA-A-gs land surface model (LSM) within the SAFRAN-ISBA-MODCOU (SIM) hydrological suite. The drainage and runoff outputs from the LSM were used to force the MODCOU hydrogeological model and were validated by comparing the simulated streamflow with over 500 river-gauge observations over France during several years. To our knowledge, this is the first article to perform an integrated validatation of LAI assimilation using a distributed hydrological model. Previous work already demonstrated that the SAFRAN atmospheric forcing underestimates short-wave and long-wave radiation by approximately $5\%$ averaged over France. We found in this study that the ISBA-A-gs model significantly underestimates the LAI for grasslands in winter compared with the observations. These issues resulted in an underestimation (overestimation) of evapotranspiration (drainage and runoff). The excess water flowing into the rivers and aquifers caused an overestimation of the SIM river discharge.

We tried to overcome these problems with four different experiments: (i) a correction of the grassland minimum value from 0.3 m$^2$/m$^2$ to 1.2 m$^2$/m$^2$ over France, (ii) a homogeneous bias-correction of the direct radiative forcing (+$5\%$) over France, (iii) the assimilation of LAI observations and (iv) the assimilation of SSM and LAI observations. The DA for (iii) and (iv) was performed with the SEKF, which uses finite differences in the Jacobian calculations in order to extract information from the observations to the prognostic variables (LAI and WG2). The assimilation of SSM observations in experiment (iv) was not expected to significantly reduce the errors caused by the systematic model and forcing deficiencies because the observations

were scaled such that the mean and standard deviation matched the model climatology. Nevertheless, it was designed to correct short term errors in SSM and was therefore expected to influence short-term changes in drainage and runoff.

Experiment (i) improved the median SIM Nash scores by $9\%$ because increasing the LAI minimum resulted in greater evapotranspiration in winter/spring, which subsequently reduced the drainage and runoff fluxes. Furthermore, experiment (ii) enhanced the median Nash scores by $18\%$ because increasing the radiative forcing significantly increased evapotranspiration during much of the year, which also reduced the drainage and runoff fluxes. Despite considerably reducing the LAI phase errors, experiment (iii) had no significant impact on the discharge Nash efficiency of SIM. This was explained by the SEKF LAI Jacobian, which spreads information from the LAI observations to the LAI prognostic variable. In accordance with Rüdiger et al. (2010), the LAI Jacobian values vary seasonally and are generally small in winter and spring. The LAI minimum was significantly underestimated in winter, but the small Jacobians dampened the analysis increment during this period and therefore prevented any significant correction. These Jacobians were physically reasonable because the vegetation was dormant in winter. The main problem was the underlying assumption made by the SEKF that the model is perfect.

Experiment (iv) resulted in spurious increases in drainage and runoff, which degraded the SIM discharge Nash efficiency by about $7\%$. In accordance with Draper et al. (2011), this problem could be traced back to the SEKF Jacobian linking WG1 with WG2. This Jacobian value was negatively correlated with WG1 itself. This resulted in large analysis increments when rainfall was detected in the surface soil moisture observations but was missed by the model, and small increments when rainfall was detected by the model but was missed by the soil moisture observations. This imbalance led to a build up of water in the WG2 analysis that was then lost through drainage and runoff, inducing a positive bias in the SIM discharge. There are three main limitations related to the SEKF and the LSM, which could explain these results. Firstly, the SEKF cannot account for model and precipitation errors. This problem could more easily be addressed with an ensemble Kalman filter (EnKF) because it can stochastically capture these errors in the ensemble spread. An EnKF with this capability is currently being developed at Météo-France. Secondly, the assimilation of SSM is performed in a very shallow surface layer (0-1 cm depth). This layer is highly sensitive to the atmospheric forcing during a 24-hour assimilation window. It is recommended that the assimilation be performed in a slightly thicker layer (1-5 cm depth), which would be less sensitive to the atmospheric forcing. Thirdly, the 3-layer ISBA model has strong nonlinearities near the soil moisture thresholds, some of which are unrealistic features of the model. These nonlinearity problems could be alleviated by improving the coupling between the surface and root-zone model layers with a new multi-layer diffusion model and the multi-energy balance version.

A seperate validation using a subset of 67 stations with low-anthropogenic influence confirmed the findings above and also demonstrated that MODCOU slightly over-estimates the streamflow over much of France. This over-estimation could be attributed to abstractions (such as drinking water and irrigation) not being accounted for in the MODCOU simulation.

Finally, the results highlight the important role that vegetation plays on the hydrological cycle. The correction of the LAI minimum in this study was based on a homogeneous value of 1.2 m$^2$/m$^2$. Work is already underway to provide a more realistic and spatially variable LAI minimum for grasslands based on observations.

*Author contributions.* TEXT

*Acknowledgements.* This work is a contribution to the IMAGINES (grant agreement 311766) project, co-funded by the European Commission within the Copernicus initiative in FP7. The work was also funded by the EUMETSAT H-SAF service. Discussions with Patrick Le Moigne were useful for understanding the SIM hydrological model. Useful feedback was also obtained through discussions with DA scientists at the Met Office. We would like to thank the two anonymous reviewers for their constructive comments. We would also like to thank Dr Jean-Philippe Vidal from IRSTEA for his useful comments and suggestions regarding anthropogenic water management in the SIM hydrological model.

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

**Table 1.** List of experiments. The bias-correct forcing option implies an increase of the direct short-wave and long-wave radiation by 5%. The SSM outliers removal applies to SSM observations outside the 90% confidence interval of the model.

| Experiment | LAI grassland min ($m^2/m^2$) | Bias-correct forcing | DA | SSM outliers removal |
|---|---|---|---|---|
| NIT | 0.3 | No | No | – |
| $NIT_m$ | 1.2 | No | No | – |
| $NIT_{bc}$ | 1.2 | Yes | No | – |
| LDAS1 | 0.3 | No | LAI | – |
| LDAS2 | 0.3 | No | LAI+SSM | No |
| $LDAS1_{bc}$ | 1.2 | Yes | LAI | – |
| $LDAS2_{bc}$ | 1.2 | Yes | LAI+SSM | No |
| $LDAS2_{QC}$ | 0.3 | No | LAI+SSM | Yes |

**Table 2.** Scores for LAI (prognosic variable compared with observations) averaged over 2007-2014. The closest fit to the observations is shown in bold font.

| Experiment | RMSD ($m^2/m^2$) | CC | Bias ($m^2/m^2$) |
|---|---|---|---|
| NIT | 1.18 | 0.56 | 0.11 |
| $NIT_m$ | 1.14 | 0.58 | 0.25 |
| $NIT_{bc}$ | 1.02 | 0.63 | 0.17 |
| LDAS1 | 0.69 | 0.82 | -0.08 |
| LDAS2 | 0.71 | 0.81 | -0.04 |
| $LDAS1_{bc}$ | **0.63** | **0.84** | -0.04 |
| $LDAS2_{bc}$ | 0.66 | 0.83 | -0.02 |
| $LDAS2_{QC}$ | 0.72 | 0.81 | 0.02 |

**Table 3.** Scores for WG1 (prognosic variable compared with observations) averaged over 2007-2014. The closest fit to the observations are shown in bold font.

| Experiment | RMSD ($m^3/m^3$) | CC | Bias ($m^3/m^3$) |
|---|---|---|---|
| NIT | 0.051 | 0.77 | 0.00 |
| $NIT_m$ | 0.049 | 0.77 | 0.00 |
| $NIT_{bc}$ | 0.051 | 0.77 | 0.00 |
| LDAS1 | 0.049 | 0.77 | 0.00 |
| LDAS2 | **0.048** | **0.78** | 0.00 |
| $LDAS1_{bc}$ | 0.049 | 0.77 | 0.00 |
| $LDAS2_{bc}$ | 0.049 | **0.78** | 0.00 |
| $LDAS2_{QC}$ | **0.048** | **0.78** | 0.00 |

**Table 4.** Median Nash efficiency (NE) and discharge ratio (Qs/Qo) scores over the 546 river gauges over France and for the subset of 67 gauges with low anthropogenic influence, calculated over 2007-2014. Also shown are the percentage of stations with a Nash score above 0.6. The best scores are shown in bold font.

| Experiment | NE for 546/67 stations | Discharge ratio for 546/67 stations | % stations with NE $> 0.6$ for 546/67 stations |
|---|---|---|---|
| NIT | 0.44/0.48 | 1.19/1.16 | 26%/44% |
| $NIT_m$ | 0.48/0.54 | 1.15/1.12 | 30%/48% |
| $NIT_{bc}$ | **0.56/0.60** | **1.02/0.99** | **42%/59%** |
| LDAS1 | 0.44/0.48 | 1.18/1.15 | 27%/44% |
| LDAS2 | 0.41/0.45 | 1.21/1.18 | 23%/40% |
| $LDAS1_{bc}$ | **0.56/0.60** | **1.02/1.00** | **42%/57%** |
| $LDAS2_{bc}$ | 0.53/0.54 | 1.08/1.06 | 38%/53% |
| $LDAS2_{QC}$ | 0.40/0.45 | 1.21/1.18 | 21%/39% |

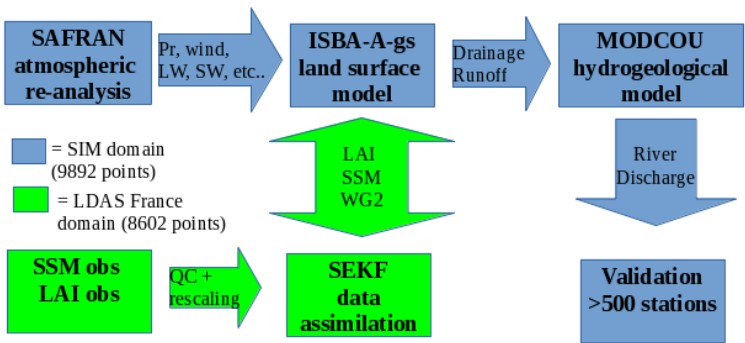

**Figure 1.** Flowchart of the SIM hydrological model and how LDAS France is connected with SIM.

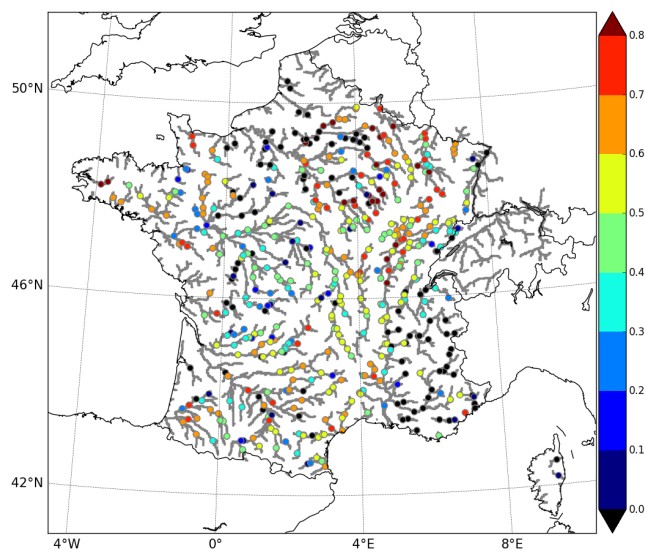

**Figure 2.** Nash efficiency scores for each station over France for the NIT simulation, calculated over the period 2007-2014. The river network is also shown.

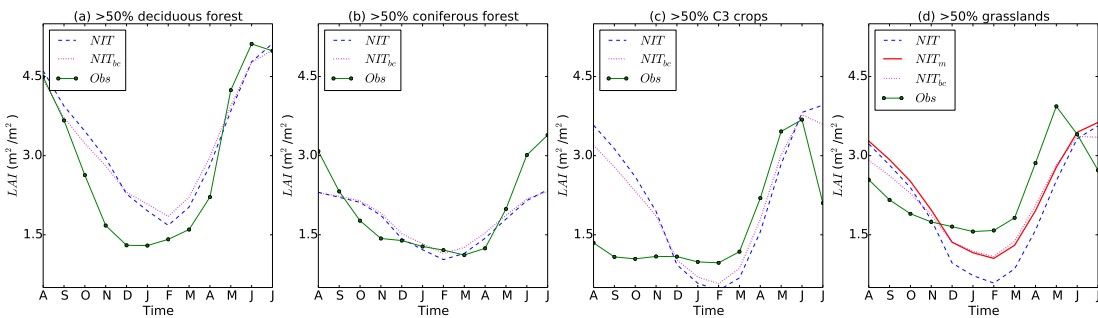

**Figure 3.** Monthly averaged LAI for the model simulations and for the gridpoints with at least $50\%$ of the four dominant vegetation types, averaged over 2007-2014 and averaged over France.

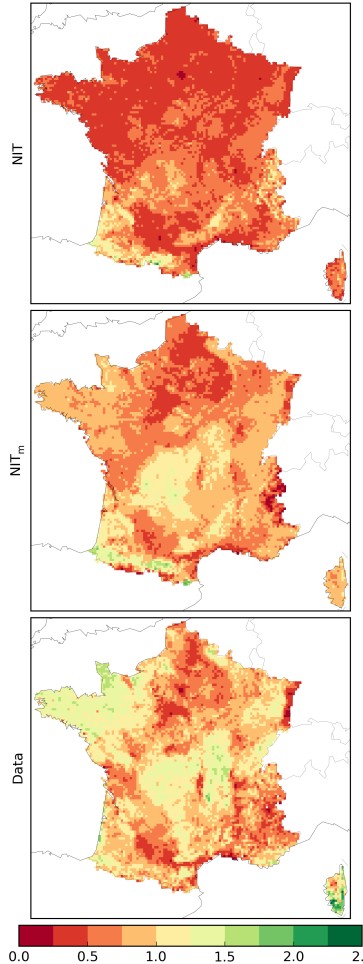

**Figure 4.** Map showing the average annual LAI minimum (2007-2014) for NIT, NIT$_m$ and the GEOV1 observations (m$^2$/m$^2$) over France.

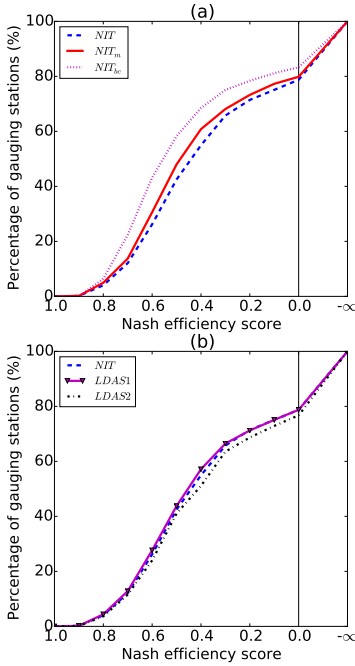

**Figure 5.** Nash efficiency scores over France for (**a**) the model simulations and (**b**) the DA methods, calculated over the period 2007-2014.

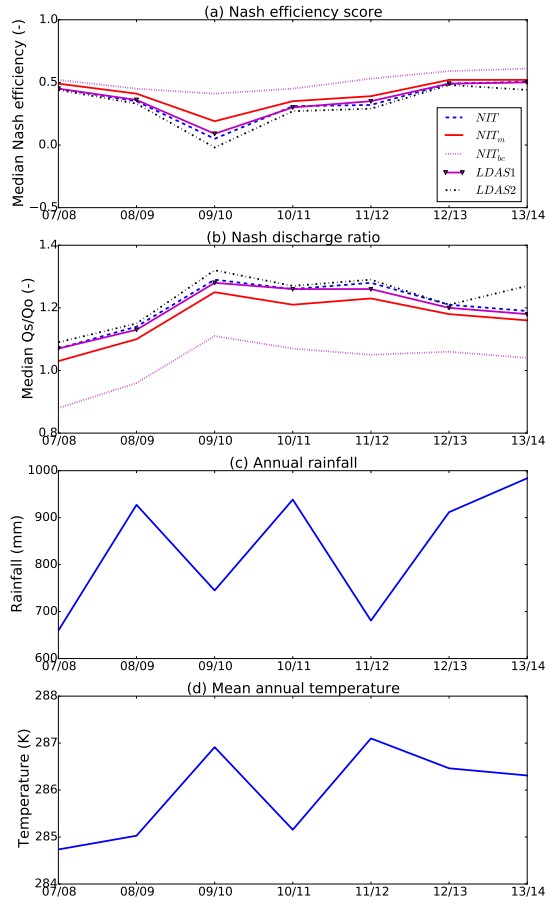

**Figure 6.** Median annual (**a**) Nash efficiency scores and (**b**) discharge ratio for each experiment. Average annual (**c**) temperature and (**d**) cumulated precipitation.

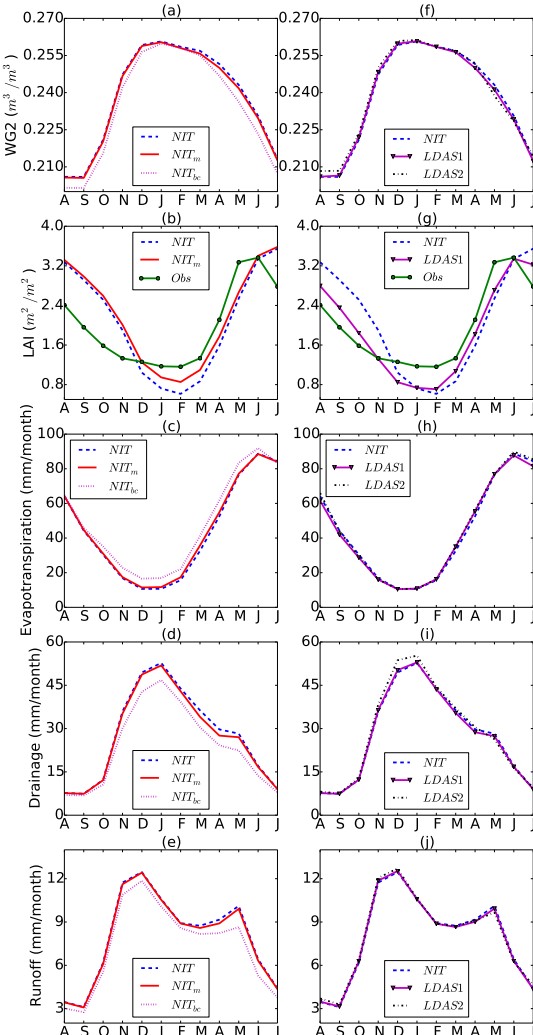

**Figure 7.** Average monthly (**a**) WG2 and (**b**) LAI; and monthly cumulative (**c**) evapotranspiration, (**d**) drainage and (**e**) runoff for NIT and the other model simulations. Plots (**f**-**j**) show NIT and the DA analyses for the equivalent variables as (**a**-**e**). Results are all averaged over the period 2007-2014 and averaged over France.

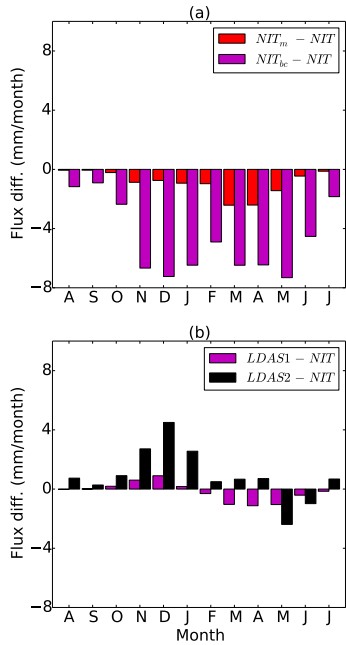

**Figure 8.** Monthly combined drainage+runoff flux differences between (**a**) NIT and the other model simulations and (**b**) NIT and the DA analyses averaged over the period 2007-2014 and over France.

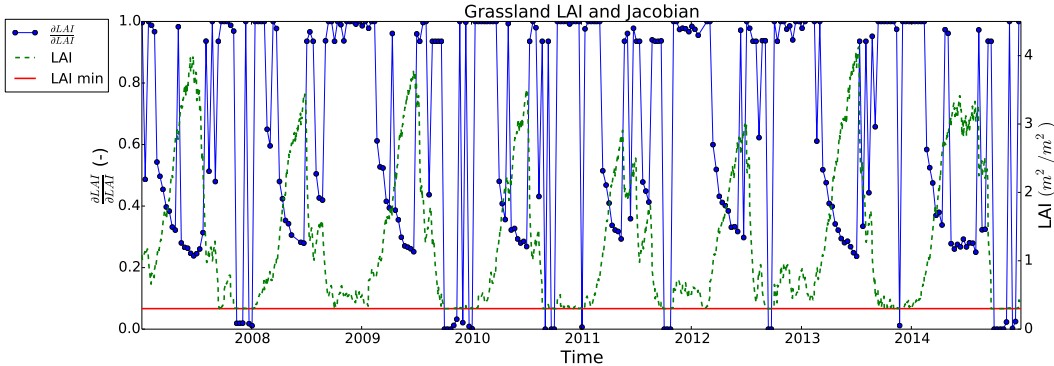

**Figure 9.** Time evolution of the LDAS1 $\frac{\partial \text{LAI}}{\partial \text{LAI}}$ Jacobian, together with the LAI analysis and the minimum LAI model parameter for the grassland patch at a point in southwest france (43.35° N, 1.30° E).

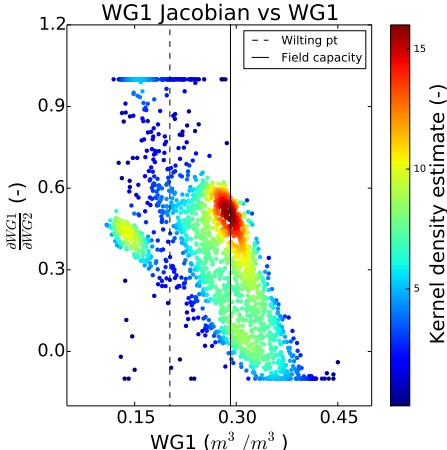

**Figure 10.** Scatter plot of WG1 against the LDAS2 $\frac{\partial \text{WG1}}{\partial \text{WG2}}$ Jacobian for the grassland patch at the same point as Figure 9.

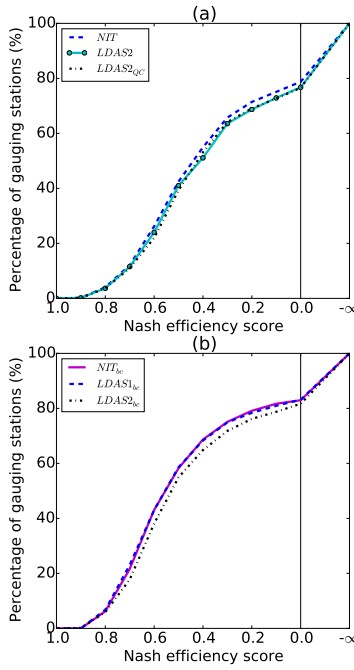

**Figure 11.** Average Nash efficiency scores over France for (**a**) the NIT, LDAS2 and LDAS2$_{QC}$ experiments and (**b**) the NIT$_{bc}$, LDAS1$_{bc}$ and LDAS2$_{bc}$ experiments.