# Peer review of "Integrated validation of assimilating satellite derived observations over France using a hydrological model"

_Hydrology and Earth System Sciences, 2016_

## Referee Comment (RC1) · Anonymous Referee #1 · 10 Jun 2016

**Version 1**

**General Comments**

This paper looks at several attempts to improve hydrological modelling for river run off when driven from a land surface model, with and without data assimilation. The validating river guages are an impressive number, across most of the river basins in france and the length of the analysis period spans several years so in my opinion can be considered robust. There have been a lot of studies over Europe looking at the impact of LAI and Soil moisture assimilation before but this is the first that i have seen that looks at impact in both the land surface model (LSM) and the hydrological runoff model. Within the paper it becomes apparent that the assimilation does not benefit the runoff, in fact in the case of soil moisture assimilation it worsens. The authors do a good job of investigative work to understand why this is, particularly focusing on the performance of the Kalman Filter Jacobians in very saturated and dry conditions. I have not seen this before to such a depth in previous published studies and this makes it novel in my opinion. There is a current focus in coupling hydrological models to limited area domains of either NWP or offline LSMs and so this work will be of interest to both the research  and operational community.

**Specific Comments**

**P2 l34** "especially near soil moisture thresholds" do you mean wilting point and saturation values? If so best to expand sentence.

**P5 l17** "The original ASCAT values are converted into SSM values..." My understandiing is that this is not correct, the ascat backscatters are converted into a soil wetness index. Is this what is assimilated in your experiments?

**Section 2.3 Data Assimilation.** Good explanation of background and observation errors for LAI, but no mention of the errors assigned to the ASCAT data. In particular  i would be interested to know if you inflate the errors to account for the oversampling issue , i.e. the same ASCAT ob covers several gridpoints.

**P10 l2** Typo on Figure number – should be Fig7?

**Section 4 Discussion.** It seems that the principle problem with the assimilation in the SEKF for this situation is that the LAI assim has little or no sensitivity during winter and the SM jacobians are unrealistically too small. One short term improvement might be to simply increase the variances in the size of the background error covariance matrix in winter which is a realistic response to the known issue of enhanced model and forcing errors. Any thoughts on this?

---

## Referee Comment (RC2) · Anonymous Referee #2 · 21 Jun 2016

This study describes the implementation of a simple Extended Kalman Filter (SEKF) to assimilate LAI and soil moisture observations into a hydrological model over France, and its validation against streamflow measurements. The topic is appropriate for the HESS journal, but the paper is not very well written. The technical approach appears sound at places and has some interesting aspects but there are many issues with the results, or at least their explanation which is not clear at all.

The NITm and NITbc simulations use a different minimum LAI (1.2 m2/m2) and a bias-corrected radiative forcing (+5%) respectively, but nothing is said about how these numbers were chosen. Was the new minimum LAI chosen based on the observations? If so, there really is no point in comparing the LAI from the new simulations with the same

data.

Additionally, the Nash scores of the NITm and NITbc simulations are shown only for the stations where at least one of the simulations had a positive score (p. 9, l. 21-22). Essentially, the average NSEs reported in Table 4 and Figure 6 are artificially better than what they ought to be since "most of the stations in northern and southeast France are excluded from this calculation". No explanation is given at to why this was done, making the discussion of the results rather dubious.

Furthermore, the assimilation doesn't appear to have much of an impact on the stream-flow simulations and actually decreases the skill (even when excluding the stations that had the negative NSE). I wonder what the rationale was of not using a more sophis-ticated data assimilation algorithm that could overcome some of the limitations in the SEKF. There are many limitations with this approach that I don't see any worthwhile scientific contribution added by this study, although there are some interesting aspects in this work.

Given these flaws, I unfortunately will have to recommend that the manuscript be re-jected. Some minor comments are outlined below, in case the editor decides on re-questing major revisions.

p. 2, l. 10: I would replace the term "network", which usually refers to in-situ measure-ments. p. 2, l. 10 "a short forecast from the past": it doesn't have to be from the past, it can be a prediction of the current time (i.e. observation time). p. 4, l. 22: can the au-thors add a sentence on what the "delayed cut-off" version of SAFRAN is? p. 5, l. 15: why were only ASCAT observations used and not SMOS for example? Is it because of the study period? p. 5, l. 21: why do the soil water index data need to be interpolated to the model resolution? Can't the SEKF handle different spatial resolutions between the model and the observations? p. 5, l. 25: has the WG1 soil moisture climatology been validated? p. 5, l. 32: were additional LAI products considered (e.g. MODIS)? p. 7, l. 25-27: this is confusing, how are the 1.2 m2/m2 and +5% values obtained? p.

8, l. 5: how are the LAI and WG1 estimates validated against satellite observations? Weren't these satellite observations assimilated into the model? p. 9, l. 28-29: I don't understand how the good performance of the NITbc is explained by the relationship between the bias in the discharge ratio and the NSE. Doesn't the NITbc just have a bias-corrected radiative forcing? Where's the causality between the simulation configuration and the performance? Wouldn't it make sense that the model with the smaller bias would have better performance in terms of NSE? p. 12, l. 14-15: but nothing is said on how the higher LAI parameter was chosen.

---

## Short Comment (SC1) · 4 Jul 2016

The authors present in this manuscript some results from experiments of data assimilation of surface soil moisture and leaf area index on streamflow simulations over France with the Safran-Isba-Modcou (SIM) hydrometeorological chain.

I will briefly comment here on a single specific issue in this study, that may however call into question all streamflow results presented in the manuscript.

The authors selected 500 gauging stations over France for comparison between observed and simulated streamflow, but they do not provide any information on the rationale behind this selection. Yet, all results from this study are based on this very

comparison, mainly through the Nash-Sutcliffe Efficiency (NSE). As a brief aside, their interpretation of the NSE detailed P8L8-9 is incorrect: a negative NSE value means that the model performs worse than a constant model with a value equal to the average of all observations.

There are three distinct but related flaws in the selection of hydrometric gauging stations here given the use of streamflow observations made by the authors:

1. They provide results as distributions of NSEs over the ensemble of stations (or a subset, but I will get back to this point later on). However, given their spatial distribution on the French river network, this distribution will be biased towards rivers along which a high number of stations are considered, like the upstream Allier river in the Massif Central mountain range (with very close yellow stations in the centre of Figure 2). This is particularly an issue when NSEs are considered, because such a criterion emphasizes the fit to high flows and these are usually highly correlated due to flood propagation along a single river like the Allier for example. This is however a relatively minor issue in my opinion compared to the next one.

2. Much more importantly, a large part of the selected stations have their observed streamflow heavily influenced by anthropogenic water management. This is somewhat and partially acknowledged by the authors P9L15-16 for southeast France. However, there are many other parts of France where anthropogenic water management stongly influences streamflow observations, through reservoir operations (see the list of dams in France, corresponding water usages, and storage capacity here: http://www.barrages-cfbr.eu/) for hydropower, irrigation, drinking water, flood and low-flow alleviation and recreation purposes, through abstractions mainly for irrigation and drinking water (see the list and quantification of all abstractions for recent years here: http://bnpe.eaufrance.fr/). This is a serious issue for this study as SIM simulates only natural streamflow, as recalled P5L12-13. As a consequence, any improvement from data assimilation in NSEs shown by the authors may be interpreted in terms of simulations closer to influenced streamflow. The data assimilation procedure may therefore
tend to drive simulations in a direction closer to the observed (influenced) streamflow but opposite to the natural (unobserved) streamflow. The overestimation of drainage and runoff by SIM recalled P1L8 and shown in Figure 6 may for example be partly attributed to the abstractions being not accounted for. Low NSE values may also be partly attributed to effects of reservoir management for reducing flood peaks, sustaining low-flows, or producing hydropower.

In order to overcome such widely recognized issues, and for disentangling climatedriven effects from local anthropogenic influences, reference hydrologic networks have been set up for different countries over the past decade or so (see e.g. Burn et al., 2012; Whitfield et al., 2012). In France, such a reference network has been set up for low flows (Giuntoli et al., 2013), with a subset also valid for high flows (Giuntoli et al., 2012). This reference network includes stations with long records, gauging catchments with low anthropogenic influence, and with high guality measurement on low (resp. high) flows. The list of stations included in this reference network, as well as corresponding streamflow data are available from the Global Runoff Data Centre (GRDC) (http://www.bafg.de/GRDC/EN/Home/homepage node.html) and identified in the GRDC as the French contribution to the "Climate sensitive stations". The record length constraint used for defining this reference network may be relaxed for the present study. It has to be noted that metadata in the French hydrometric database (http://www.hydro.eaufrance.fr/) are often too optimistic when anthropogenic influence comments are concerned, because (1) such metadata are not always filled in and (2) the default label is "Not influenced".

3. As already pointed out by Referee #2, the various NSE graphs show simulation results for only a artificially chosen (and not clearly identified spatially) subset of stations based on positive values of the NSE for any simulation. This is indeed a rather dubious way of dealing with anthropogenically influenced stations, but also with badly simulated stations with high groundwater influences (9L16-17).

Given the three points above, I would strongly recommend the authors:
1. To consider only catchments with low anthropogenic influence in order not to compare apples and oranges and avoid drawing conclusions on the ability of SIM (with or without data assimilation) to simulate anthropogenically influenced streamflow,

2. To show scatter plots of NSEs instead of distributions (possibly with marginal distributions) to reduce the potential spatial bias effect mentioned above.

References

Burn, D. H., Hannaford, J., Hodgkins, G. A., Whitfield, P. H., Thorne, R. & Marsh, T.: Reference hydrologic networks II. Using reference hydrologic networks to assess climate-driven changes in streamflow, Hydrological Sciences Journal, 57, 1580-1593. doi: 10.1080/02626667.2012.728705, 2012.

Giuntoli, I., Renard, B. & Lang, M.: Floods in France. In: Changes in Flood risk in Europe (Kundzewicz, Z. K., Ed.), 199-211. IAHS Special Publication 10. CRC Press. ISBN 9780415621892, 2012

Giuntoli, I., Renard, B., Vidal, J.-P. & Bard, A.: Low flows in France and their relationship to large-scale climate indices, Journal of Hydrology, 482, 105-118. doi: 10.1016/j.jhydrol.2012.12.038, 2013.

Whitfield, P. H., Burn, D. H., Hannaford, J., Higgins, H., Hodgkins, G. A., Marsh, T. & Looser, U.: Reference hydrologic networks I. The status and potential future directions of national reference hydrologic networks for detecting trends, Hydrological Sciences Journal, 57, 1562-1579. doi: 10.1080/02626667.2012.728706, 2012.

---

## Author Comment (AC1) · 26 Jul 2016

Fairbairn *et. al.*, (2016):

http://www.hydrol-earth-syst-sci-discuss.net/hess-2016-195/

Response to comments from Referee 1

July 26, 2016

Firstly, we would like to thank the reviewer for his/her constructive comments. A point by point response is given below.

**Response to specific comments:**

1. P2 l34 "especially near soil moisture thresholds" do you mean wilting point and saturation values? If so best to expand sentence.

   **Response**: Yes. We will replace "especially near soil moisture thresholds" with "especially near the wilting point and field capacity thresholds".

2. P5 l17 "The original ASCAT values are converted into SSM values..." My understandiing is that this is not correct, the ascat backscatters are converted into a soil wetness index. Is this what is assimilated in your experiments?

   **Response**: In order to explain this more clearly, we will replace lines 17-12 (starting with "The original ASCAT values...") with the following: "The original ASCAT values are converted into the surface degree of saturation (SDS, with values between 0 and 1) using a change detection technique, which was developed at the Vienna University of Technology (Tu-Wien) and is detailed in Wagner *et al.* (1999); Bartalis *et al.* (2007). The historically lowest and highest backscatter coefficients are assigned values for dry and saturated soils respectively. The Copernicus Global Land Service then calculates the surface water index (SWI) by applying a recursive exponential filter to these SDS values (Albergel *et al.*, 2008) using a time-scale that may vary between 1 and 100 days. The SWI represents the soil wetness over the soil profile and also has values between 0 (dry) and 1 (saturated). The longer the time-scale of the exponential filter, the deeper the

representative soil profile. In this study we use a time-scale of one day (SWI-001 product), which represents the SWI for <5 cm of soil. We then interpolate the SWI-001 data to the 8 km resolution model grid. As in Draper *et al.* (2011) an additional screening step is performed to remove observations with an altitude greater than 1500m, frozen regions and areas with an urban fraction greater than 15%."

As mentioned in lines 24-31, we apply a linear rescaling to the SWI-001 data, which scales them such that the mean and standard deviations match the WG1 layer climatology (Calvet and Noilhan, 2000; Scipal *et al.*, 2008). The rescaling is designed to remove biases between the model and the observations and in the process the SWI-001 data are converted into the same units as the model, expressed in volumetric soil moisture ($m^3/m^3$). These rescaled SSM observations are assimilated into the WG1 model layer.

3. Section 2.3 Data Assimilation. Good explanation of background and observation errors for LAI, but no mention of the errors assigned to the ASCAT data. In particular, I would be interested to know if you inflate the errors to account for the oversampling issue, i.e. the same ASCAT obs covers several gridpoints.

**Response**: On page 7, line 14 we mentioned that the SSM observation error is prescribed a value of $0.4(w_{fc}\text{-}w_{wilt})$, where $w_{fc}$ is the field capacity and $w_{wilt}$ is the wilting point (note there is a typo, we will replace "WG1" with "SSM"). The scaling by $(w_{fc}\text{-}w_{wilt})$ assumes that there is linear relationship between the soil moisture errors and the dynamic range (Mahfouf *et al.*, 2009). Averaged over France, this observation error is equal to 0.034 $m^3/m^3$. This underestimates the median SDS estimated error of 0.05 $m^3/m^3$ by Draper *et al.* (2011). We have therefore rerun the LDAS2 experiments, but with a larger SSM observation error standard deviation of $0.65(w_{fc}\text{-}w_{wilt})$. This averages to 0.055 $m^3/m^3$ over France. We used a slightly larger value than Draper *et al.* (2011) in order to account for the oversampling issue. This is comparable with observation errors expected for remotely sensed SSM observations (de Jeu *et al.*, 2008; Draper *et al.*, 2013). The LDAS2 results for this larger observation error will be included in the revised version of the paper. We will use the following description of the SSM observation error: "The SSM observation error standard deviation was set to $0.65(w_{fc}\text{-}w_{wilt})$, which is about 0.055 $m^3/m^3$ averaged over France. This value is slightly larger than the median ASCAT-derived SDS error of 0.05 $m^3/m^3$ estimated by Draper *et al.* (2011) because it also approximates the oversampling issue i.e. the same ASCAT observation covers several gridpoints."

Increasing the observation error standard deviation reduces the impact of the SSM assimilation for the LDAS2 experiments. Table R1.1 (at the end of the document) shows the new WG1 scores (prognosic variable compared with SSM observations) averaged over 2007-2014. The fit of LDAS2 to the observations is slightly reduced relative to the original results (Table 3 in the paper). However, as mentioned by the reviewer, the poor performance of the soil moisture fluxes for the SEKF was explained by the observation operator Jacobians. Therefore changing the SSM observation error does not change the conclusions of the study. Table R1.2 (at the end of this document) will replace Table 4 in the paper and shows the median Nash efficiency scores for the all the experiments. Following a comment from reviewer 2, the median Nash efficiency scores are calculated for all the stations instead of the mean. The median is a more appropriate metric for our experiments as it is less sensitive to extreme outliers and is a better indicator for highly skewed distributions (Moriasi *et al.*, 2007). Following a comment from Dr Jean-Philippe Vidal, the scores are also shown for the 67 stations with low anthropogenic influence (also see response to Dr J.-P. Vidal for details). The relative performances of the experiments in Table R1.2 are very similar to Table 4 in the paper and therefore the conclusions of the experiments remain unchanged.

4. P10 l2 Typo on Figure number  should be Fig7?

   **Response**: Yes, we have changed this.

5. Section 4 Discussion. It seems that the principle problem with the assimilation in the SEKF for this situation is that the LAI assim has little or no sensitivity during winter and the SM jacobians are unrealistically too small. One short term improvement might be to simply increase the variances in the size of the background error covariance matrix in winter which is a realistic response to the known issue of enhanced model and forcing errors. Any thoughts on this?

   **Response**: These are interesting suggestions. However, we have tried increasing the LAI variances in winter but this does not help. Part of the problem if that the LAI observations are infrequent (every 10 days). We found that the model quickly returns to its underestimated minimum value between cycles, regardless of the size of the analysis increments. Perhaps one way to tackle this problem would be to implement a Kalman smoother with a long assimilation window of 10 days, but this is beyond the scope of this study. The problem with the SSM Jacobians cannot be resolved by simply increasing the estimated background errors in winter. The problem occurs because the SSM Jacobian value is negatively correlated with WG1 itself. This results in large analysis increments when rainfall is detected in the surface soil moisture observations but is missed by the model, and small increments when rainfall is detected by the model but is missed by the soil moisture observations. This problem would still occur with larger background-error variances. A potential solution to this problem is to assimilate the SWI product into a deeper soil moisture layer, which is less sensitive to the model and atmospheric forcing over the 24 hour assimilation window. We are currently testing this idea with the new multi-layer diffusion based model (ISBA-DIF).

Table R1.1: Scores for WG1 (prognosic variable compared with observations) averaged over 2007-2014. The closest fit to the observations are shown in bold font.

| Experiment | RMSD ($m^3/m^3$) | CC | Bias ($m^3/m^3$) |
|------------|------------------|------|-------------------|
| NIT | 0.051 | 0.77 | 0.00 |
| $NIT_m$ | 0.049 | 0.77 | 0.00 |
| $NIT_{bc}$ | 0.051 | 0.77 | 0.00 |
| LDAS1 | 0.049 | 0.77 | 0.00 |
| LDAS2 | **0.048** | **0.78** | 0.00 |
| $LDAS1_{bc}$ | 0.049 | 0.77 | 0.00 |
| $LDAS2_{bc}$ | 0.049 | **0.78** | 0.00 |
| $LDAS2_{QC}$ | **0.048** | **0.78** | 0.00 |

Table R1.2: Median Nash efficiency (NE) and discharge ratio (Qs/Qo) scores over the 546 river gauges over France and for the subset of 67 gauges with low anthropogenic influence, calculated over 2007-2014. Also shown are the % of stations with a Nash score above 0.6. The best scores are shown in bold font.

| Experiment | NE for 546/67 stations | Discharge ratio for 546/67 stations | % stations with NE > 0.6 for 546/67 stations |
|---|---|---|---|
| NIT | 0.44/0.48 | 1.19/1.16 | 26%/44% |
| $NIT_m$ | 0.48/0.54 | 1.15/1.12 | 30%/48% |
| $NIT_{bc}$ | **0.56/0.60** | **1.02/0.99** | **42%/59%** |
| LDAS1 | 0.44/0.48 | 1.18/1.15 | 27%/44% |
| LDAS2 | 0.41/0.45 | 1.21/1.18 | 23%/40% |
| $LDAS1_{bc}$ | **0.56/0.60** | **1.02/1.00** | **42%/57%** |
| $LDAS2_{bc}$ | 0.53/0.54 | 1.08/1.06 | 38%/53% |
| $LDAS2_{QC}$ | 0.40/0.45 | 1.21/1.18 | 21%/39% |

**References**

C. Albergel, C. Rüdiger, T. Pellarin, J.-C. Calvet, N. Fritz, F. Froissard, D. Suquia, A. Petitpa, and B. Piguet. From near-surface to root-zone soil moisture using an exponential filter: an assessment of the method based on in-situ observations and model simulations. *Hydrol. Earth Syst. Sci*, 12:1323–1337, 2008.

Z. Bartalis, W. Wagner, V. Naeimi, S. Hasenauer, K. Scipal, H. Bonekamp, J. Figa, and C. Anderson. Initial soil moisture retrievals from the metop-a advanced scatterometer (ascat). *Geophys. Res. Lett.*, 34:10.1029/2007GL031088, 2007.

J.-C. Calvet and J. Noilhan. From Near-Surface to Root-Zone Soil Moisture Using Year-Round Data. *J. Hydrometeor*, 1:393–411, 2000.

R. A. M. de Jeu, W. Wagner, T. R. H. Holmes, A.J. Dolman, N. C. de Giesen, and J. Friesen. Global soil moisture patterns observed by space borne microwave radiometers and scatterometers. *Surv. Geophys.*, 29:399420, 2008.

C.S. Draper, J.-F. Mahfouf, J.-C. Calvet, E. Martin, and W. Wagner. Assimilation of ASCAT near-surface soil moisture into the SIM hydrological model over France. *Hydrol. Earth Syst. Sci*, 15:3829 – 3841, 2011.

C.S. Draper, R.H. Reichle, R. de Jeu, V. Naeimi, R. Parinussa, and W. Wagner. Estimating root mean square errors in remotely sensed soil moisture over continental scale domains. *Remote Sensing of Environment*, 137:288–298, 2013.

J.-F. Mahfouf, K. Bergaoui, C. Draper, C. Bouyssel, F. Taillefer, and L. Taseva. A comparison of two off-line soil analysis schemes for assimilation of screen-level observations. *J. Geophys. Res*, 114:D08105, 2009.

D.N. Moriasi, J.G. Arnold, M.W. Van Liew, R.L. Bingner, R.D. Harmel, and T.L. Veith. Model evaluation guidelines for systematic quantification of accuracy in watershed simulations. *American Society of Agricultural and Biological Engineers*, 50:885–900, 2007.

K. Scipal, M. Drusch, and W. Wagner. Assimilation of a ERS scatterometer derived soil moisture index in the ECMWF numerical weather prediction system. *Adv. Water. Resour*, 31:11011112, 2008.

W. Wagner, G. Lemoine, and H. Rott. A method for estimating soil moisture from ERS scatterometer and soil data. *Remote Sens. Environ.*, 70:191–207, 1999.

---

## Author Comment (AC2) · 26 Jul 2016

Fairbairn *et. al.*, (2016):

http://www.hydrol-earth-syst-sci-discuss.net/hess-2016-195/

Response to comments from Referee 2

July 26, 2016

Firstly, we would like to thank the reviewer for his/her constructive comments. A point by point response is given below.

**Response to major comments:**

**1**

**1.1**

**Referee comment** This study describes the implementation of a simple Extended Kalman Filter (SEKF) to assimilate LAI and SSM observations into a hydrological model over France, and its validation against streamflow measurements.

**Response:**
This is not exactly the objective of this study. It is important to mention that the assimilation is into a land surface model and not into a hydrological model. We will add this sentence to the introduction (Section 1): "In this study, the Simplified Extended Kalman Filter (SEKF) is used to assimilate LAI and SSM observations to update LAI and root-zone soil moisture (WG2) in the ISBA-A-gs land surface model. The drainage and runoff outputs from the land surface model are then used to force the MODCOU hydrogeological model and are validated by comparing the simulated streamflow with observations." It is important to clarify that the land surface model and the assimilated land surface observations are independent of the hydrogeological validation. This is different to other studies such as Thirel *et al.* (2010), where streamflow observations were assimilated and used to update the soil moisture in the ISBA land surface model. We will try to make this more clear throughout the paper, including the abstract, introduction and conclusions.

**1.2**

The topic is appropriate for the HESS journal, but the paper is not very well written. The technical approach appears sound at places and has some interesting aspects but there are many issues with the results, or at least their explanation which is not clear at all.

**Response:**

We agree with the reviewer that the experimental setup was not well explained and there were some mistakes in the way we presented the results. We hope that by offering clearer explanations we can resolve these problems.

**1.3**

The NITm and NITbc simulations use a different minimum LAI (1.2 m2/m2) and a bias-corrected radiative forcing (+5%) respectively, but nothing is said about how these numbers were chosen.

**Response:**

We admit this was not well explained and we will clarify this in the experimental setup (Section 2.4 of the paper). Fig. R2.1 (at the end of the document) shows a histogram of the observed average annual LAI minimum (GEOV1 satellite-derived observations) for the 133 grid-points over France with predominantly grasslands (the grassland patch fraction exceeding 70%). We chose an augmented grassland LAI minimum value of 1.2 $m^2/m^2$ for NITm because over 99% of the predominantly grassland points in Fig. R2.1 have an observed average annual LAI minimum above this value. The average annual LAI minimum over France for the original simulation (NIT), the new simulation (NITm) and the GEOV1 data are shown in Fig. R2.2 (note that the original Fig. 4 in the paper did not have the correct scale). Fig. R2.2 emphasizes that the LAI minimum was underestimated (compared to the GEOV1 data) over much of France for the original model simulation. By increasing the grassland LAI minimum to 1.2 $m^2/m^2$, the model agrees much better with the data over most regions. Szczypta *et al.* (2011) and Le Moigne (2002) demonstrated that the direct short-wave and long-wave radiative forcing respectively are underestimated by approximately 5% averaged over France. We followed Decharme *et al.* (2013) in bias-correcting the direct radiative forcing by +5%.

**1.4**

Was the new minimum LAI chosen based on the observations? If so, there really is no point in comparing the LAI from the new simulations with the same data.

**Response:**

The aim of this study was not to carry out an independent validation of LAI for each experiment, for which we would need independent observations. We will add the following in the experimental setup (Section 2): "A system validation was performed by comparing the LAI and WG1 states with the LAI and SSM observations respectively for all the simulations and data assimilation experiments. Note that this is not an independent validation of the performance of the system, for which we would need independent observations. The rationale was to check the effectiveness of the SEKF i.e. to see if it improved the fit between the model simulations and the observations. The fit to the observations was determined by the root mean square difference (RMSD), the correlation coefficient (CC) and the bias. These checks were important because the performance of the SEKF had an important impact on the drainage and runoff fluxes."

**1.5**

Additionally, the Nash scores of the NITm and NITbc simulations are shown only for the stations where at least one of the simulations had a positive score (p. 9, l. 21-22). Essentially, the average NSEs reported in Table 4 and Fig. 6 are artificially better than what they ought to be since most of the stations in northern and southeast France are excluded from this calculation. No explanation is given at to why this was done, making the discussion of the results rather dubious.

**Response:**

We agree with the reviewer that we did not present these results correctly and have therefore shown the results with all the stations (including the negative scores) included in the calculations. Fig. R2.3(a) shows a scatter plot of the Nash efficiency for the NIT simulation against the NITbc simulation. The density of the points is derived from the kernel density estimation of Scott (1992). The NIT simulation is the original simulation. The NITbc simulation is the new simulation with the augmented LAI minimum and radiative forcing. The results are improved for about 80% of the stations, including most of the stations with negative Nash scores. Fig. R2.3(b) shows a scatter plot of the NIT against LDAS2 (NIT with the assimilation of SSM and LAI). The assimilation degrades the SIM discharge scores for about 70% of the stations. These results are consistent with the original conclusions of the study. Note that the LDAS2 experiment

was performed with a more appropriate estimate of the observation error standard deviation than the original experiment. We used a value of $0.65(w_{fc}\text{-}w_{wilt})$ instead of the original value of $0.4(w_{fc}\text{-}w_{wilt})$, where $w_{fc}$ is the field capacity and $w_{wilt}$ is the wilting point. This averages to 0.055 $m^3/m^3$ over France. We used a slightly larger error than the estimated error of 0.05 $m^3/m^3$ by Draper *et al.* (2011) in order to approximate the oversampling issue i.e. the same ASCAT observation covers several gridpoints. This is comparable with observation errors expected for remotely sensed SSM observations (de Jeu *et al.*, 2008; Draper *et al.*, 2013). Note that this larger observation error slightly reduces the impact of the assimilation of SSM relative to the original experiment, but the conclusions of the study remain unchanged. Table R2.1 (at the end of the document) shows the new WG1 scores (model state compared with SSM observations) and will replace Table 3 in the paper.

Table R2.2 will replace Table 4 in the paper and shows the median Nash efficiency scores for the all the experiments. Following a comment from Jean-Philippe Vidal (see 1.6 below), the scores are also shown for the 67 stations with low anthropogenic influence. Note that the median is calculated rather than the mean because the majority of stations ($> 80\%$) have positive Nash efficiency scores, but a few outliers have scores near to -100. The median is a more appropriate metric as it is less sensitive to extreme outliers and is a better indicator for highly skewed distributions (Moriasi *et al.*, 2007). The results in Table R2.2 are very similar to Table 4 in the paper except that the LDAS2 experiment has less impact (due to the larger observation error). Therefore, including the stations with negative scores does not change the conclusions of this study.

In the revised version of the paper, we will replace Figure 6 with the Median Nash efficiency scores for all the stations. Note that Figure 5 in the paper was actually correctly presented and there was a mistake in the caption - all the stations were considered in the calculations, not just the stations with positive scores. Fig. R2.3 will be included in a supplement.

**1.6**

Dr Jean-Phillipe Vidal from IRSTEA posted a short comment, which is related to comment 1.5. He was right to point out that many of the stations included in the calculations are influenced by anthropogenic water management, which is not simulated by the MODCOU hydrogeological model. He was concerned that the results might be interpreted as being closer to anthropogenically influenced streamflow. He suggested the following:

1. To consider only catchments with low anthropogenic influence in order not to compare apples and oranges and avoid drawing conclusions on the ability of

SIM (with or without data assimilation) to simulate anthropogenically influenced streamflow,

2. To show scatter plots of NSEs instead of distributions (possibly with marginal distributions) to reduce the potential spatial bias effect mentioned above.

**Response:**

We have decided to follow Dr Jean-Phillipe Vidal's suggestions by showing the results for the stations with low-anthropogenic influence. We have used the suggested reference networks of Giuntoli *et al.* (2012, 2013) to extract a subset of 67 river gauges with low-anthropogenic influence from the original 546 stations, valid for both low and high flows. A map of these stations is shown in Fig. R2.4. A scatter plot is shown of the Nash efficiency of these stations (labeled as "Low anth. influence") and all the other stations (labeled as "High anth. influence") in Fig. R2.5. The same results are shown as in Fig. R2.3, but for the sake of clarity, in Fig. R2.5 only the stations are shown in the range of Nash scores -1.0 to 1.0. The 'low anth. influence' stations follow a similar pattern to the 'high anth. influence' stations. Furthermore, we calculated the Median Nash efficiency scores for the 67 stations in Table R2.2. The scores for this subset are improved relative to the 546 stations in Table R2.2, as expected. In particular, the percentage of stations with good scores (Nash efficiency > 0.6) is increased significantly. The discharge bias is also slightly less for the stations with low anthropogenic influence relative to the 546 stations. This supports Jean-Phillipe Vidal's suggestion that part of the positive bias in the discharge ratio of the NIT simulation for the 546 stations could be attributed to abstractions not being accounted for. However, the majority of the discharge bias in the NIT simulation is still present with the 67 stations with low anthropogenic influence. Moreover, the relative performance of the experiments is very similar to the original 546 stations. Therefore, the conclusions of the experiments are not affected by the ability of SIM (with or without data assimilation) to simulate anthropogenically influenced streamflow.

We will explain these results in Section 3.4 of the paper. Figures R2.4 and R2.5 will be included in a supplement.

**1.7**

Furthermore, the assimilation doesnt appear to have much of an impact on the streamflow simulations and actually decreases the skill (even when excluding the stations that had the negative NSE). I wonder what the rationale was of not using a more sophisticated data assimilation algorithm that could overcome some of the limitations in the SEKF. There are many limitations with this approach that I dont see any worthwhile scientific contribution added by this study, although there are some interesting aspects

in this work.

**Response:**

The reviewer is right that in many data assimilation applications with hydrological models, more sophisticated algorithms are commonly used that take into account the "errors of the day". We will clarify in the introduction the differences between hydrological and land surface data assimilation, and why we chose the SEKF for our experiments.

Many studies have investigated the assimilation of SSM and streamflow observations into hydrological models in order to improve streamflow predictions and hydrological parameters (Thirel *et al.*, 2010; Aubert *et al.*, 2003; Clark *et al.*, 2008; Moradkhani *et al.*, 2005, 2012). In large-scale streamflow assimilation, a DA method is typically chosen that can take into account lateral background-error covariances and flow-dependence. These features are important because streamflow has important horizontal interactions. For example, Thirel *et al.* (2010) used the Best Linear Unbiased Estimate (BLUE) method to assimilate steamflow observations into the MODCOU hydrogeological model, which they used to update soil moisture in the ISBA land surface model (LSM). Although they used a fixed diagonal background-error covariance at the start of each window, they generated implicit background-error covariances between the river sub-basins using finite differences in the observation operator Jacobian calculation. This led to improved streamflow predictions.

LSMs concern water and energy fluxes between the soil and atmosphere. Unlike hydrological models, layer-based LSMs such as the ISBA model are typically pointwise (there is no horizontal interaction between the gridpoints), since this greatly reduces the computational expense. It is also common to use a DA method with 1D Kalman filtering (where observations are used to update colocated gridpoints) as opposed to 2D Kalman filtering (where observations are used to update colocated gridpoints and neighbouring gridpoints). Moreover, a study by Gruber *et al.* (2015) over the contiguous US found that there was no advantage of 2D Kalman filtering over 1D Kalman filtering when assimilating ASCAT SSM data into a soil moisture model. They explained these results using an analytical evaluation of the impact of spatial-error autocorrelations on the steady-state Kalman gain.

In large-scale land surface DA, it is common to assimilate satellite-derived surface soil moisture (SSM) observations and screen-level temperature and humidity observations into a LSM, in order to improve soil moisture and screen-level variables. Typically WG2 is of more interest than SSM as it has a much larger water capacity and therefore has a greater influence on vegetation and water fluxes. Land surface DA is commonly performed using a 1D ensemble Kalman filter (EnKF) or a simplified extended Kalman

filter (SEKF). The SEKF simplifies the EKF by assuming that the errors in the background state are fixed and uncorrelated between gridpoints. It uses finite differences for computing Jacobians necessary to extract information from the observations to the prognostic variables.

There has been increasing interest in ensemble DA for LSMs over the last two decades (Reichle *et al.*, 2002, 2008; Zhou *et al.*, 2006; Muñoz Sabater *et al.*, 2007; Draper *et al.*, 2012; Carrera *et al.*, 2015), partly because these methods can estimate the "errors of the day" in the background-error covariance. At Environment Canada, the development of an operational EnKF using the same ISBA 3-layer model we employed is also motivated by the requirements of coupling land surface DA with NWP ensemble prediction (Carrera *et al.*, 2015). However, the correct representation of the "errors of the day" is challenging in land surface DA. A large proportion of the errors in LSMs come from the model and the atmospheric forcing, rather than the initial conditions. The integrating nature and the nonlinear interactions in LSMs mean that short-term errors dissipate over time, including random errors in the precipitation. For example, a study by Maggioni *et al.* (2011) found that errors in WG2 are not very sensitive to the rainfall error modelling approach. Indeed, experiments assimilating in situ SSM observations with the ISBA-A-gs model have demonstrated that the EnKF and the SEKF produce a WG2 analysis with comparable accuracy and both methods improve on the model simulation (Muñoz Sabater *et al.*, 2007; Fairbairn *et al.*, 2015).

Due to its efficacity, simplicity and low computational cost, the SEKF is the preferred method at several meteorological operational centres for analyzing soil moisture and screen level variables. Hess (2001) developed a simplified 2D-Var (theoretically equivalent to an SEKF) scheme for the assimilation of screen-level temperature and humidity at the German Weather service (DWD). The European Centre for Medium Range Weather Forecasts (ECMWF) assimilate screen-level temperature and humidity operationally with an SEKF (de Rosnay *et al.*, 2013). An SEKF was developed for research purposes to assimilate ASCAT satellite derived soil moisture at Meteo-France (Draper *et al.*, 2009; Mahfouf, 2010). Recently, ECMWF have also modified their SEKF to assimilate ASCAT derived SSM observations (ECMWF, 2016). At the UK Met Office, an SEKF has been developed for research purposes for the assimilation of a wide variety of observation types, including screen-level variables and satellite derived SSM observations (Candy *et al.*, 2012).

In our study, we use an SEKF to assimilate LAI and SSM observations to update LAI and WG2 in the ISBA-A-gs LSM in the SAFRAN-ISBA-MODCOU (SIM) hydrological suite. SIM consists of three stages: (1) An atmospheric reanalysis (SAFRAN) over France, which forces (2) the ISBA-A-gs land surface model, which then provides drainage and runoff inputs to (3) the MODCOU hydrogeological model. The drainage

and runoff outputs from ISBA-A-gs are validated by comparing the simulated stream-flow from MODCOU with observations. Our study is different to the hydrological studies mentioned earlier because the LSM and the DA are independent of the hydro-geological model. This study is relevant to the land surface DA community because several operational centres assimilate SSM observations using an SEKF to update WG2. Many studies have demonstrated that the force-restore dynamics of the ISBA 3-layer model can skillfully simulate soil moisture and propagate the increments downwards from the surface to the root-zone (Muñoz Sabater *et al.*, 2007; Draper *et al.*, 2009; Mahfouf *et al.*, 2009; Barbu *et al.*, 2011, 2014). An integrated validation using SIM has also demonstrated that the ISBA 3-layer model can skillfully simulate drainage and runoff fluxes over France (Habets *et al.*, 2008). But relatively few studies have assessed the SEKF performance using an integrated validation of the soil moisture fluxes. To our knowledge, this is the first article to perform this type of validatation for LAI assimilation. Moreover, the validation is robust because it is performed using more than 500 river gauges over France and the length of the analysis period spans several years.

In the discussion section we mentioned that in future studies we would like to test the EnKF over France with a stochastic representation of precipitation and model errors using a similar hydrogeological evaluation to this study. The EnKF would not be affected by some of the issues we encountered with the SEKF, including the collapse of the observation operator Jacobians during wet conditions. However, the choice of DA method is not the only problem. In the discussion section we also mentioned important deficiencies in the 3-layer ISBA-A-gs land surface model, including no vertical variability in WG2. These deficiencies inhibit the SEKF performance. We expect the SEKF to perform significantly better with a new multi-layer diffusion based model (ISBA-DIF). For example, with the 3-layer model we assimilate SSM observations into a very shallow layer (0-1 cm), which is very sensitive to the atmospheric forcing over the 24 hour assimilation window. It is possible with ISBA-DIF to assimilate them into a slightly deeper layer (1-5 cm), which is less sensitive to the atmospheric forcing.

**1.8 Minor comments**

1. p. 2, l. 10: I would replace the term network, which usually refers to in-situ measure- ments.

   **Response**: Agreed, we will replace "network" by "coverage".

2. p. 2, l. 10 "a short forecast from the past": it doesnt have to be from the past, it can be a prediction of the current time (i.e. observation time).

**Response**: Agreed, we will replace "a short forecast from the past" with "a short forecast from the previous analysis".

3. p. 4, l. 22: can the authors add a sentence on what the delayed cut-off version of SAFRAN is?

   **Response**: We will add: "The delayed cut-off version of SAFRAN includes additional observations obtained after the real-time cut-off, which makes it more accurate. The delayed cut-off version of SAFRAN uses additional observations from over 3000 climatological observing stations, which report once monthly".

4. p. 5, l. 15: why were only ASCAT observations used and not SMOS for example? Is it because of the study period?

   **Response**: Yes, ASCAT has the advantage of being available over the study period and ASCAT-like data will be available for decades to come. Also, the SEKF at Meteo-France is calibrated to assimilate ASCAT observations and the assimilation has already been performed in a number of studies (Draper *et al.*, 2009; Barbu *et al.*, 2014). The aim of the study was not to test new soil moisture data sources, but to validate the soil water fluxes of the existing system using a hydrogeological model. The assimilation of multiple satellite products is being explored in a different study.

5. p. 5, l. 21: why do the soil water index data need to be interpolated to the model resolution? Can the SEKF not handle different spatial resolutions between the model and the observations?

   **Response**: The SEKF assimilates observations in model space (i.e. the same grid as the model), so it is necessary to perform this interpolation.

6. p. 5, l. 21: p. 5, l. 25: has the WG1 soil moisture climatology been validated?

   **Response**: The ISBA model soil moisture states have been compared with satellite or in situ observations in several studies and generally show a good level of skill (e.g. Draper *et al.* (2009); Muñoz Sabater *et al.* (2007); Albergel *et al.*

(2008, 2010b); Barbu *et al.* (2014); Fairbairn *et al.* (2015)). Most of these studies have also demonstrated that the SEKF can significantly improve the soil moisture scores.

7. p. 5, l. 32: were additional LAI products considered (e.g. MODIS)?

   **Response**: Extensive comparisons of GEOV1 and MODIS are available from the Copernicus GLS website (http://land.copernicus.eu/global/sites/default/files /products/GIOGL1_VR_LAIV1_I1.10.pdf and http://land.copernicus.eu/global/sites/default/files/products/GIOGL1 _QAR_PROBAV-GEOV1_I3.10.pdf). The direct validation based on in situ LAI observations shows that the GEOV1 products present slightly better scores than MODIS.

   In any case, the aim of this study was not to test new observation datasets but to work with the existing system. The SEKF is already set up to assimilate GEOV1 observations (Barbu *et al.*, 2014).

8. p. 7, l. 25-27: this is confusing, how are the 1.2 m2/m2 and +5% values obtained?

   **Response**: Please see 1.3 above.

9. 8, l. 5: how are the LAI and WG1 estimates validated against satellite observations? Werent these satellite observations assimilated into the model?

   **Response**: We agree with the reviewer that this sentence is misleading: "The LAI and WG1 state estimates for the experiments are validated using the satellite observations". We will remove this sentence and replace it with the following: "A system validation was performed by comparing the LAI and WG1 states with the LAI and SSM observations respectively for all the simulations and data assimilation experiments. Note that this is not an independent validation of the performance of the system, for which we would need independent observations. The rationale was to check the effectiveness of the SEKF i.e. to see if it improved the fit between the model simulations and the observations. The fit to the observations was determined by the root mean square difference (RMSD), the correlation coefficient (CC) and the bias. These checks were important because the performance of the SEKF had an important impact on the drainage

and runoff fluxes."

10. p. 9, l. 28-29: I don't understand how the good performance of the NITbc is explained by the relationship between the bias in the discharge ratio and the NSE. Doesnt the NITbc just have a bias-corrected radiative forcing? Where is the causality between the simulation configuration and the performance? Wouldn't it make sense that the model with the smaller bias would have better performance in terms of NSE?

**Response**: We agree this could be clearer. We explained the causality in the original paper in the following paragraph (starting line 32, page 9) by examining the impact of the different simulations on the soil water fluxes. On line 3 of page 10 we mention that: "The NITbc simulation increases the direct radiative forcing by 5%, which results in increased evapotranspiration and lower WG2 during the year. This significantly reduces the drainage and runoff from October to June." We suggest adding another sentence: "The reduced drainage and runoff feeding into the MODCOU hydrogeological model results in less river discharge, which reduces the positive discharge bias. This in turn improves the Nash efficiency scores."

11. p. 12, l. 14-15: but nothing is said on how the higher LAI parameter was chosen.

**Response**: Please see 1.3 above.

Table R2.1: Scores for WG1 (model state compared with observations) averaged over 2007-2014. The closest fit to the observations are shown in bold font.

| Experiment | RMSD $(m^3/m^3)$ | CC | Bias $(m^3/m^3)$ |
|---|---|---|---|
| NIT | 0.051 | 0.77 | 0.00 |
| $NIT_m$ | 0.049 | 0.77 | 0.00 |
| $NIT_{bc}$ | 0.051 | 0.77 | 0.00 |
| LDAS1 | 0.049 | 0.77 | 0.00 |
| LDAS2 | **0.048** | **0.78** | 0.00 |
| $LDAS1_{bc}$ | 0.049 | 0.77 | 0.00 |
| $LDAS2_{bc}$ | 0.049 | **0.78** | 0.00 |
| $LDAS2_{QC}$ | **0.048** | **0.78** | 0.00 |

Table R2.2: Median Nash efficiency (NE) and discharge ratio (Qs/Qo) scores over the 546 river gauges over France and for the subset of 67 gauges with low anthropogenic influence, calculated over 2007-2014. Also shown are the % of stations with a Nash score above 0.6. The best scores are shown in bold font.

| Experiment | NE for 546/67 stations | Discharge ratio for 546/67 stations | % stations with NE > 0.6 for 546/67 stations |
|---|---|---|---|
| NIT | 0.44/0.48 | 1.19/1.16 | 26%/44% |
| $NIT_m$ | 0.48/0.54 | 1.15/1.12 | 30%/48% |
| $NIT_{bc}$ | **0.56/0.60** | **1.02/0.99** | **42%/59%** |
| LDAS1 | 0.44/0.48 | 1.18/1.15 | 27%/44% |
| LDAS2 | 0.41/0.45 | 1.21/1.18 | 23%/40% |
| $LDAS1_{bc}$ | **0.56/0.60** | **1.02/1.00** | **42%/57%** |
| $LDAS2_{bc}$ | 0.53/0.54 | 1.08/1.06 | 38%/53% |
| $LDAS2_{QC}$ | 0.40/0.45 | 1.21/1.18 | 21%/39% |

[Figure]

Figure R2.1: Histogram of the average annual LAI minimum (2007-2014) values for the observations over predominantly grassland points ($> 70\%$ grasslands) ($m^2/m^2$) over France. Also shown is the $NIT_m$ LAI minimum parameter for grasslands.

[Figure]

Figure R2.2: Map showing the average annual LAI minimum (2007-2014) for NIT, $NIT_m$ and the observations ($m^2/m^2$) over France.

[Figure]

Figure R2.3: Scatter plots of the SIM discharge Nash efficiency scores for all 546 stations for (a) NIT vs NITbc and (b) for NIT vs LDAS2. The scores are calculated over 2007-2014.

[Figure]

Figure R2.4: Map of the SIM discharge Nash efficiency scores for the 67 stations with low-anthropogenic influence over France for the NIT simulation, calculated over the period 2007-2014. The river network is also shown.

[Figure]

Figure R2.5: Same as Fig. R2.3, but the stations are classified with either low (67 stations) or high anthropogenic influence (479 stations). For the sake of clarity, the Nash scores are shown between -1.0 and 1.0.

**References**

C. Albergel, J.-C. Calvet, P. de Rosnay, G. Balsamo, W. Wagner, S. Hasenauer, V. Naeimi, E. Martin, E. Bazile, F. Bouyssel, and J.-F. Mahfouf. Cross-evaluation of modelled and remotely sensed surface soil moisture with in situ data in southwestern France. *Hydrol. Earth Syst. Sci*, 14:2177–2191, 2010b.

C. Albergel, C. Rüdiger, T. Pellarin, J.-C. Calvet, N. Fritz, F. Froissard, D. Suquia, A. Petitpa, and B. Piguet. From near-surface to root-zone soil moisture using an exponential filter: an assessment of the method based on in-situ observations and model simulations. *Hydrol. Earth Syst. Sci*, 12:1323–1337, 2008.

D. Aubert, C. Loumagne, and L. Oudin. Sequential assimilation of soil moisture and streamflow data in a conceptual rainfall-runoff model. *J. Hydrol.*, 280:145161, 2003.

A.L. Barbu, J.C. Calvet, J.F. Mahfouf, C. Albergel, and S. Lafont. Assimilation of soil wetness index and leaf area index into the isba-a-gs land surface model: grassland case study. *Biogeosciences*, 8:1971–1986, 2011.

A.L. Barbu, J.C. Calvet, J.F. Mahfouf, and S. Lafont. Integrating ascat surface soil moisture and geov1 leaf area index into the surfex modelling platform: a land data assimilation application over france. *Hydrol. Earth Syst. Sci.*, 18:173192, 2014.

B. Candy, K. Bovis, I. Dharssi, and B. Macpherson. Development of an Extended Kalman Filter for the Land Surface. Technical report, Met Office, Exeter, UK, 2012.

M.L. Carrera, S. Bélair, and B. Bilodeau. The Canadian Land Data Assimilation System (CaLDAS): Description and Synthetic Evaluation Study. *J. Hydrometeorol.*, 16:1293–1294, 2015.

M.P. Clark, D.E. Rupp, R.A. Woods, X. Zheng, R.P. Ibbitt, A.G. Slater, J. Schmidt, and M.J. Uddstrom. Hydrological data assimilation with the ensemble kalman filter: Use of streamflow observations to update states in a distributed hydrological model. *Adv. Water Resour.*, 31:13091324, 2008.

R. A. M. de Jeu, W. Wagner, T. R. H. Holmes, A.J. Dolman, N. C. de Giesen, and J. Friesen. Global soil moisture patterns observed by space borne microwave radiometers and scatterometers. *Surv. Geophys.*, 29:399420, 2008.

P. de Rosnay, Matthias Drusch, Drasko Vasiljevic, Gianpaolo Balsamo, Clment Albergel, and Lars Isaksen. A simplified Extended Kalman Filter for the global operational soil moisture analysis at ECMWF. *Q. J. R. Meteorol. Soc.*, 139:1199–1213, 2013.

B. Decharme, E. Martin, and S. Faroux. Reconciling soil thermal and hydrological lower boundary conditions in land surface models. *J. Geophys. Res. Atmos.*, 118: 78197834, 2013.

C.S. Draper, J.-F. Mahfouf, J.-C. Calvet, E. Martin, and W. Wagner. Assimilation of ASCAT near-surface soil moisture into the SIM hydrological model over France. *Hydrol. Earth Syst. Sci*, 15:3829 – 3841, 2011.

C.S. Draper, J.-F. Mahfouf, and J.P. Walker. An EKF assimilation of AMSR-E soil moisture into the ISBA surface scheme. *Journal of Geophysical Research*, 114:D20104, 2009.

C.S. Draper, R.H. Reichle, R. de Jeu, V. Naeimi, R. Parinussa, and W. Wagner. Estimating root mean square errors in remotely sensed soil moisture over continental scale domains. *Remote Sensing of Environment*, 137:288–298, 2013.

C.S. Draper, R.H. Reichle, G.J.M. De Lannoy, and Q. Liu. Assimilation of passive and active microwave soil moisture retrievals. *Geophys. Res. Lett.*, 39:L04401, 2012.

ECMWF. Annual report 2015. ECMWF website, 2016. URL `http://www.ecmwf.int/sites/default/files/elibrary/2016/16478-annual-report-2015.pdf`. Last accessed July 2016.

D. Fairbairn, A.L. Barbu, J.-C. Calvet J.-F. Mahfouf, and E. Gelati. Comparing the ensemble and extended Kalman filters for in situ soil moisture assimilation with contrasting conditions. *Hydrol. Earth. Syst. Sci.*, 19:4811–4830, 2015.

I. Giuntoli, B. Renard, and M. Lang. *Changes in Flood risk in Europe*, pages 199–211. Kundzewicz, Z. K., Ed., 2012. ISBN 9780415621892. IAHS Special Publication 10.

I. Giuntoli, B. Renard, J.-P. Vidal, and A. Bard. Low flows in france and their relationship to large-scale climate indices. *Journal of Hydrology*, 482:105–118, 2013.

A. Gruber, W. Crow, W. Dorigo, and W. Wagner. The potential of 2D Kalman filtering for soil moisture data assimilation. *Remote Sensing of Environment*, 171:137–148, 2015.

F. Habets, A. Boone, J. L. Champeaux, P. Etchevers, L. Franchistéguy, E. Leblois, E. Ledoux, P. Le Moigne, E. Martin, S. Morel, J. Noilhan, P. Quintana Segu, F. Rousset-Regimbeau, and P. Viennot. The safran-isba-modcou hydrometeorological model applied over france. *J. Geophys. Res.*, 113:D06113, 2008.

H. Hess. Assimilation of screen-level observations by variational soil moisture analysis. *Meteorol. Atmos. Phys.*, 77:145–154, 2001.

P. Le Moigne. Description de l'analyse des champs de surface sur la France par le système SAFRAN. Technical report, Meteo-France/CNRM, 2002.

V. Maggioni, R.H. Reichle, and E.N. Anagnostou. The effect of satellite rainfall error modeling on soil moisture prediction uncertainty. *J. Hydrometeor*, 12:413428, 2011.

J.-F. Mahfouf. Assimilation of satellite-derived soil moisture from ascat in a limited-area nwp model. *Q.J.R. Meteorol. Soc.*, 136:784798, 2010.

J.-F. Mahfouf, K. Bergaoui, C. Draper, C. Bouyssel, F. Taillefer, and L. Taseva. A comparison of two off-line soil analysis schemes for assimilation of screen-level observations. *J. Geophys. Res*, 114:D08105, 2009.

H. Moradkhani, C. M. DeChant, and S. Sorooshian. Evolution of ensemble data assimilation for uncertainty quantification using the particle filter-markov chain monte carlo method. *Water Resour. Res.*, 48:W2520, 2012.

H. Moradkhani, K.-L. Hsu, H. Gupta, and S. Sorooshian. Uncertainty assessment of hydrologic model states and parameters: Sequential data assimilation using the particle filter. *Water Resour. Res.*, 41:W05012, 2005.

D.N. Moriasi, J.G. Arnold, M.W. Van Liew, R.L. Bingner, R.D. Harmel, and T.L. Veith. Model evaluation guidelines for systematic quantification of accuracy in watershed simulations. *American Society of Agricultural and Biological Engineers*, 50:885–900, 2007.

J. Muñoz Sabater, L. Jarlan, J.-C. Calvet, and F. Boyssel. From near-surface to root-zone soil moisture using different assimilation techniques. *J. Hydrometeor*, 8:194–206, 2007.

R.H. Reichle, W.T. Crow, and C.L. Keppenne. An adaptive ensemble Kalman filter for soil moisture data assimilation. *Water Resour. Res*, 44:WO3243, 2008.

R.H. Reichle, J.B. Walker, R.D. Koster, and P.R. Houser. Extended vs Ensemble Kalman Filtering for Land Data Assimilation. *J. Hydrometeor*, 3:728–740, 2002.

D.W. Scott. *Multivariate Density Estimation: Theory, Practice, and Visualization.* John Wiley & Sons, New York, Chicester, 1992.

C. Szczypta, J.-C. Calvet, C. Albergel, G. Balsamo, S. Boussetta, D. Carrer, S. Lafont, and C. Meurey. Verification of the new ECMWF ERA-Interim reanalysis over France. *Hydrol. Earth Syst. Sci.*, 15:647–666, 2011.

G. Thirel, E. Martin, J.-F. Mahfouf, S. Massart, and F. Habets. A past discharges assimilation system for ensemble streamflow forecasts over france part 1: Description and validation of the assimilation system. *Hydrol. Earth Syst. Sci.*, 14:1623–1637, 2010.

Y. Zhou, D. McLaughlin, and D. Entekhabi. Assessing the Performance of the Ensemble Kalman Filter for Land Surface Data Assimilation. *Mon. Wea. Rev.*, 134:21282142, 2006.

---

## Author Comment (AC3) · 26 Jul 2016

Fairbairn *et. al.*, (2016):

http://www.hydrol-earth-syst-sci-discuss.net/hess-2016-195/

Response to comments from Dr Jean-Philippe Vidal

July 26, 2016

Firstly, we would like to thank Dr Jean-Philippe Vidal for his comments and useful suggestions.

**1 Major comment**

Dr Jean-Philippe Vidal was right to point out that many of the stations included in the calculations are influenced by anthropogenic water management, which is not simulated by the MODCOU hydrogeological model. He was concerned that the results might be interpreted as being closer to anthropogenically influenced streamflow. He suggested the following:

1. To consider only catchments with low anthropogenic influence in order not to compare apples and oranges and avoid drawing conclusions on the ability of SIM (with or without data assimilation) to simulate anthropogenically influenced streamflow,

2. To show scatter plots of NSEs instead of distributions (possibly with marginal distributions) to reduce the potential spatial bias effect mentioned above.

**Response:**

We have decided to follow Dr Jean-Phillipe Vidal's suggestions by showing the results for the stations with low-anthropogenic influence. We have used the suggested reference networks of Giuntoli *et al.* (2012, 2013) to extract a subset of 67 river gauges with low-anthropogenic influence from the original 546 stations, valid for both low and high flows. A map of these stations is shown in Fig. R3.1 (at the end of the document). A scatter plot is shown of the Nash efficiency of these stations (labeled as "Low anth. influence") and all the other stations (labeled as "High anth. influence") in Fig. R3.2.

For the sake of clarity, in Fig. R3.2 only the stations are shown in the range of Nash scores -1.0 to 1.0. The 'low anth. influence' stations follow a similar pattern to the 'high anth. influence' stations. Furthermore, we calculated the Median Nash efficiency scores for the 67 stations in Table R3.1 (at the end of the document). Note that the median is calculated rather than the mean because the majority of stations ($> 80\%$) have positive Nash efficiency scores, but a few outliers have scores near to -100. The median is a more appropriate metric as it is less sensitive to extreme outliers and is a better indicator for highly skewed distributions (Moriasi *et al.*, 2007). The scores for this subset are improved relative to the 546 stations in Table R3.1, as expected. In particular, the percentage of stations with good scores (Nash efficiency $> 0.6$) is increased significantly. The discharge bias is also slightly less for the stations with low anthropogenic influence relative to the 546 stations. This supports Jean-Phillipe Vidal's suggestion that part of the positive bias in the discharge ratio of the NIT simulation for the 546 stations could be attributed to abstractions not being accounted for. However, the majority of the discharge bias in the NIT simulation is still present with the 67 stations with low anthropogenic influence. Moreover, the relative performance of the experiments is very similar. Therefore, the conclusions of the experiments are not affected by the ability of SIM (with or without data assimilation) to simulate anthropogenically influenced streamflow.

We will explain these results in Section 3.4 of the paper. Figures R3.1 and R3.2 will be included in a supplement.

**2   Minor comment**

Their interpretation of the NSE detailed P8L8-9 is incorrect:a negative NSE value means that the model performs worse than a constant model with a value equal to the average of all observations.

**Response:** We agree. We will replace "The Nash efficiency can range from $-\infty$ to 1, with 1 corresponding to a perfect match of the model to the observed data and scores less than zero implying that the model mean is a worse predictor than the observations." with "The Nash efficiency can range from $-\infty$ to 1, with 1 corresponding to a perfect match of the model to the observed data and a negative value implying that the model performs worse than a constant model with a value equal to the average of all the observations."

[Figure]

Figure R3.1: Map of the SIM discharge Nash efficiency scores for the 67 stations with low-anthropogenic influence over France for the NIT simulation, calculated over the period 2007-2014. The river network is also shown.

[Figure]

Figure R3.2: Scatter plots of the SIM discharge Nash efficiency scores for the 546 stations over France for (a) NIT vs NITbc and (b) for NIT vs LDAS2. The stations are classified with either low (67 stations) or high anthropogenic influence (479 stations). For the sake of clarity, the Nash scores are shown between -1.0 and 1.0. The scores are calculated over 2007-2014.

Table R3.1: Median Nash efficiency (NE) and discharge ratio (Qs/Qo) scores over the 546 river gauges over France and for the subset of 67 gauges with low anthropogenic influence, calculated over 2007-2014. Also shown are the % of stations with a Nash score above 0.6. The best scores are shown in bold font.

| Experiment | NE for 546/67 stations | Discharge ratio for 546/67 stations | % stations with NE > 0.6 for 546/67 stations |
|---|---|---|---|
| NIT | 0.44/0.48 | 1.19/1.16 | 26%/44% |
| $NIT_m$ | 0.48/0.54 | 1.15/1.12 | 30%/48% |
| $NIT_{bc}$ | **0.56/0.60** | **1.02/0.99** | **42%/59%** |
| LDAS1 | 0.44/0.48 | 1.18/1.15 | 27%/44% |
| LDAS2 | 0.41/0.45 | 1.21/1.18 | 23%/40% |
| $LDAS1_{bc}$ | **0.56/0.60** | **1.02/1.00** | **42%/57%** |
| $LDAS2_{bc}$ | 0.53/0.54 | 1.08/1.06 | 38%/53% |
| $LDAS2_{QC}$ | 0.40/0.45 | 1.21/1.18 | 21%/39% |

**References**

I. Giuntoli, B. Renard, and M. Lang. *Changes in Flood risk in Europe*, pages 199–211. Kundzewicz, Z. K., Ed., 2012. ISBN 9780415621892. IAHS Special Publication 10.

I. Giuntoli, B. Renard, J.-P. Vidal, and A. Bard. Low flows in france and their relationship to large-scale climate indices. *Journal of Hydrology*, 482:105–118, 2013.

D.N. Moriasi, J.G. Arnold, M.W. Van Liew, R.L. Bingner, R.D. Harmel, and T.L. Veith. Model evaluation guidelines for systematic quantification of accuracy in watershed simulations. *American Society of Agricultural and Biological Engineers*, 50:885–900, 2007.

---

## Author Response (AR2)

**hess-2016-195**

**"Integrated validation of assimilating satellite derived observations
over France using a hydrological model"
by D. Fairbairn et al.**

New title: "*The effect of satellite-derived surface soil moisture and leaf area index land data assimilation on streamflow simulations over France*"

1 March 2017.

Dear Professor Wolfgang Wagner,

The authors' response to the comments of the two anonymous referees have been accounted for in the revised version of the paper.

All changes relative to the previous version of the paper are detailed in the pdf of the new manuscript. They include all the response elements given by the authors in response to the reviewers' comments: blue and red for Reviewers 1 and 2 respectively (other changes are in green).

The Discussion and Conclusion sections were re-written. Title was changed as suggested by Reviewer 1.

Five new Figures were included in the Supplement in order to address issues mentioned by Reviewer 1.

Yours sincerely,

JC Calvet, D. Fairbairn.

**Response to comments from Referee 1**

**March 2, 2017**

We would like to thank the reviewer for their constructive comments. The corrections in the revised manuscript for Referee 1 are marked in blue.

**Response to major comments:**

**1**

**1.1**

**Referee comment** - *The introduction sounds like a twisted excuse to not follow recent advances in land surface data assimilation and to get away with a suboptimal system. Please acknowledge the true state-of-the-art: a) P.2, L29: one (?) study found no advantage of 2D Kalman filtering over 1D Kalman filtering. Maybe. Yet, very many other studies use 2D/3D Kalman filtering and that is the only correct way of doing Kalman filtering if we deal with different spatial resolutions. b) The SEKF may be a preferred method at some operational centers, but in most other centers, there is a push towards the EnKF.*

**Response:**

The introduction was changed in response to the previous round of reviews in order to justify our use of the SEKF. As of yet we have found no evidence that the 1D EnKF performs better than the SEKF (in terms of root-zone soil moisture) for our LSM, although we have only tested the EnKF on a dozen sites (Fairbairn et al., 2015). But the reviewer is right that we should not have implied that the SEKF is superior to the EnKF, so we have given a more balanced discussion in the introduction of the revised paper. On page 3, line 2 of the revised manuscript: "The SEKF simplifies the EKF by using fixed and uncorrelated background errors at the start of each cycle. Importantly, the SEKF generates flow-dependence and implicit background-error covariances from additional model integrations in the observation operator Jacobian calculations. Draper et al. (2009) found the flow-dependence from a 24-hour assimilation window

was sufficient to enable the SEKF to perform similarly to an EKF (which cycles the background-error covariance). Likewise, Muñoz Sabater et al. (2007); Fairbairn et al. (2015) found that the SEKF and EnKF performed similarly, in spite of different linear assumptions."

In future research, the EnKF could still be attractive as it can be designed to account for model/forcing errors and 2D background-error covariances. However, as already discussed in our paper and evidenced in the literature (e.g. Maggioni et al. (2012); Gruber et al. (2015); Fairbairn et al. (2015)), the EnKF has its own set of challenges to overcome including its own linear assumptions. Therefore we must also be cautious about recommending it. In the revised manuscript we have mentioned in the discussion and in the conclusion that we need to test the EnKF over France in the context of the SIM hydrological model. On page 17, line 28: "Fairbairn et al. (2015) found that an EnKF with a simple stochastic rainfall error estimation demonstrated similar WG2 scores to the SEKF over 12 sites in southwest France (validated using in situ observations). Both methods were also affected by nonlinearity problems. We intend to test an EnKF over France using a similar validation employed in this study."

**1.2**

**Referee comment** - *Why did the authors continue to use the SEKF without (or w/ minimal) alterations and then run into the same problems as already reported in Draper et al. (2011)? One of the conclusions is that an EnKF and assimilation in a slightly deeper surface soil layer may alleviate the problems that are experienced in this paper: trying out these recommendations would be food for a paper, but rerunning the same problems, is not so nice.*

**Response:**

We use the SEKF because it is the most mature technique developed for land surface data assimilation within SURFEX. Our intention was to validate this system at a large scale by using independent streamflow observations.

The paper by Draper et al. (2011) partly motivated our experiments, but there would be no point trying to recreate their results. The novelty of our work consists of 1) assimilation of LAI and demonstration of its impact on the soil moisture fluxes and 2) validation of our experiments using streamflow observations. We also examine some issues regarding the model (and the resulting influence on the fluxes), namely the underestimated LAI minimum and a systematic overestimation of the radiative forcing. This is made clear now in the introduction (pages 3 and 4).

**1.3**

**Referee comment** *Would the EnKF really help, as suggested in the abstract? The introduction says that the EnKF and SEKF produce results with similar accuracies. (p.3 L13).*

**Response:**

Please see response to Question 1.1.

**1.4**

**Referee comment** *The model is run at 8 km, the ASCAT data are at 25 km, the LAI data are at 1 km resolution. Data assimilation should take care of these spatial discrepancies, especially to downscale the coarser data to the finer resolution. A priori interpolation just does not make sense: this adds unnecessary errors and the subsequent 1D assimilation wrongly assumes that spatially independent observations are assimilated, while in reality there is a perfect oversampling with perfect spatial error correlations. The latter is especially a problem when (p.7, L26) the SEKF analysis is calculated independently for each patch, with the same obs used for all patches in de grid-box.*

**Response:**

We agree with the reviewer that interpolation errors would be introduced regardless of the approach we employ, even if we interpolated the model gridpoints to the observation grid in the SEKF analysis. Please note that one line 5 of page 6 of the revised manuscript we have added the following sentence: "After screening, the data were projected onto the 8 km resolution model grid by averaging all the data within 0.15 degrees of each gridpoint (Barbu et al., 2014)." Then it is considered that there is not a perfect oversampling because each gridpoint uses a different set of observations, albeit with some overlap.

Also the sentence: SEKF analysis is calculated independently for each patch, with the same obs used for all patches in de grid-box. (line 14, page 8) was changed to: The SEKF analysis is calculated independently for each patch using the Jacobians for each individual patch but with one mean observation per grid box.

**1.5**

**Referee comment** *If the spatial errors are discarded, then the observation error variance should at least be increased which is done, but far too little to make any difference (i.e. from 0.050 to 0.055 m3/m3). In addition, the observation error should be adjusted in line with the rescaling and be spatially variable if the same obs is used for different vegetation classes. Yet, there is no linkage between obs errors and vegetation class in*

*this paper.*

**Response:**

We agree that the observation error variance should be increased in our case from 0.050 to 0.055 m3/m3. This reduces the WG2 analysis increments by about 10%. This has been clarified on line 21, page 8 of the revised manuscript.

One has to be aware that the vegetation class concept implies already the definition of a mean land use type. The LAI observations have an original resolution of 1 km and have been aggregated to the grid cell resolution (8 km). However, at 1km resolution, there is still a high degree of landscape heterogeneity over France. The analysis is adapted to plant functional types via the patch fractions and via the Jacobians.

**1.6   Specific comments**

1. **Comment:** *P.6, L28: How is it possible to assume zero (error, I assume) covariances between LAI and WG2, and at the same time derive meaningful Jacobians that calculate e.g. dLAI/dWG2. Isnt this a basic contradiction? Please explain.*
   **Response**: We explain this in the revised manuscript (page 7, lines 6-10): "The SEKF simplifies the EKF by using fixed and uncorrelated background errors at the start of each cycle. Implicit background-error covariances between the layers and the prognostic variables are generated at the analysis time by the model integration in the observation operator Jacobians."

2. *Why calculate dLAI/dLAI by adding perturbations? Should this not be simply =1? Why not?*
   **Response**: We would like to make clear this statement.

   The Jacobians of the observation operator are defined on page 7 of the revised manuscript (equations 4 and 5). The perturbation is applied at the start of the window, while the finite difference from the model integrations in the Jacobian calculation is considered at the end of the window (the observation time). Therefore it depends on the model dynamics. The seasonal variability of the Jacobians was clearly demonstrated by Rüdiger et al. (2010).

3. **Comment:** *P.4, L26: Why is soil moisture rescaled, whereas LAI is not? The KF assumes unbiased innovations in either case: does this work out in the end?*
   **Response**: Yes, we believe that it is necessary to rescale SSM observations to match the SSM model climatology because small-scale discrepancies in soil textural properties can cause very large systematic differences between the observations and the model. (Page 6, line 12 of the revised manuscript).

   In order to explain our approach we have added the following sentences in the revised manuscript (page 6, lines 27-35, page 7, lines 1-5).

"When considering removing systematic differences between the model and the observations, a linear rescaling of the LAI observations to the model climatology would be problematic because the model-observation bias is linked to model deficiencies. When considering removing systematic differences between the model and the observations, a linear rescaling of the LAI observations to the model climatology would be problematic because the model-observation bias is linked to model deficiencies. For SSM, systematic errors are related to the mis- specification of physiographic parameters, such as the wilting point and the field capacity. As mentioned by several authors (e.g. Koster et al. (2009); Albergel et al. (2012)), the information content of soil moisture does not necessarily rely on its absolute magnitude but instead on its time variations. For SSM, the systematic bias between the model and the data consists mainly in their magnitude rather than their seasonal variability. Therefore this justifies the common approach used in land surface data assimilation studies for the SSM variable. Contrary to SSM, the LAI bias between the model and the data has two components: one in magnitude and the other one in timing (see e.g. Figure 6 in Barbu et al. (2014)). When compared with the satellite data, the LAI model dynamics clearly shows a shift in the seasonal cycle, mainly caused by model errors. The remote sensing LAI measurements potentially encapsulate realistic environmental features that are not or incorrectly represented by the model. Forcing the data to conform to the model climatology would result in a loss of relevant information. Therefore, in this context, a rescaling of the LAI data to the model climatology was not considered. Furthermore, Barbu et al. (2014) found that the assimilation without rescaling can cope with these model errors."

On the other hand, systematic differences between the model and the observations can be removed by modifying the model parameters (Kumar et al., 2012), which was the motivation for correcting the LAI minimum in our study (Page 9, lines 14-16 of the revised manuscript). This substantially reduced the RMSD between the model and the observations. We repeated the cluster of experiments (NIT, LDAS1 and LDAS2) before and after correcting the systematic errors in the LAI minimum and the short-wave radiative forcing and found similar relative performances. We explained the results by analyzing the observation operator Jacobians (Section 3.3), which are not related to biases in the model.

4. **Comment:**P.6, L30: why is WG1 not part of the state vector?

   **Response**: We have mentioned in the revised manuscript (page 7 line 13-14): "The WG1 layer is not included in the analysis update because it is shallow layer (1 cm depth) that is driven by the atmospheric forcing rather than the initial conditions (Draper et al., 2009; Barbu et al., 2014)". In practice it has little

bearing on the soil moisture fluxes.

5. *P.7, L2: what do you mean by a 24-hour assimilation window? A filter is used, not a smoother. Data are assimilated every 3 days for soil moisture and every 10 days for LAI.*
   **Response**: We would like to state this clearly. The model integrations used in the observation operator Jacobians operate over a 24-hour period, otherwise the assimilation window length would be irrelevant. This is explained on page 7, line 28 of the revised manuscript.

6. Comment: *In short, the paper needs a thorough and in depth acknowledgement of all the violated assumptions in the KF: no spatial error correlations where perfect error correlations are present (and consequently, an exaggerated impact of the DA); either a relationship or none between LAI and WG2 in the setup of the background errors and the Jacobians, the choice for the observation error variance, etc. I would suggest to validate the optimality of the data assimilation system (e.g. white innovations in space and time?) and hopefully it comes out just fine: in that case, the paper could perhaps be considered for publication, otherwise it becomes hard to justify.*
   **Response**: There is evidence in the literature that the SEKF does work effectively. The flow-dependence generated by the model integration in the observation operator Jacobians allows the SEKF to spread information between layers (SSM and WG2) and prognostic variables (WG2 and LAI). This flow-dependence is limited to the 24 hour assimilation window, but this is sufficient for the SEKF to perform as well as the EKF (Draper et al., 2009; Muñoz Sabater et al., 2007) and similarly to an EnKF (Fairbairn et al., 2015). This is explained in the introduction of the revised manuscript (page 3, lines 2-7).

   As suggested by the reviewer, several figures are added to the supplement which illustrates the temporal LAI and SSM innovation evolutions (Figures S1.7 and Figure S1.8), as well as the LAI and SSM innovation distributions together with their respective Gaussian fit curves (Figure S1.4, S1.5 and S1.6). "These differences are illustrated in terms of probability distribution in Fig.S1.4 which shows the innovation histogram and the Gaussian fitting curve of the SSM product before rescaling" (page 6, lines 14-15).

   In addition, in the Supplement, Fig.S15 and Fig.S1.6 show the histograms of the innovations (difference between the model-predicted observations and the data) and residual (difference between the analysis and the data). The pdf for SSM agrees very well with Kalman theory, since it closely fits the Gaussian distribution. The pdf for the LAI is not far away from its normal fit. The LAI innovations

present a left tailed distribution. As expected, the standard deviation of residuals is reduced compared to those of innovations. For an optimal filter the innovation time series should be uncorrelated in time. For both SSM and LAI the temporal evolutions of innovations are illustrated in Fig. S1.7 and in Fig. S1.8 of the Supplement, respectively.

7. Comment: *The title is not very representative: Integrated validation sets high expectations, we do expect advanced validation methods, or at least more than just only streamflow (i.e. maybe include turbulent fluxes, groundwater, in situ soil moisture,). I would rephrase it as something like The effect of [assimilation] on streamflow estimates. Secondly, the authors responded to one of the reviewers that he/she was inaccurate about referring to assimilation in a hydrological model yet, that is exactly what the title says.*
   **Response**: We agree that this was confusing. The title "integrated validation" was meaning that we integrate a land surface model and a river routing model to perform the validation. As suggested by the reviewer the title is changed now into "The effect of satellite derived surface soil moisture and leaf area index assimilation on streamflow simulations over France"

8. Comment: *P.2, L10: observation network –¿ observation coverage (already mentioned in earlier review, and I agree that this needs to change)*
   **Response**: Sorry, we should have already corrected this. It is corrected in the revised manuscript.

9. Comment: *P.2, L9: instruments are subject to retrieval errors. –¿ The \*data\* are subject to retrieval errors. And besides: we could assimilate raw radiance or backscatter observations and circumvent retrieval errors (but errors would be elsewhere)*
   **Response**: It is changed in the revised manuscript.

10. Comment: *p.4, L2: integrated validation of the soil moisture fluxes. What is a soil moisture flux? Did you mean groundwater recharge, runoff, river discharge, ?*
    **Response**: It is replaced with " integrated validation of the drainage and runoff fluxes.

11. Comment: *p.9, L13 : SIM is not a tool to validate. SIM provides model simulations which can be validated using in situ observations, and using some specific validation metrics.*
    **Response**: It is replaced with: "The SIM hydrological model was used to validate

the drainage and runoff from ISBA-A-gs by comparing the simulated streamflow from MODCOU with observations".

12. Comment: *p.4, L17: Draper et al. (2011) already assimilated SSM into SIM, but they did not perform the validation against streamflow, which is the whole goal of the current paper. To be complete, this paper should include a validation of these older results (Draper et al., 2011) against streamflow.*
    **Response**: As already mentioned in the response to question 1.2 and already stated in the introduction of the revised manuscript, there is no point trying to recreate the results of the paper by (Draper et al., 2011). The novelty of our work consists of 1) assimilation of LAI and demonstration of its impact on the soil moisture fluxes and on the streamflow and 2) joint assimilation of LAI and SSM and demonstration of its impact on the soil moisture fluxes and on the streamflow.

    In addition, it is difficult to reproduce Draper's results because several modifications and improvements concerning the model and the data have been done in the meantime.

13. Comment: *P.12, section 3.2: Are all the WG1 and LAI metrics simple internal checks? I.e. comparison against data that are assimilated into the model? Please repeat that here. It would make much more sense to validate against independent soil moisture observations. What is the statistical significance level when it is stated that LDAS1 significantly improves the fit*
    **Response**: Yes, they are internal checks, as stated in Section 2.5.1 on page 10 of the revised manuscript. We would prefer to validate WG2 rather than WG1, but WG2 cannot be validated using satellite observations. In situ measurements of WG2 are scarce and, in addition, point measurements are not necessarily representative of a coarser pixel scale, and thus are difficult to interpret when compared to model results. This is one of the reasons we are validating the drainage and runoff fluxes rather than soil moisture.

    We have re-phrased "LDAS1 significantly improves the fit" to "LDAS1 substantially improves the fit".

14. Comment: *What is the model integration time step? Since the assimilation is done at 9:00 UTC, whereas the data are taken at 9:30 UTC, I assume that there must be a very long model time step (one hour or more?). Could the long model integration time step be another cause of inferior model performance at some times?*
    **Response**: We added the following clarification to the revised manuscript.

The model integration is performed every 15 minutes. However, the atmospheric forcing is assumed constant over hourly intervals for instantaneous measurements such as precipitation. Therefore any discrepancies in SSM are small (page 6, line 20 of the revised manuscript).

**Response to comments from Referee 2**

March 2, 2017

We would like to thank the reviewers for their constructive comments. The corrections in the revised manuscript for Referee 2 are marked in red.

**Response to major comments:**

**1**

**1.1**

**Referee comment** *I understand how correcting the minimum LAI for an important land cover type (grassland) in the model has the desired effect of reducing streamflow overestimations. However, what is the scientific merit of replacing the minimum LAI simulation by the minimum of satellite observations? Firstly, it would be more justifiable to address these LAI shortcomings in the modeling itself, rather than replacing the values that are undesired. Alternatively, external observations can be used to correct the model LAI, but in that case, I believe it should not be restricted to only the LAI minimum for grassland. Only correcting this feature and not any other features is quite arbitrarily. Looking at Figure 3, it seems that, besides underestimating observations over grassland in winter, the model LAI is significantly larger than the observations over deciduous forest (almost the entire year) and over C3 crops in summer and fall. The latter discrepancies seem to be equally important (if not more) as the underestimation over grassland. Furthermore, looking at Table 3, correcting the grassland minimum LAI even significantly increases the average bias against the LAI observations, which feels like a counter-intuitive approach.*

*Therefore, I recommend the authors to perform an additional experiment, with complete rescaling of the model LIA to the climatology of the LAI observations, and to re-evaluate the simulations of discharge, or to provide extensive scientific proof (e.g. based on other LAI products like MODIS) that the winter grasslands are really the main or only concern.*

**Response:**

In the ISBA-A-gs model the LAI minimum is a model parameter. Using satellite data to determine the value of this parameter is relevant. The reviewer is right that simply increasing the LAI minimum parameter is not a good long term solution. It would be preferable to tackle the deficiencies in the model directly, which seem to be linked to the response of photosynthesis to temperature rather than the parameter itself, but this is a modelling problem that is beyond the scope of this paper. We have acknowledged this in the abstract/discussion/conclusions.

A thorough comparison of the ISBA-A-gs simulated LAI with both SPOT-VGT (used in our experiments) and MODIS data over south-west France was performed by Brut et al. (2009). They did notice significant discrepancies between all three data sets, suggesting that there is significant uncertainty in both the model and the observations. However, they also noticed that the modelled LAI of the C3 natural herbaceous (grasslands)/C3 crops had a delayed onset relative to both satellite products (see Figure 4 in Brut et al. (2009)). They found that this was particularly problematic for grasslands in mountainous regions. By comparing the data with in situ measurements, they found that the generic temperature response of photosynthesis used in the model is not appropriate for plants adapted to the cold climatic conditions of the mountainous areas. This problem was also linked to the reduced and prolonged LAI minimum in the model relative to the observations. Lafont et al. (2012) found similar issues when comparing the same products over France. These problems would explain the delayed onset and underestimated LAI minimum for both grasslands and C3 crops in Figure 3 in our study. Indeed, Figure 4 in our paper shows that the NIT LAI minimum was particularly underestimated in the grassland areas of the Massif Central mountains in central France, but not so much in lower regions further north. We have mentioned this in the discussion (page 16, lines 3-23).

Evidently from Figure 3 there are significant discrepancies between the model and the observations for C3 crops and deciduous forests as well as grasslands. We focused on grasslands partly because it represents the most common vegetation type over France (32%) and partly because other authors have discovered similar issues for grasslands (Brut et al., 2009; Lafont et al., 2012; Barbu et al., 2014). We have acknowledged that research is needed to improve the modelled values for the other vegetation types in the discussion. One way would be to assimilate observations at the patch scale (Response to 1.3.3 gives details).

We found that it was necessary to rescale SSM observations to match the SSM model climatology, partly because differences in the representation of the soil texture can cause very large systematic differences between the observations and the model (Page 6, line 11 of the revised manuscript). But the current ASCAT product is affected by vegetation effects (Vreugdenhill et al., 2016) and a seasonal CDF matching is needed in DA

systems assimilating ASCAT SSM. But this procedure is still sub-optimal. A solution to this problem is to go towards the implementation of an observation operator in order to assimilate the backscattering coefficients directly. In this way, the vegetation information content in the ASCAT signal could be used to analyse vegetation biomass and would also provide information for the analysis of root-zone soil moisture, in addition to the microwave soil moisture signal (page 17, lines 20-24).

A linear rescaling of the LAI observations to the model climatology would be problematic because the model-observation bias is linked to known deficiencies in the model, rather than due to specifications of the SSM physiographic parameters, such as the wilting point and the field capacity. As mentioned by several authors (e.g. Koster et al. (2009); Albergel et al. (2012)), the information content of soil moisture does not necessarily rely on its absolute magnitude but instead on its time variation. For SSM the systematic bias between the model and the data consists mainly in their magnitude rather than their seasonal variability. Therefore this common approach used in land surface data assimilation studies for the SSM variable is justified. Contrary to SSM, the LAI bias between the model and the data has two components: one in magnitude and the other one in timing. When compared with the satellite data, the LAI model dynamics clearly shows a shift in the seasonal cycle, mainly caused by model errors. The remote sensing LAI measurements potentially encapsulate realistic environmental features that are not or incorrectly represented by the model. Forcing the data to conform to the model climatology would result in a loss of important information. Therefore, in this context, a rescaling of the LAI data to the model climatology was considered inappropriate. Furthermore, Barbu et al. (2014) found that the assimilation mechanism employed without rescaling can cope with these model errors. This is now explained on page 6, lines 27-35 of the revised manuscript. Also, it does not seem consistent to us to rescale the modelled LAI to match the observations while the opposite is done for SSM (the observations are rescaled to match the WG1 climatology).

Systematic differences between the model and the observations can be removed by modifying the model parameters (Kumar et al., 2012), which was the motivation for correcting the LAI minimum in our study (Page 9, lines 14-16 of the revised manuscript). This substantially reduced the RMSD between the model and the observations. We repeated the cluster of experiments (NIT, LDAS1 and LDAS2) before and after correcting the grassland LAI minimum and the radiative forcing in the LAI minimum and the short-wave radiative forcing and found similar relative performances. We explained the results by analyzing the observation operator Jacobians (Section 3.3), which are not related to biases in the model.

**1.2**

**Referee comment** *I have some concerns regarding the choice of not correcting biases in LAI prior to the assimilation. I can see that the purpose of this experiment is to verify whether the assimilation can mediate the bias in LAI. Although being appealing, data assimilation is theoretically not designed to correct model bias. Again, as mentioned in the previous comment, such bias should actually be resolved in the model itself, or by rescaling the observations. I believe the same principle applies for LAI as for soil moisture, for which biases were appropriately removed in this study. The assimilation of biased LAI may not have had the effected that was hoped for, because of the small Jacobians in winter, but it could for instance also have disturbed the model behavior in summer, when larger Jacobians as well as large biases in LAI over crops were present. I believe this is an important issue that is not addressed in the paper. Moreover, the assimilation of (biased) LAI may also impact the climatology of the soil moisture simulations, which then again becomes biased with respect to the (previously bias-corrected) ASCAT retrievals. This could potentially be another important reason why the soil moisture assimilation, in combination with LAI assimilation, wets the lower layer and fails to improve discharge simulations.*

*I suggest the authors to include an assimilation run for which the bias in LAI between model and observations is removed a priori, e.g. by rescaling the model LAI to the observations, cfr. comment 1 above. The impact of the summer bias over cropland (in combination with large Jacobians), and of deciduous forest bias need to be better addressed. Also, the impact of LAI assimilation on soil moisture bias between simulations and ASCAT retrievals requires further investigation. I believe the bias-correction of the soil moisture observations should best be done with respect to the LAI assimilation experiment.*

**Response:**

We agree with the reviewer that DA methods are not theoretically designed to correct systematic model errors. A bias in the forecast model invalidates the assumption of bias-blind data assimilation (Dee, 2005). We admit that the DA experiments should not be motivated by correcting systematic model errors but instead DA plays an important role in correcting random errors in the initial conditions. We have now made this clear throughout the revised paper, including the abstract, introduction and conclusion. Note that this assumption does not affect the conclusions of the experiments because we repeated the cluster of experiments (NIT, LDAS1 and LDAS2) before and after correcting the systematic errors in the LAI minimum and the radiative forcing and found similar relative performances. Moreover, we explained the results in terms of the observation operator Jacobians.

We would not recommend rescaling the observed or modelled LAI (please see Section 1.3.1).

The reviewer is right that the assimilation of LAI causes a small net negative bias in LAI, which is evident in Table 2 in the paper. The bias is similar for LDAS1 and LDAS2 because it is caused by the $\frac{\partial LAI}{\partial LAI}$ Jacobian, rather than the $\frac{\partial LAI}{\partial WG2}$ Jacobian. Given that there is no significant wettening of WG2 in LDAS1, it is not possible for this bias to be causing the wettening of the WG2 layer and the resulting increase in drainage/runoff for LDAS2. The $\frac{\partial LAI}{\partial WG2}$ tends to be positive and is largest during summer/autumn (Barbu et al., 2014). There is no evidence that it leads to long term increases in WG2 or results in increased drainage/runoff for LDAS2 in winter/spring. We have mentioned this in Section 3.3 of the revised manuscript (page 14, lines 5-6). Moreover, Draper et al. (2011) assimilated only SSM observations and discovered similar issues to the LDAS2 experiment in our paper.

**1.3 Specific comments**

**1.3.1**

**Referee comment** *P2.L30-P3.L22: After a comment from the previous review, parts of the introduction have been rewritten to more strongly motivate the use of an SEFK relative to an ENKF. However, I feel like the text is rewritten as to actually recommend the SEFK over the ENKF. I dont believe there is really a need to defend the use of the SEFK so strongly, almost in a confrontational way with respect to the ENKF. Each has its own advantages. I would suggest to rewrite some of the phrases, making it less of a confrontation between both methods. It also feels a bit strange that the authors commend the SEKF in the introduction, whereas in the conclusion, they recommend future use of an ENKF.*

*Similarly, the authors mention the study by Gruber et al. (2015) in P2.L29 to defend the choice for a 1D Kalman filter. I dont have any problem with using a 1D filter and mentioning this paper as a support, but suggest to make a more cautious statement. Theoretically one would expect a better performance with 2D systems in case of coarser-resolution observations, as they decrease the representativeness error between model forecasts and observations. As it stands, it feels like a general statement that advises the use of a 1D filter, with which I do not agree.*

**Response:**
Indeed, the introduction was changed in response to the previous round of reviews in order to justify our use of the SEKF. As of yet we have found no evidence that the 1D EnKF performs better than the SEKF (in terms of root-zone soil moisture) for our

LSM, although we have only tested the EnKF on a dozen sites. But the reviewer is right that we should not have implied that the SEKF is superior to the EnKF, so we have given a more balanced discussion in the introduction of the revised paper. On page 3, line 2 of the revised manuscript: "The SEKF simplifies the EKF by using fixed and uncorrelated background errors at the start of each cycle. Importantly, the SEKF generates flow-dependence and implicit background-error covariances from additional model integrations in the observation operator Jacobian calculations. Draper et al. (2009) found the flow-dependence from a 24-hour assimilation window was sufficient to enable the SEKF to perform similarly to an EKF (which cycles the background-error covariance). Likewise, Muñoz Sabater et al. (2007); Fairbairn et al. (2015) found that the SEKF and EnKF performed similarly, in spite of different linear assumptions."

In future research, the EnKF could still be attractive as it can be designed to account for model/forcing errors and 2D background-error covariances. However, as already discussed in our paper and evidenced in the literature (e.g. Maggioni et al. (2012); Gruber et al. (2015); Fairbairn et al. (2015)), the EnKF has its own set of challenges to overcome including its own linear assumptions. Therefore we must also be cautious about recommending it. In the revised manuscript we have mentioned in the discussion that we need to test the EnKF over France in the context of the SIM hydrological model. On page 16, line 19: "Fairbairn et al. (2015) found that an EnKF with a simple stochastic rainfall error estimation demonstrated similar WG2 scores to the SEKF over 12 sites in southwest France (validated using in situ observations). Both methods were affected by nonlinearity problems. We intend to test an EnKF over France using a similar validation employed in this study."

**1.3.2**

**Referee comment** *P6.L9-11: The processing of the ASCAT data needs a little bit more explanation. How are data interpolated to the 8-km grid? Which temperature threshold was used for filtering frozen areas (zero Celsius or larger)? Have open water fractions or snow been dealt with? Why is altitude used instead of the topographic slope, or the topographic complexity flag in the ASCAT product? A plateau at high altitude seems more preferable to me than a terrain with strong slopes at low elevation, considering the backscattering mechanisms at hand. Adjusting these processing issues could be a way forward for getting improved results in future studies.*

**Response:**
In order to better explain the ASCAT data processing, we have added the following to Section 2.2, pages 6: "A surface-state flag is provided with the ASCAT product, which identifies frozen conditions, the presence of snow cover or temporary melting/water

on the surface. Observations are screened during frozen surface conditions or when snow-cover is present if the ASCAT flag is set to frozen. Additionally, observations with a topographic complexity flag greater than 15% and/or a wetland fraction greater than 5% (both provided with the ASCAT data) are removed. More information about ASCAT quality flags can be found in (Scipal et al., 2005).

After screening, the data were projected onto the 8 km resolution model grid by averaging all the data within 0.15 degrees of each gridpoint (Barbu et al., 2014). As in Draper et al. (2011) an additional screening step was performed to remove observations whenever frozen conditions were detected in the model using a threshold temperature of zero Celsius. In addition, observations with an altitude greater than 1500 m and with an urban fraction greater than 15% in the ECOCLIMAP database were removed."

**1.3.3**

**Referee comment** *P7.L26: The authors assimilate the same (aggregated) LAI observation for all grid patches, which may for instance contain a forest patch and grassland patch?*

*Is there any benefit of doing so? I would think this may cause very large increments that could potentially destabilize the model? Updating the aggregated grid cell LAI seems both more theoretically correct and computationally efficient to me. Please comment, and potentially modify the approach.*

**Response:**
Taking into account the grid heterogeneity has been the justication for including vegetation patches in the model and in the assimilation scheme. The assimilation scheme uses the hypothesis that the distribution of innovations is proportional to the cover area. The analysis is adapted to plant functional types via the patch fractions and via the Jacobians. However, the assimilation of LAI has a relatively small impact on the soil moisture fluxes compared with the assimilation of SSM, partly because LAI is assimilated much less frequently (every 10 days as opposed to every 3 days on average). Moreover, it is not the primary cause of the wettening of WG2, as explained in the response to comment 1.2. These explanations are now included in the revised manuscript (page 8, lines 17-19).

**1.3.4**

**Referee comment** *P25.Table2: The assimilation of LAI only seems to flip the sign of the bias against observations, i.e. from +0.11 to -0.08. How is this possible? If the assimilation correctly balances forecast and observation errors (set equal for LAI), it should provide a result that is in-between the observations and simulations? Please*

*comment.*

**Response:**
*This is linked to seasonal changes in the $\frac{\partial LAI}{\partial LAI}$ Jacobian. The behaviour of these Jacobian values was explained in Section 3.3 of the original paper. During the winter/spring the LAI observations are higher than the model, but the $\frac{\partial LAI}{\partial LAI}$ is frequently equal to zero, thus preventing any significant analysis correction. However, during the late summer/autumn the opposite is true; the observations are smaller than the model and the Jacobians are large, so the analysis correction is significantly increased. This results in a time-averaged negative bias of the LAI analysis relative to the observations (shown in Table 2). In Section 3.3 of the revised manuscript we have linked the seasonal changes in the Jacobian to the negative LAI bias in Table 2.*

**1.3.5**

**Referee comment** *P25.Table2: Also, the impact of the LAI assimilation seems to be very large. Is it potentially overdone? This is in large contrast with the results of the soil moisture assimilation, for which Table 3 shows almost negligible impact. Could you please comment? I was also wondering if the impacts on surface soil moisture are comparable to those observed by Draper et al. (2011)?*

**Response:**
*Table 3 in our paper shows the differences between WG1 and the observed SSM at the analysis time. Although SSM is assimilated in our experiments, it is not an analysis variable i.e. WG1 is not updated by the SEKF. Therefore WG1 is only modified indirectly during model interactions with the analysis variables (WG2 and LAI). Draper et al. (2011) updated both WG1 and WG2 with the SEKF, which explains why the assimilation of SSM had a relatively large impact on WG1 in their experiments compared to ours. We have mentioned in the revised manuscript (page 7 line 13-14): "The WG1 layer is not included in the analysis update because it is shallow layer (1 cm depth) that is driven by the atmospheric forcing rather than the initial conditions (Draper et al., 2009; Barbu et al., 2014). It has little bearing on the soil moisture fluxes." Unfortunately it is not possible to observe WG2 with satellite observations so instead we compared WG1 with ASCAT derived SSM. The assimilation did slightly improve the fit to the observations as a result of interactions between SSM and the updated WG2.*

*The average magnitude of the WG2 analysis increments for our experiments was about 0.07 mm/day and the average magnitude of the WG2 analysis increment for Draper et al. (2011) was about 0.1 mm/day. The smaller size of the analysis increments for our experiments is probably a result of slightly larger observation errors prescribed*

*to SSM (we prescribed an average value of 0.055 $m^3/m^3$ and Draper et al. (2011) used the ASCAT SDS estimated values with a mean of about 0.050 $m^3/m^3$).*

the "errors of the day" in the background-error covariance. The operational EnKF at Environment Canada is also motivated by coupling land surface DA with ensemble weather forecasting (Carrera et al., 2015). On the other hand, the SEKF simplifies the EKF by using fixed and uncorrelated background errors at the start of each cycle. Importantly, the SEKF generates flow-dependence and implicit background-error covariances from additional model integrations in the observation operator Jacobian calculations. Draper et al. (2009) found the flow-dependence from a 24-hour assimilation window was sufficient to enable the SEKF to perform similarly to an EKF (which cycles the background-error covariance). Likewise, Muñoz Sabater et al. (2007); Fairbairn et al. (2015) found that the SEKF and EnKF performed similarly, in spite of different linear assumptions.

Historically, the SEKF originated from a simplified 2D-Var (theoretically equivalent to an SEKF) scheme for the assimilation of screen-level temperature and humidity at the German Weather service (DWD) (Hess, 2001). An SEKF has been developed for research purposes to assimilate satellite derived soil moisture at Météo-France (Mahfouf, 2010) and the UK Met Office (Candy et al., 2012), amongst other variables. The European Centre for Medium Range Weather Forecast (ECMWF) model assimilates screen-level temperature and humidity operationally with an SEKF (de Rosnay et al., 2013) and more recently assimilates ASCAT derived SSM observations (ECMWF, 2016).

In our study, we use an SEKF to assimilate LAI and SSM observations to update LAI and WG2 in the ISBA LSM within the SAFRAN-ISBA-MODCOU (SIM) hydrological suite. This study makes use of the A-gs version of ISBA that allows for physiological processes. SIM is operational at Météo-France and its streamflow and soil moisture outputs are used as a tool by the French National flood alert services (Thirel et al., 2010). SIM consists of three stages: (1) An atmospheric reanalysis (SAFRAN) over France, which forces (2) the ISBA-A-gs land surface model, which then provides drainage and runoff inputs to (3) the MODCOU distributed hydrogeological model. The drainage and runoff outputs from ISBA-A-gs are validated by comparing the simulated streamflow from MODCOU with observations. This study is relevant to the land surface DA community because several operational centres assimilate SSM observations using an SEKF to update WG2. Many studies have demonstrated that the force-restore dynamics of the ISBA 3-layer model can effectively simulate soil moisture and propagate the increments downwards from the surface to the root-zone (Muñoz Sabater et al., 2007; Draper et al., 2009; Mahfouf et al., 2009). An integrated validation using SIM has demonstrated that the ISBA 3-layer model can skillfully simulate drainage and runoff fluxes over France (Habets et al., 2008). The dynamic vegetation model in ISBA-A-gs is also capable of modelling seasonal changes in LAI (Jarlan et al., 2008; Brut et al., 2009; Barbu et al., 2011, 2014). But relatively few studies have assessed the SEKF performance using an integrated validation of the drainage and runoff fluxes. To our knowledge, this is the first article to consider this type of validation for LAI assimilation. Furthermore, the validation is robust because it is performed using more than 500 river gauges over France during several years.

This work is partly motivated by the study of Draper et al. (2011), who investigated the influence of assimilating ASCAT derived SSM with an SEKF on SIM over France. They used a version of SIM with high quality atmospheric forcing to represent the "truth" and lower quality atmospheric forcing for the model. Although the SEKF seemed to improve the results in their study, they acknowledged that this may have been related to a bias in the SEKF rather than the assimilation accurately responding to the precipitation errors. Despite the fact that SAFRAN can be considered as a high quality atmospheric forcing, studies by Szczypta et al. (2011) and Le Moigne (2002) have found underestimations of about $5\%$ in the direct short-wave

and long-wave radiative fluxes respectively, averaged over France. In addition to these problems with radiative forcing, we demonstrate in this study that the LSM substantially underestimates LAI for grasslands in winter (compared with satellite retrievals). The specification of the LAI minimum in the model is important because it prevents vegetation mortality and allows the regrowth of vegetation in the spring period (Gibelin et al., 2006). We use SIM to validate the impact of four experiments on the drainage and runoff fluxes:

    i. Correcting the model under-estimated LAI minimum parameter;

    ii. Bias-correcting the SAFRAN radiative forcing;

    iii. Assimilating only LAI observations with an SEKF;

    iv. Assimilating SSM and LAI observations with an SEKF.

The first two experiments attempt to resolve systematic model issues, while experiments (iii) and (iv) assimilate data in order to correct random errors in the initial conditions.

Since Draper et al. (2011) already investigated the impact of assimilating SSM in ISBA on river discharges with MODCOU, it was not necessary to perform an experiment with the assimilation of SSM only. We validate the performance of these experiments using observations from more than 500 river gauges over France during the period July 2007 to August 2014. We include an additional validation using a subset of 67 stations with low-anthropogenic influence because the MODCOU hydrogeological model only accounts for natural features. It should be noted that a bias in the forecast model invalidates the assumption of bias-blind data assimilation (Dee, 2005). We therefore repeat experiments (iii) and (iv) after applying (i) and (ii) in to explore whether the systematic model errors impact the SEKF performance.

The paper is structured as follows. The methods and materials are given in Sect. 2, which includes a description of the LSM, the assimilated observations, the DA methods, the experimental setup and the SIM validation. The results are presented in Sect. 3, including the impact of the model simulations and DA on the model state variables and the river discharge. A discussion in Sect. 4 considers potential solutions to the problems encountered in this study. Finally, the conclusions are given in Sect. 5.

**2  Methods and materials**

**2.1  ISBA-A-gs land surface model**

In our study, the ISBA-A-gs LSM was forced by the atmospheric variables provided by the "Système d'Analyse Fournissant des Renseignements à la Neige" (SAFRAN). The analyses of temperature, humidity, wind speed, and cloudiness are originally performed every 6 h using the ARPEGE (Action de Recherche Petite Echelle Grande Echelle) NWP (Numerical Weather Prediction) model (Courtier et al., 2001). The original precipitation analysis is performed daily at 0600 UTC, to include in the analysis the numerous rain gauges that measure precipitation on a daily basis. A linear interpolation converts these values

to the hourly SAFRAN forcing values (Quitana-Ségui et al., 2008). Instantaneous variables such as precipitation are assumed constant for each 15 minute model time-step during these hourly intervals, while other variables are linearly interpolated. The SAFRAN forcing is assumed to be homogeneous over 615 specified climate zones. The forcing is interpolated from these zones to a Lambert projected grid with a horizontal resolution of 8 km (Durand et al., 1993). The delayed cut-off version of SAFRAN was employed, which uses information from an additional 3000 climatological observing stations (which report one-monthly) over France (Quitana-Ségui et al., 2008; Vidal et al., 2010) after the real-time cut-off, which makes the resulting analyses more accurate.

Version 8.0 of SURFEX was used in the experiments, which contains the "Interactions between Soil, Biosphere and Atmosphere" (ISBA) LSM (Noilhan and Mahfouf, 1996). The model uses the same horizontal grid resolution as SAFRAN of 8 km. The ISBA-A-gs version was used, which allows for the influence of physiological processes, including photosynthesis (Calvet et al., 1998). Each grid cell is split into twelve vegetation types (so called "patches"). Soil and vegetation parameters are derived from the ECOCLIMAP database (Faroux et al., 2013). The nitrogen dilution version (referred to as "NIT" hereafter) of ISBA-A-gs was applied, which dynamically simulates the LAI evolution (Gibelin et al., 2006). The NIT version allows for the effects of atmospheric conditions on the LAI, including the carbon dioxide concentrations.

The three-layer version of ISBA was adopted for this study (Boone et al., 1999). This includes the WG1 layer with depth 0-1 cm. The WG2 layer includes WG1 and is 1-3 m deep, with the depth depending on the patch type. A recharge zone exists below the WG2 layer. The model water transfers are governed by the force-restore method of Deardorff (1977). The surface and root-zone layers are forced by the atmospheric variables and restored towards an equilibrium value. The drainage and runoff outputs from ISBA-A-gs drive the MODCOU hydrogeological model. The gravitational drainage is proportional to the water amount exceeding the field capacity (the effective limit where gravitational drainage ceases) (Mahfouf and Noilhan, 1996). It is driven by the hydraulic conductivity of the soil, which depends on its texture. A small residual drainage below field capacity was introduced by Habets et al. (2008) to account for unresolved aquifers. Runoff occurs when the soil moisture exceeds the saturation value.

**2.2 Assimilated observations**

The SSM observations were retrieved from ASCAT C-band spaceborne radar observations, which observe at 5.255 GHz and a resolution of approximately 25 km. The radar is on board EUMETSAT's Meteorological Operational (MetOP) satellites. The assimilation of ASCAT data was chosen because it was available throughout the analysis period. The original backscatter values were converted into a surface degree of saturation (SDS, with values between 0 and 1) using a change detection technique, which was developed at the Vienna University of Technology (Tu-Wien) and is detailed in Wagner et al. (1999); Bartalis et al. (2007). The historically lowest and highest backscatter coefficient values are assigned to dry and saturated soils respectively. The Copernicus Global Land Service then calculates a soil wetness index (SWI) by applying a recursive exponential filter to these SDS values (Albergel et al., 2008) using a time-scale that may vary between 1 and 100 days. The SWI represents the soil wetness over the soil profile and also has values between 0 (dry) and 1 (saturated). The longer the time-scale of the exponential filter, the deeper the representative soil profile. The SWI-001 version 2.0 product was used in this study, which has a one day

timescale and represents the SWI for a depth up to 5 cm. A surface-state flag is provided with the ASCAT product, which identifies frozen conditions, the presence of snow cover or temporary melting/water on the surface. Observations are screened during frozen surface conditions or when snow-cover is present if the ASCAT flag is set to frozen. Additionally, observations with a topographic complexity flag greater than 15% and/or a wetland fraction greater than 5% (both provided with the ASCAT data) are removed. More information about ASCAT quality flags can be found in (Scipal et al., 2005). After screening, the data were projected onto the 8 km resolution model grid by averaging all the data within 0.15 degrees of each gridpoint (Barbu et al., 2014). As in Draper et al. (2011) an additional screening step was performed to remove observations whenever frozen conditions were detected in the model using a threshold temperature of zero Celsius. In addition, observations with an altitude greater than 1500 m and with an urban fraction greater than 15% in the ECOCLIMAP database were removed.

In order to remove biases between model and observations, a linear rescaling to the SWI-001 data, which scales them such that the mean and standard deviations match the WG1 layer climatology (Calvet and Noilhan, 2000; Scipal et al., 2008). We found that it was necessary to rescale the SSM observations to match the SSM model climatology, partly because differences in the representation of the soil texture can cause very large systematic differences between the observations and the model. These differences are illustrated in terms of probability distribution in Figure S1.4 of the Supplement. It shows the innovation histogram and the Gaussian fitting curve of the SSM product before rescaling.

This rescaling is a linear approximation of the cumulative distribution matching technique, which uses higher order moments (Reichle et al., 2004; Drusch et al., 2005). As in Barbu et al. (2014), we applied a seasonal rescaling using a 3-month moving average over the experiment period (2007-2014). In the rescaling process the SWI-001 data are converted into the same units as the model, expressed in volumetric soil moisture ($m^3/m^3$). The rescaled SSM observations were assimilated into the WG1 model layer. The observations were assumed to occur at the same time as the analysis at 09:00 UTC and had a temporal frequency of about 3 days. This was a reasonable assumption since the satellite overpass is at 09:30 UTC and the atmospheric forcing is assumed constant over hourly intervals for instantaneous measurements such as precipitation. Therefore any discrepancies in SSM due to this 30 minute time difference are small.

The GEOV1 LAI product is part of the European Copernicus Global Land service. The LAI observations were retrieved from the SPOT-VGT (August 2007 to June 2014) and PROBA-V (June 2014-July 2014) satellite data. The retrieval methodology is discussed by Baret et al. (2013). Following Barbu et al. (2014), the 1 km resolution observations were interpolated to the 8 km model gridpoints, provided that observations were present for at least 32 of the observation gridpoints (just over half the maximum amount). The observations were averaged over a 10-day period and assimilated at 09:00 UTC. This assumption was reasonable given that LAI evolves slowly. When considering removing systematic differences between the model and the observations, a linear rescaling of the LAI observations to the model climatology would be problematic because the model-observation bias is linked to model deficiencies. On the other hand, for SSM, systematic errors are related to the misspecification of physiographic parameters, such as the wilting point and the field capacity. As mentioned by several authors (e.g. Koster et al. (2009); Albergel et al. (2012)), the information content of soil moisture does not necessarily rely on its absolute magnitude but instead on its time variations. For SSM, the systematic bias between the model and the data consists mainly in their magnitude rather than their seasonal variability. Therefore this justifies the common approach used in land

surface data assimilation studies for the SSM variable. Contrary to SSM, the LAI bias between the model and the data has two components: one in magnitude and the other one in timing (see e.g. Figure 6 in Barbu et al. (2014)). When compared with the satellite data, the LAI model dynamics clearly shows a shift in the seasonal cycle, mainly caused by model errors. The remote sensing LAI measurements potentially encapsulate realistic environmental features that are not or incorrectly represented by the model. Forcing the data to conform to the model climatology would result in a loss of relevant information. Therefore, in this context, a rescaling of the LAI data to the model climatology was not considered. Furthermore, Barbu et al. (2014) found that the assimilation without rescaling can cope with these model errors.

**2.3   Data assimilation**

The SEKF simplifies the extended Kalman filter (EKF, (Jazwinski, 1970)) by using a fixed estimate of the background-error variances and zero covariances at the start of each cycle (Mahfouf et al., 2009). Implicit background-error covariances between the layers and the prognostic variables are generated at the analysis time by the model integration in the observation operator Jacobians. We used the same SEKF formulation as Barbu et al. (2014) for the assimilation of SSM and LAI observations over France. The prognostic variables are LAI and WG2. The WG1 layer is not included in the analysis update because it is shallow layer (1 cm depth) that is driven by the atmospheric forcing rather than the initial conditions (Draper et al., 2009; Barbu et al., 2014). The background state ($\boldsymbol{x}^{b}$) at time $t_i$ is a model propagation of the previous analysis ($\boldsymbol{x}^a(t_{i-1})$) to the end of the 24 hour assimilation window:

$$\boldsymbol{x}^{\mathrm{b}}(t_i) = M_{i-1}(\boldsymbol{x}^{\mathrm{a}}(t_{i-1})), \tag{1}$$

where $M$ is the (nonlinear) ISBA-A-gs model. The observation was assimilated at the analysis time, at 09 UTC, at the end of the 24-hour assimilation window. The analysis was calculated from the generic Kalman filter equation:

$$\boldsymbol{x}^{\mathrm{a}}(t_i) = \boldsymbol{x}^{\mathrm{b}}(t_i) + \mathbf{K}_i(\boldsymbol{y}_i^{\mathrm{o}} - \boldsymbol{y}_i), \tag{2}$$

where $\boldsymbol{y}^{\mathrm{o}}$ is the assimilated observation and $\boldsymbol{y}_i = H(\boldsymbol{x}^{\mathrm{b}}(t_i))$ is the model predicted value of the observation at the analysis time. The Kalman gain is defined as:

$$\mathbf{K}_i = \mathbf{B}_i\mathbf{H}_i^{\mathrm{T}}(\mathbf{H}_i\mathbf{B}_i\mathbf{H}_i^{\mathrm{T}} + \mathbf{R}_i)^{-1}, \tag{3}$$

where $\mathbf{H}$ is the Jacobian matrix of the linearized observation operator, $\mathbf{B}$ is the background-error covariance matrix and $\mathbf{R}$ is the observation-error covariance matrix. The observation operator Jacobians were calculated using finite differences for observation $k$ and model variable $l$:

$$\mathbf{H}_i^{kl} = \frac{H_i^k(M_{i-1}(\boldsymbol{x}(t_{i-1}) + \Delta x_{i-1}^l)) - H_i^k(M_{i-1}(\boldsymbol{x}(t_{i-1})))}{\Delta x_{i-1}^l}, \tag{4}$$

where $\Delta x^l$ is a model perturbation applied to model variable $l$. The WG2 and LAI perturbations were set to $1.0\times 10^{-4}\times$(w$_{fc}$-w$_{wilt}$) and $1.0\times 10^{-3}\times$LAI respectively. These were within the range of acceptable perturbation sizes based on the experiments

of Draper et al. (2009) and Rüdiger et al. (2010). Equation (4) requires a 24-hour model simulation for each prognostic variable, which implicitly propagates the background-error covariance from the start of the window to the time of the observations at the end of the window. The linear assumptions in deriving the Jacobians are generally acceptable for these perturbation sizes. However, occasionally the linear assumptions can break down, especially during dry periods in summer (Draper et al., 2009; Fairbairn et al., 2015). For this reason we set an upper bound on the soil moisture Jacobians of 1.0. It is worth mentioning that in situations where the model and atmospheric forcing errors are not properly taken into account the SEKF analysis will be suboptimal even if the Jacobians are accurately computed. The Jacobian matrix derived from Eq. (4) is defined as follows:

$$\mathbf{H} = \left( \begin{array}{cc} \frac{\partial \mathrm{WG1}}{\partial \mathrm{WG2}} & \frac{\partial \mathrm{WG1}}{\partial \mathrm{LAI}} \\ \frac{\partial \mathrm{LAI}}{\partial \mathrm{WG2}} & \frac{\partial \mathrm{LAI}}{\partial \mathrm{LAI}} \end{array} \right). \tag{5}$$

When assimilating just LAI, only the $\frac{\partial \mathrm{LAI}}{\partial \mathrm{WG2}}$ and $\frac{\partial \mathrm{LAI}}{\partial \mathrm{LAI}}$ terms are included. The $\frac{\partial \mathrm{WG1}}{\partial \mathrm{LAI}}$ is generally small, since the LAI does not substantially influence the surface layer (Barbu et al., 2014). The $\frac{\partial \mathrm{WG1}}{\partial \mathrm{WG2}}$ Jacobian couples WG1 with WG2 (Draper et al., 2009). The $\frac{\partial \mathrm{LAI}}{\partial \mathrm{WG2}}$ couples LAI with WG2 (Barbu et al., 2014). The $\frac{\partial \mathrm{LAI}}{\partial \mathrm{LAI}}$ Jacobian was studied by Rüdiger et al. (2010) and has a strong seasonal dependence. As we will demonstrate in Sect. 3.3, the examination of these Jacobians is essential in order to understand the performance of the SEKF.

SURFEX is implemented using the mosaic approach of Koster and Suarez (1992), where each model grid-box is split into 12 vegetation patches. The SEKF analysis is calculated independently for each patch using the Jacobians for each individual patch but with one mean observation per grid box. The analysis for the gridpoint is calculated by aggregating the analyses over the 12 patches, which are weighted according to their patch fractions (see Barbu et al. (2014) for further details). Taking into account the grid heterogeneity has been the justication for including vegetation patches in the model and in the assimilation scheme. The assimilation scheme uses the hypothesis that the distribution of innovations is proportional to the cover area. The analysis is adapted to plant functional types via the patch fractions and via the Jacobians.

Following Draper et al. (2011), the WG2 background-error standard deviation was set to $0.2(\mathrm{w}_{fc}\text{-}\mathrm{w}_{wilt})$, where $\mathrm{w}_{fc}$ is the field capacity and $\mathrm{w}_{wilt}$ is the wilting point. The scaling by $(\mathrm{w}_{fc}\text{-}\mathrm{w}_{wilt})$ assumes that there is linear relationship between the soil moisture errors and the dynamic range, which depends on soil texture (Mahfouf et al., 2009). The SSM observation error standard deviation was set to $0.65(\mathrm{w}_{fc}\text{-}\mathrm{w}_{wilt})$, which is about $0.055$ m$^3$/m$^3$ averaged over France. This value is slightly larger than the median ASCAT-derived SDS error of $0.05$ m$^3$/m$^3$ estimated by Draper et al. (2011) because it also approximates the oversampling issue i.e. the same ASCAT observation covers several gridpoints. This reduces the size of the analysis increments by approximately 10%. This value is comparable with observation errors expected for remotely sensed SSM observations (de Jeu et al., 2008; Draper et al., 2013). As in Barbu et al. (2011) the LAI background and observation error standard deviations were proportional to the LAI values themselves and a value of $0.2 \times \mathrm{LAI}$ was used for LAI values greater than $2$ m$^2$/m$^2$. For LAI values below $2$ m$^2$/m$^2$ the LAI errors were fixed at $0.4$ m$^2$/m$^2$. Both the background-error and observation-error covariance matrices of the SEKF are diagonal (zero covariances between layers), but implicit background-error covariances are derived from the **H** matrix at the analysis time. The SEKF is a point-wise method i.e. it cannot take into account horizontal covariances between gridpoints.

**2.4 Experimental setup**

The main experiments in this study are summarised in Table 1. The SIM river discharge was compared with the observations from 546 stations over France. Firstly the baseline experiment (NIT) was performed, which shows the impact of the biased radiative forcing and the under-estimated LAI minimum on the SIM river discharge. Thereafter, two potential solutions to these deficiencies were investigated, as set out in the introduction: (i) $\text{NIT}_m$, which was equivalent to NIT but with an elevated LAI minimum of 1.2 m$^2$/m$^2$ for grasslands (as opposed to 0.3 m$^2$/m$^2$ with NIT), (ii) $\text{NIT}_{bc}$, which used both the elevated LAI minimum of 1.2 m$^2$/m$^2$ and the bias-corrected radiative forcing (+5% for direct long-wave and short-wave over France). Two data assimilation experiments were undertaken to correct random errors in the initial conditions: (iii) LDAS1, which used the SEKF to assimilate LAI only with the NIT model and (iv) LDAS2, which assimilated both LAI and SSM observations with the NIT model. The LAI minimum parameter is required to calculate a minimum level of photosynthesis at the start of the growing season. The default model value is arbitrarily fixed at 0.3 m$^2$/m$^2$ for grasslands, which is low enough to account for possible fluctuations in the LAI minimum due to climatic and interannual variability over France (Gibelin et al., 2006). However, we found that over 99% of points with a high percentage of grassland (the grassland patch fraction exceeding 70%) had an observed average annual LAI minimum above 1.2 m$^2$/m$^2$ during the experiment period (2007-2014). But the modelled LAI is frequently kept at the prescribed LAI minimum parameter during winter dormancy and is therefore systematically underestimated over most grassland regions in winter when compared to the satellite derived observations. Similar issues were found by Brut et al. (2009); Lafont et al. (2012); Barbu et al. (2014) when comparing the model with both MODIS and SPOT-VGT satellite derived observations. Systematic differences between the model and the observations can be removed by calibrating model parameters (Kumar et al., 2012), which was the motivation for increasing the grassland LAI minimum parameter from 0.3 m$^2$/m$^2$ to 1.2 m$^2$/m$^2$ in our study. Szczypta et al. (2011) and Le Moigne (2002) demonstrated that the direct short-wave and long-wave radiative forcing respectively are underestimated by approximately 5% averaged over France. We followed Decharme et al. (2013) in bias-correcting the direct radiative forcing by +5% for $\text{NIT}_{bc}$.

Three additional experiments in Table 1 explored whether SSM observation outliers, the under-estimated LAI minimum or the radiative forcing bias might impact the performance of the DA. The $\text{LDAS2}_{QC}$ was equivalent to LDAS2 but with a strict quality control of the SSM observations. The outliers were removed by rejecting observations outside the 90% confidence interval of the model (as in Eq. (1) and (2) of Albergel et al. (2010b)) after the observations had been rescaled. The $\text{LDAS1}_{bc}$ and $\text{LDAS2}_{bc}$ experiments were equivalent to LDAS1 and LDAS2 respectively, except they used the $\text{NIT}_{bc}$ model. The SSM observations for $\text{LDAS2}_{bc}$ were rescaled such that the standard deviation and mean matched those of $\text{NIT}_{bc}$.

The MODCOU hydrogeological model does not account for anthropogenic water management. However, there are many parts of France where anthropogenic water management strongly influences streamflow observations, including the reservoir operations, for hydropower, irrigation, drinking water, flood and low-flow alleviation and recreation purposes. We used the reference networks of Giuntoli et al. (2012, 2013) to extract a subset of 67 river gauges with low-anthropogenic influence from the original 546 stations, valid for both low and high flows. We compared the results for these 67 stations with the 546 stations

in order to determine if the results were affected by the ability of SIM (with or without DA) to simulate anthropogenically influenced streamflow.

**2.5 Performance diagnostics**

**2.5.1 System validation**

5 A system validation was performed by comparing the LAI and WG1 states with the LAI and SSM observations respectively for all the simulations and data assimilation experiments. Note that this was not an independent validation of the performance of the system, for which we would have needed independent observations. The rationale was to check the effectiveness of the SEKF i.e. to examine if it improved the fit between the model simulations and the observations. The fit to the observations was determined by the root mean square difference (RMSD), the correlation coefficient (CC) and the bias.

10 In addition, Figure S1.5 and Figure S1.6 of the Supplement show the histograms of the innovations (difference between the model-predicted observations and the data) and residuals (difference between the analysis and the data). The SSM innovation pdf agrees very well with Kalman theory, since it closely fits the Gaussian distribution. The LAI innovation pdf is also close to its normal fit, but presents a left tailed distribution. As expected, the standard deviation of residuals is reduced compared to those of innovations. For an "optimal" filter the innovation time series should be uncorrelated in time. For both SSM and LAI

15 the temporal evolutions of innovations are illustrated in Figures S1.7 and S1.8 of the Supplement, respectively.

[revised manuscript text omitted]

**25   4.1   Could LAI assimilation be improved?**

In LDAS1, the seasonal variability in the analysis LAI increments was uneven, with large negative increments in late summer/autumn and small positive increments in winter/spring. This occurred because the LAI Jacobian ($\frac{\partial \text{LAI}}{\partial \text{LAI}}$) was frequently equal to zero during winter and therefore the LAI remained at its incorrect minimum value after the analysis update. Moreover, LAI is only assimilated every 10 days so the model LAI would drift back to its underestimated minimum value between
30   cycles. Consequently, the average LAI analysis was negatively biased. These Jacobian values are physically sensible, since the vegetation is dependent on the atmospheric conditions and is often dormant during the winter period. The problem is related to the lack of a model error term in the SEKF.

The lowest LAI values could be corrected with a full EKF and a model error term, but it would be complicated to parameterize the model-error covariance matrix because the LAI minimum is linked to several factors concerning the atmospheric conditions and the vegetation type. A short-term solution to the underestimated LAI minimum was demonstrated in the experiments, which was to set a higher LAI minimum parameter in the model based on observations. However, it would be more sensible in the long-term to resolve the underlying issues with the model physics. A thorough comparison of the ISBA-A-gs simulated LAI with both SPOT-VGT (used in our experiments) and MODIS data over south-west France was performed by Brut et al. (2009). They did notice significant discrepancies between all three data sets, suggesting that there is significant uncertainty in both the model and the observations. However, they also noticed that the modelled LAI of the C3 natural herbaceous (grasslands)/C3 crops had a delayed onset relative to both satellite products (see Figure 4 in Brut et al. (2009)). They found that this was particularly problematic for grasslands in mountainous regions. By comparing the data with in situ measurements, they found that the generic temperature response of photosynthesis used in the model is not appropriate for plants adapted to the cold climatic conditions of the mountainous areas. This problem was also linked to a prolonged LAI minimum in the model relative to the observations. Lafont et al. (2012) found similar issues when comparing the same products over France. Indeed, Figure 4 in our study shows that the NIT LAI minimum was particularly underestimated in the grassland areas of the Massif Central mountains in central France, but not so much in lower regions further north. Finally, these problems could explain the delayed onset and underestimated LAI minimum for both grasslands and C3 crops in Figure 3 in our study.

It should be recognized that errors in the modelled LAI are not just present over grasslands, but also over other vegetation types. Figure 3 shows there are significant discrepancies between the model and the observations for C3 crops and deciduous forests as well. Given that these discrepancies vary substantially between different vegetation types, it is not optimal to assimilate a gridpoint averaged observation. This issue is currently addressed by disaggregating the LAI for each patch individually.

Finally, as already mentioned, LAI is assimilated every 10-days. LAI data availability could be improved using higher spatial and temporal resolution products in order to limit the impact of clouds.

**4.2 Why does SSM assimilation degrade river discharges?**

It is important to point out that it is physically sensible for WG1 to decouple from WG2 during precipitation events. The precipitation forcing leads to a saturation of the surface layer and subsequently WG1 becomes less dependent on WG2. The degradation of drainage and runoff can be caused by limitations in the SEKF, in the land surface model and in the data.

Firstly, as recognized by Draper et al. (2011), an important problem is that the SEKF is not designed to capture the uncertainty in the model and the precipitation forcing, which should increase during precipitation events and therefore compensates for the smaller Jacobians. The SAFRAN precipitation forcing performs well for a mesoscale analysis and has a higher spatial resolution than global satellite products such as ERA-interim (Quitana-Ségui et al., 2008; Vidal et al., 2010). However, by design the precipitation is assumed to be homogeneous over 615 specified climate zones. Errors are therefore introduced from the spatial heterogeneity of the precipitation, particularly in mountainous regions (Quitana-Ségui et al., 2008).

Secondly, the 3-layer ISBA model has strong nonlinearities near the soil moisture thresholds, some of which lead to unrealistic behaviours of the model Jacobians. During dry conditions in summer the SEKF $\frac{\partial WG1}{\partial WG2}$ Jacobian can be excessive. This is

linked to a rapid increase in transpiration when water is added to WG2 following dry conditions (Draper et al., 2009; Fairbairn et al., 2015). The origin of this nonlinearity is partly related to an unrealistic feature of the surface energy balance. One single surface temperature is used to represent the vegetation and the surface layer, which causes the transpiration to increase too quickly after water is added to WG2 (Draper et al., 2009; Mahfouf, 2014). This problem could be relieved to some extent by introducing the new version of ISBA with a multiple energy balance (MEB, (Boone et al., 2017)) and by using a multi-layer diffusion model (ISBA-DIF, (Decharme et al., 2011)).

Lastly, regarding observations, the current ASCAT product is affected by vegetation (Vreugdenhill et al., 2016) and a seasonal CDF matching is needed in DA systems assimilating ASCAT SSM. This procedure is however sub-optimal. A solution to this problem is to go towards the implementation of an observation operator in order to assimilate the backscattering coefficients directly. In this way, the vegetation information content in the ASCAT signal could be used to analyse vegetation biomass and would also provide information for the analysis of root-zone soil moisture, in addition to the microwave soil moisture signal.

**4.3   Could more sophisticated DA methods improve SSM assimilation?**

The presence of the uncertainties in the model and in the forcing could more easily be addressed with an EnKF than an SEKF because an EnKF can stochastically represent model and precipitation errors (Maggioni et al., 2012; Carrera et al., 2015). Fairbairn et al. (2015) found that an EnKF with a simple stochastic rainfall error estimation demonstrated similar WG2 scores to the SEKF over 12 sites in southwest France (validated using in situ observations). Both methods were affected by nonlinearity problems.

There are DA methods, such as particle filters, designed to handle model nonlinearities. Moradkhani et al. (2012) demonstrated that good results on a hydrological model could be achieved with a particle filter with about 200 members. However, it is substantially more computationally expensive than an EnKF, which typically requires about 20 members to overcome sampling error problems for LSMs (Maggioni et al., 2012; Carrera et al., 2015; Fairbairn et al., 2015). Therefore we intend to test an EnKF over France using the same validation framework used in this study.

**5   Conclusions**

This study assessed the impact on streamflow simulations of assimilating surface soil moisture (SSM) and leaf area index (LAI) observations into the ISBA-A-gs land surface model (LSM). The drainage and runoff outputs were used to force the MODCOU hydrogeological model and were validated by comparing the simulated streamflow with over 500 river-gauge observations over France during several years. To our knowledge, this is the first article to examine the impact of LAI assimilation on streamflow simulations using a distributed hydrological model. Furthermore, this study highlights the importance of systematic model/forcing deficiencies on the streamflow simulations. The validation is robust due to to the large number of river gauge observations employed and the long evaluation period (2007-2014). The results from this study could also have ramifications for flood warning accuracy since SIM is used oprerationally by Meteo-France as a tool for flood forecasting.

Increasing the LAI minimum parameter resulted in greater evapotranspiration in winter/spring and bias-correcting the radiative forcing increased evapotranspiration during much of the year. Both corrections effectively reduced the positive bias in the drainage/runoff fluxes and substantially improved the Nash efficiency scores. Although DA is not theoretically designed to correct systematic model deficiencies, it was found that assimilating only LAI observations substantially reduced the LAI phase errors in the model. However, this induced a net negative bias in the LAI analysis relative to the observations. Given that drainage and runoff occurs predominantly in late winter and spring, the LAI assimilation had negligible impact on these fluxes.

Assimilating SSM resulted in spurious increases in drainage and runoff, which degraded the SIM discharge Nash efficiency.

An issue in DA experiments was the underlying assumption made by the SEKF that the model is perfect. Allowing for model and atmospheric forcing errors could more easily be addressed with an ensemble Kalman filter (EnKF) method than the SEKF, although both methods are affected by nonlinearity issues. In the future we will test the EnKF using a similar validation employed in this study. Regarding LAI assimilation, the SEKF assimilates the LAI observations by aggregating the different vegetation patches in each gridbox. This approach is not optimal because each vegetation type exhibits unique seasonal variability. Given the high resolution of LAI observations (1 km), work is underway to disaggregate the observations.

While the ISBA LSM is well established and is used operationally at Meteo-France, this study has helped us to identify some limitations that need to be addressed. A new multi-layer diffusion model should improve representation of the coupling between the surface and root-zone soil moisture. Furthermore, a new multiple energy balance version should decouple the bare soil evaporation and the transpiration processes that lead to an unphysical link in ISBA between surface and deep soil moisture. Previous research has demonstrated that the generic temperature response of photosynthesis used in the model is not appropriate for plants adapted to the cold climatic conditions of the mountainous areas. This is consistent with the phase errors and the underestimated grassland LAI minimum in our study. Solving this problem would presumably increase the LAI minimum in winter, which would be more sensible than simply fitting the LAI minimum to observations. Finally, the LDAS should benefit from further improvement of the satellite-derived LAI and SSM. Using an observation operator for the ASCAT backscattering coefficients would permit accounting for the vegetation information content in the ASCAT signal.

*Acknowledgements.* This work is a contribution to the IMAGINES (grant agreement 311766) project, co-funded by the European Commission within the Copernicus initiative in FP7. The work was also funded by the EUMETSAT H-SAF service. Discussions with Patrick Le Moigne were useful for understanding the SIM hydrological model. Useful feedback was also obtained through discussions with DA scientists at the Met Office. We would like to thank the two anonymous reviewers for their constructive comments. We would also like to thank Dr Jean-Philippe Vidal from IRSTEA for his useful comments and suggestions regarding anthropogenic water management in the SIM hydrological model.

[revised manuscript text omitted]

**The effect of satellite-derived surface soil moisture and leaf area index land data assimilation on streamflow observations over France**

D. Fairbairn[1], A. L. Barbu[1], A. Napoly[1], C. Albergel[1], J.-F. Mahfouf[1], and J.-C. Calvet[1]

[1]CNRM, UMR 3589 (Météo-France, CNRS), Toulouse, France

February 28, 2017

**Supplement**

[Figure]

Figure S1.1: Map of the SIM discharge Nash efficiency scores for the 67 stations with low-anthropogenic influence over France for the NIT simulation, calculated over the period 2007-2014. The river network is also shown.

[Figure]

Figure S1.2: Scatter plots of the SIM discharge Nash efficiency scores for all 546 stations for (a) NIT vs NITbc and (b) for NIT vs LDAS2. The scores are calculated over 2007-2014.

[Figure]

Figure S1.3: Same as Fig. S1.2, but the stations are classified with either low (67 stations) or high anthropogenic influence (479 stations). For the sake of clarity, the Nash scores are shown between -1.0 and 1.0.

[Figure]

Figure S1.4: Innovation (thick line) histogram and its Gaussian fit (dashed line) for the SSM product without seasonalCDF matching.

[Figure]

Figure S1.5: Innovation (green thick line) and residual (red thick line) histograms, as well as their Gaussian fits respectively, for the SSM product with seasonal CDF matching.

[Figure]

Figure S1.6: Innovation (green thick line) and residual (red thick line) histograms, as well as their Gaussian fits respectively for LAI product.

[Figure]

Figure S1.7: Temporal evolution of SSM innovations.

[Figure]

Figure S1.8: Temporal evolution of LAI innovations.

---

## Author Response (AR3)

**Response to Editor**

March 20, 2017

**Response to minor comments:**

**1**

*I do not entirely agree with the statement of the authors in the reply to comment 3 from referee 1 and comment 1 from referee 2: a. Biases in SSM can show (systematic) variability over time. For instance, saturating and particularly drying rates are often different between models and observations, which could be due to deficiencies in the model, rather than in the parameters (wilting point and porosity). Applying seasonal CDF-matching could therefore eliminate part of the soil moisture information contained in the observations, potentially reducing the impact and benefits of the assimilation. b. Regarding LAI, a linear rescaling (as referred to in the reply to the comments) would not impact the timing of the seasonal cycle, unless it is applied separately over each season (as for the SSM bias-correction). Therefore, even with bias-correction (but depending on the method applied), it should still be possible to correct for the model phase error by the assimilation. But at least, such bias-correction would converge the magnitude (and variability) of the LAI observations and simulations. If the authors do not want to eliminate the climatological information (seasonal cycle) from the LAI observations, this is achievable by not performing bias-correction over each season separately. If the idea is to be consistent with SSM bias-correction, then the latter should probably also conserve the seasonal dynamics (which is not the case with seasonal CDF-matching).*

**Response:**

We agree that biases in soil moisture may be due to errors in the model. In addition, model SSM can only be as good as the precipitation forcing.

The importance of accounting for seasonal corrections in the SSM CDF matching was discussed by several authors (Draper et al. 2009, De Rosnay et al, 2013, Barbu et al, 2014). As mentioned in the manuscript, due to the close relationship between the bias in soil moisture and vegetation, our approach was motivated by the fact that at least

a part of this bias can be attributed to the vegetation seasonality.

Regarding LAI, we agree with the reviewer's suggestions. We will take them into account in the future.

The following sentences are added in the revised manuscript (page 6, lines 26-27). We should be aware that biases in soil moisture can show systematic variability which may be due to model deficiencies rather that to the misspecification of certain parameters. It is not always possible to clearly determine which of the model features is to blame for the bias.

**2**

*I suggest the authors add at least a caveat that the LAI innovations (Fig. S1.8) are quite strongly correlated over time, which is not optimal.*

**Response:**

We agree with this. The following sentence is added at page 10, lines 17-18 of the revised manuscript.

The SSM temporal sequences of innovations are close to a white noise time series (Fig. S1.7), while the LAI innovations (Fig. S1.8) are quite strongly correlated over time, which is not optimal.

**3**

*In general, I think it is important to clearly stress that the reasons for not obtaining improvements in discharge simulations may be due to model deficiencies and the particular setup of the assimilation system, for not to discourage future research efforts on the joint assimilation of SSM and LAI, which I believe is a promising way forward.*

**Response:**

We agree with this. The following sentences is added in Conclusions (page 17, lines 19-20) of the revised manuscript.

Reasons for not obtaining improvements in discharge simulations are related to model deficiencies, model nonlinearities and the set-up of the assimilation system.

**The effect of satellite-derived surface soil moisture and leaf area index land data assimilation on streamflow simulations over France**

D. Fairbairn[1], A. L. Barbu[1], A. Napoly[1], C. Albergel[1], J.-F. Mahfouf[1], and J.-C. Calvet[1]

[1]CNRM, UMR 3589 (Météo-France, CNRS), Toulouse, France

*Correspondence to:* J.-C. Calvet (jean-christophe.calvet@meteo.fr)

**Abstract.** This study evaluates the impact of assimilating surface soil moisture (SSM) and leaf area index (LAI) observations into a land surface model using the SAFRAN-ISBA-MODCOU (SIM) hydrological suite. SIM consists of three stages: (1) An atmospheric reanalysis (SAFRAN) over France, which forces (2) the 3-layer ISBA land surface model, which then provides drainage and runoff inputs to (3) the MODCOU hydro-geological model. The drainage and runoff outputs from ISBA are validated by comparing the simulated river discharge from MODCOU with over 500 river-gauge observations over France and with a subset of stations with low-anthropogenic influence, during several years. This study makes use of the A-gs version of ISBA that allows for physiological processes. The atmospheric forcing for the ISBA-A-gs model underestimates direct short-wave and long-wave radiation by approximately $5\%$ averaged over France. The ISBA-A-gs model also substantially underestimates the grassland LAI compared with satellite retrievals during winter dormancy. These differences result in an underestimation (overestimation) of evapotranspiration (drainage and runoff). The excess runoff flowing into the rivers and aquifers contributes to an overestimation of the SIM river discharge. Two experiments attempted to resolve these problems: (i) a correction of the minimum LAI model parameter for grasslands, (ii) a bias-correction of the model radiative forcing. Two data assimilation experiments were also performed, which are designed to correct random errors in the initial conditions: (iii) the assimilation of LAI observations and (iv) the assimilation of SSM and LAI observations. The data assimilation for (iii) and (iv) was done with a simplified extended Kalman filter (SEKF), which uses finite differences in the observation operator Jacobians to relate the observations to the model variables. Experiments (i) and (ii) improved the median SIM Nash scores by about $9\%$ and $18\%$ respectively. Experiment (iii) reduced the LAI phase errors in ISBA-A-gs but had little impact on the discharge Nash efficiency of SIM. In contrast, experiment (iv) resulted in spurious increases in drainage and runoff, which degraded the median discharge Nash efficiency by about $7\%$. The poor performance of the SEKF originates from the observation operator Jacobians. These Jacobians are dampened when the soil is saturated and when the vegetation is dormant, which leads to positive biases in drainage/runoff and insufficient corrections during winter, respectively. Possible ways to improve the model are discussed, including a new multi-layer diffusion model and a more realistic response of photosynthesis to temperature in mountainous regions. The data assimilation should be advanced by accounting for model/forcing uncertainties.

**1 Introduction**

Soil moisture influences the flow of water to rivers and aquifers on weekly to monthly timescales, which makes it an important factor in hydrological models. In the last two decades there have been considerable advances in soil moisture data assimilation (DA) using remotely sensed near-surface soil moisture (Houser et al., 1998; Crow and Wood, 2003; Reichle and Koster, 2005; Draper et al., 2012; de Rosnay et al., 2013). The estimation of global-scale soil moisture states has benefited considerably from a huge expansion of the satellite coverage, namely the Advanced Scatterometer (ASCAT) instrument on board the METOP satellites (Wagner et al., 2007), the Soil Moisture and Ocean Salinity (SMOS) Mission (Kerr et al., 2001) and the Soil Moisture Active Passive (SMAP) Mission (Entekhabi et al., 2010), amongst others. However, these instruments can only indirectly observe the top 1-3 cm of soil moisture and the data are subject to retrieval errors. There are also spatial and temporal gaps in the observation coverage. The vegetation influences the soil moisture state through evapotranspiration and the vegetation coverage can be estimated by the leaf area index (LAI). This is a dimensionless quantity that represents the one-sided green leaf area per unit ground surface area (Gibelin et al., 2006). The LAI can be derived from satellite measurements in the visible range. However, over France it is available from polar-orbiting satellites at a relatively low temporal frequency (on average every 10 days) compared with soil moisture satellite observations (about every 3 days) due to cloud cover. The aim of DA methods is to combine these observations with a model forecast from the previous analysis (the background state) to provide an improved estimate of the state of the system (the analysis). DA methods are necessary to account for the errors in the observations and the model, and to spread the information through space and time.

Many studies have investigated the assimilation of surface soil moisture (SSM) and streamflow observations into hydrological models in order to improve streamflow predictions and hydrological parameters (Aubert et al., 2003; Moradkhani et al., 2005; Clark et al., 2008; Thirel et al., 2010; Moradkhani et al., 2012). For example, Thirel et al. (2010) used the Best Linear Unbiased Estimate (BLUE) method to assimilate streamflow observations into the MODCOU hydrogeological model over France, which they used to update soil moisture in the ISBA land surface model (LSM).

LSMs concern water and energy fluxes between the soil and atmosphere. Unlike hydrological models, layer-based LSMs such as the ISBA model are typically point-wise (there is no horizontal interaction between the gridpoints), which greatly reduces the computational expense. A 1D Kalman filtering approach (where observations are used to update collocated gridpoints only) is also implemented in this study, which is commonly applied to 1D LSMs (Reichle et al., 2002; Draper et al., 2009; de Rosnay et al., 2013; Barbu et al., 2014).

In large-scale land surface DA, it is common to assimilate satellite derived SSM observations and screen-level temperature and humidity observations into a LSM, in order to improve soil moisture and screen-level variables. Typically the root-zone soil moisture (WG2) (1-3 m deep) is of more interest than SSM as it has a much larger water capacity and a long memory (from weeks to months). Land surface DA is often performed using an ensemble Kalman filter (EnKF) or a simplified extended Kalman filter (SEKF).

There has been increasing interest in ensemble DA for LSMs over the last two decades (Reichle et al., 2002, 2008; Zhou et al., 2006; Muñoz Sabater et al., 2007; Draper et al., 2012; Carrera et al., 2015), partly because these methods can estimate

the "errors of the day" in the background-error covariance. The operational EnKF at Environment Canada is also motivated by coupling land surface DA with ensemble weather forecasting (Carrera et al., 2015). On the other hand, the SEKF simplifies the EKF by using fixed and uncorrelated background errors at the start of each cycle. Importantly, the SEKF generates flow-dependence and implicit background-error covariances from additional model integrations in the observation operator Jacobian

5  calculations. Draper et al. (2009) found the flow-dependence from a 24-hour assimilation window was sufficient to enable the SEKF to perform similarly to an EKF (which cycles the background-error covariance). Likewise, Muñoz Sabater et al. (2007); Fairbairn et al. (2015) found that the SEKF and EnKF performed similarly, in spite of different linear assumptions.

Historically, the SEKF originated from a simplified 2D-Var (theoretically equivalent to an SEKF) scheme for the assimilation of screen-level temperature and humidity at the German Weather service (DWD) (Hess, 2001). An SEKF has been developed

10  for research purposes to assimilate satellite derived soil moisture at Météo-France (Mahfouf, 2010) and the UK Met Office (Candy et al., 2012), amongst other variables. The European Centre for Medium Range Weather Forecast (ECMWF) model assimilates screen-level temperature and humidity operationally with an SEKF (de Rosnay et al., 2013) and more recently assimilates ASCAT derived SSM observations (ECMWF, 2016).

In our study, we use an SEKF to assimilate LAI and SSM observations to update LAI and WG2 in the ISBA LSM within

15  the SAFRAN-ISBA-MODCOU (SIM) hydrological suite. This study makes use of the A-gs version of ISBA that allows for physiological processes. SIM is operational at Météo-France and its streamflow and soil moisture outputs are used as a tool by the French National flood alert services (Thirel et al., 2010). SIM consists of three stages: (1) An atmospheric reanalysis (SAFRAN) over France, which forces (2) the ISBA-A-gs land surface model, which then provides drainage and runoff inputs to (3) the MODCOU distributed hydrogeological model. The drainage and runoff outputs from ISBA-A-gs

20  are validated by comparing the simulated streamflow from MODCOU with observations. This study is relevant to the land surface DA community because several operational centres assimilate SSM observations using an SEKF to update WG2. Many studies have demonstrated that the force-restore dynamics of the ISBA 3-layer model can effectively simulate soil moisture and propagate the increments downwards from the surface to the root-zone (Muñoz Sabater et al., 2007; Draper et al., 2009; Mahfouf et al., 2009). An integrated validation using SIM has demonstrated that the ISBA 3-layer model can skilfully simulate

25  drainage and runoff fluxes over France (Habets et al., 2008). The dynamic vegetation model in ISBA-A-gs is also capable of modelling seasonal changes in LAI (Jarlan et al., 2008; Brut et al., 2009; Barbu et al., 2011, 2014). But relatively few studies have assessed the SEKF performance using an integrated validation of the drainage and runoff fluxes. To our knowledge, this is the first article to consider this type of validation for LAI assimilation. Furthermore, the validation is robust because it is performed using more than 500 river gauges over France during several years.

30  This work is partly motivated by the study of Draper et al. (2011), who investigated the influence of assimilating ASCAT derived SSM with an SEKF on SIM over France. They used a version of SIM with high quality atmospheric forcing to represent the "truth" and lower quality atmospheric forcing for the model. Although the SEKF seemed to improve the results in their study, they acknowledged that this may have been related to a bias in the SEKF rather than the assimilation accurately responding to the precipitation errors. Despite the fact that SAFRAN can be considered as a high quality atmospheric forcing,

35  studies by Szczypta et al. (2011) and Le Moigne (2002) have found underestimations of about $5\%$ in the direct short-wave

and long-wave radiative fluxes respectively, averaged over France. In addition to these problems with radiative forcing, we demonstrate in this study that the LSM substantially underestimates LAI for grasslands in winter (compared with satellite retrievals). The specification of the LAI minimum in the model is important because it prevents vegetation mortality and allows the regrowth of vegetation in the spring period (Gibelin et al., 2006). We use SIM to validate the impact of four experiments

5    on the drainage and runoff fluxes:

    i.  Correcting the model under-estimated LAI minimum parameter;

    ii.  Bias-correcting the SAFRAN radiative forcing;

    iii.  Assimilating only LAI observations with an SEKF;

    iv.  Assimilating SSM and LAI observations with an SEKF.

10    The first two experiments attempt to resolve systematic model issues, while experiments (iii) and (iv) assimilate data in order to correct random errors in the initial conditions.

    Since Draper et al. (2011) already investigated the impact of assimilating SSM in ISBA on river discharges with MODCOU, it was not necessary to perform an experiment with the assimilation of SSM only. We validate the performance of these experiments using observations from more than 500 river gauges over France during the period July 2007 to August 2014.

15    We include an additional validation using a subset of 67 stations with low-anthropogenic influence because the MODCOU hydrogeological model only accounts for natural features. It should be noted that a bias in the forecast model invalidates the assumption of bias-blind data assimilation (Dee, 2005). We therefore repeat experiments (iii) and (iv) after applying (i) and (ii) in to explore whether the systematic model errors impact the SEKF performance.

    The paper is structured as follows. The methods and materials are given in Sect. 2, which includes a description of the

20    LSM, the assimilated observations, the DA methods, the experimental setup and the SIM validation. The results are presented in Sect. 3, including the impact of the model simulations and DA on the model state variables and the river discharge. A discussion in Sect. 4 considers potential solutions to the problems encountered in this study. Finally, the conclusions are given in Sect. 5.

**2   Methods and materials**

25    ### 2.1   ISBA-A-gs land surface model

In our study, the ISBA-A-gs LSM was forced by the atmospheric variables provided by the "Système d'Analyse Fournissant des Renseignements à la Neige" (SAFRAN). The analyses of temperature, humidity, wind speed, and cloudiness are originally performed every 6 h using the ARPEGE (Action de Recherche Petite Echelle Grande Echelle) NWP (Numerical Weather Prediction) model (Courtier et al., 2001). The original precipitation analysis is performed daily at 0600 UTC, to include in

30    the analysis the numerous rain gauges that measure precipitation on a daily basis. A linear interpolation converts these values

to the hourly SAFRAN forcing values (Quitana-Ségui et al., 2008). Instantaneous variables such as precipitation are assumed constant for each 15 minute model time-step during these hourly intervals, while other variables are linearly interpolated. The SAFRAN forcing is assumed to be homogeneous over 615 specified climate zones. The forcing is interpolated from these zones to a Lambert projected grid with a horizontal resolution of 8 km (Durand et al., 1993). The delayed cut-off version of SAFRAN was employed, which uses information from an additional 3000 climatological observing stations (which report one-monthly) over France (Quitana-Ségui et al., 2008; Vidal et al., 2010) after the real-time cut-off, which makes the resulting analyses more accurate.

Version 8.0 of SURFEX was used in the experiments, which contains the "Interactions between Soil, Biosphere and Atmosphere" (ISBA) LSM (Noilhan and Mahfouf, 1996). The model uses the same horizontal grid resolution as SAFRAN of 8 km. The ISBA-A-gs version was used, which allows for the influence of physiological processes, including photosynthesis (Calvet et al., 1998). Each grid cell is split into twelve vegetation types (so called "patches"). Soil and vegetation parameters are derived from the ECOCLIMAP database (Faroux et al., 2013). The nitrogen dilution version (referred to as "NIT" hereafter) of ISBA-A-gs was applied, which dynamically simulates the LAI evolution (Gibelin et al., 2006). The NIT version allows for the effects of atmospheric conditions on the LAI, including the carbon dioxide concentrations.

The three-layer version of ISBA was adopted for this study (Boone et al., 1999). This includes the WG1 layer with depth 0-1 cm. The WG2 layer includes WG1 and is 1-3 m deep, with the depth depending on the patch type. A recharge zone exists below the WG2 layer. The model water transfers are governed by the force-restore method of Deardorff (1977). The surface and root-zone layers are forced by the atmospheric variables and restored towards an equilibrium value. The drainage and runoff outputs from ISBA-A-gs drive the MODCOU hydrogeological model. The gravitational drainage is proportional to the water amount exceeding the field capacity (the effective limit where gravitational drainage ceases) (Mahfouf and Noilhan, 1996). It is driven by the hydraulic conductivity of the soil, which depends on its texture. A small residual drainage below field capacity was introduced by Habets et al. (2008) to account for unresolved aquifers. Runoff occurs when the soil moisture exceeds the saturation value.

**2.2 Assimilated observations**

The SSM observations were retrieved from ASCAT C-band spaceborne radar observations, which observe at 5.255 GHz and a resolution of approximately 25 km. The radar is on board EUMETSAT's Meteorological Operational (MetOP) satellites. The assimilation of ASCAT data was chosen because it was available throughout the analysis period. The original backscatter values were converted into a surface degree of saturation (SDS, with values between 0 and 1) using a change detection technique, which was developed at the Vienna University of Technology (Tu-Wien) and is detailed in Wagner et al. (1999); Bartalis et al. (2007). The historically lowest and highest backscatter coefficient values are assigned to dry and saturated soils respectively. The Copernicus Global Land Service then calculates a soil wetness index (SWI) by applying a recursive exponential filter to these SDS values (Albergel et al., 2008) using a time-scale that may vary between 1 and 100 days. The SWI represents the soil wetness over the soil profile and also has values between 0 (dry) and 1 (saturated). The longer the time-scale of the exponential

filter, the deeper the representative soil profile. The SWI-001 version 2.0 product was used in this study, which has a one day timescale and represents the SWI for a depth up to 5 cm.

A surface-state flag is provided with the ASCAT product, which identifies frozen conditions, the presence of snow cover or temporary melting/water on the surface. Observations are screened during frozen surface conditions or when snow-cover is present if the ASCAT flag is set to frozen. Additionally, observations with a topographic complexity flag greater than 15% and/or a wetland fraction greater than 5% (both provided with the ASCAT data) are removed. More information about ASCAT quality flags can be found in (Scipal et al., 2005). After screening, the data were projected onto the 8 km resolution model grid by averaging all the data within 0.15 degrees of each gridpoint (Barbu et al., 2014). As in Draper et al. (2011) an additional screening step was performed to remove observations whenever frozen conditions were detected in the model using a threshold temperature of zero Celsius. In addition, observations with an altitude greater than 1500 m and with an urban fraction greater than 15% in the ECOCLIMAP database were removed.

In order to remove biases between model and observations, a linear rescaling to the SWI-001 data, which scales them such that the mean and standard deviations match the WG1 layer climatology (Calvet and Noilhan, 2000; Scipal et al., 2008). We found that it was necessary to rescale the SSM observations to match the SSM model climatology, partly because differences in the representation of the soil texture can cause very large systematic differences between the observations and the model. These differences are illustrated in terms of probability distribution in Figure S1.4 of the Supplement. It shows the innovation histogram and the Gaussian fitting curve of the SSM product before rescaling.

This rescaling is a linear approximation of the cumulative distribution matching technique, which uses higher order moments (Reichle et al., 2004; Drusch et al., 2005). As in Barbu et al. (2014), we applied a seasonal rescaling using a 3-month moving average over the experiment period (2007-2014). In the rescaling process the SWI-001 data are converted into the same units as the model, expressed in volumetric soil moisture ($m^3/m^3$). The rescaled SSM observations were assimilated into the WG1 model layer. The observations were assumed to occur at the same time as the analysis at 09:00 UTC and had a temporal frequency of about 3 days. This was a reasonable assumption since the satellite overpass is at 09:30 UTC and the atmospheric forcing is assumed constant over hourly intervals for instantaneous measurements such as precipitation. Therefore any discrepancies in SSM due to this 30 minute time difference are small.

The GEOV1 LAI product is part of the European Copernicus Global Land service. The LAI observations were retrieved from the SPOT-VGT (August 2007 to June 2014) and PROBA-V (June 2014-July 2014) satellite data. The retrieval methodology is discussed by Baret et al. (2013). Following Barbu et al. (2014), the 1 km resolution observations were interpolated to the 8 km model gridpoints, provided that observations were present for at least 32 of the observation gridpoints (just over half the maximum amount). The observations were averaged over a 10-day period and assimilated at 09:00 UTC. This assumption was reasonable given that LAI evolves slowly. When considering removing systematic differences between the model and the observations, a linear rescaling of the LAI observations to the model climatology would be problematic because the model-observation bias is linked to model deficiencies. On the other hand, for SSM, systematic errors are related to the misspecification of physiographic parameters, such as the wilting point and the field capacity. As mentioned by several authors (e.g. Koster et al. (2009); Albergel et al. (2012)), the information content of soil moisture does not necessarily rely on its absolute magnitude but instead on its time variations. For SSM, the systematic bias between the model and the data consists mainly in their magnitude rather than their seasonal variability. Therefore this justifies the common approach used in land surface data assimilation studies for the SSM variable. We should be aware that biases in soil moisture can show systematic variability which may be due to model deficiencies rather that to the misspecification of certain parameters. It is not always possible to clearly determine which of the model features is to blame for the bias.

Contrary to SSM, the LAI bias between the model and the data has two components: one in magnitude and the other one in timing (see e.g. Figure 6 in Barbu et al. (2014)). When compared with the satellite data, the LAI model dynamics clearly shows a shift in the seasonal cycle, mainly caused by model errors. The remote sensing LAI measurements potentially encapsulate realistic environmental features that are not or incorrectly represented by the model. Forcing the data to conform to the model climatology would result in a loss of relevant information. Therefore, in this context, a rescaling of the LAI data to the model climatology was not considered. Furthermore, Barbu et al. (2014) found that the assimilation without rescaling can cope with these model errors.

**2.3 Data assimilation**

The SEKF simplifies the extended Kalman filter (EKF, (Jazwinski, 1970)) by using a fixed estimate of the background-error variances and zero covariances at the start of each cycle (Mahfouf et al., 2009). Implicit background-error covariances between the layers and the prognostic variables are generated at the analysis time by the model integration in the observation operator Jacobians. We used the same SEKF formulation as Barbu et al. (2014) for the assimilation of SSM and LAI observations over France. The prognostic variables are LAI and WG2. The WG1 layer is not included in the analysis update because it is shallow layer (1 cm depth) that is driven by the atmospheric forcing rather than the initial conditions (Draper et al., 2009; Barbu et al., 2014). The background state $(\boldsymbol{x}^b)$ at time $t_i$ is a model propagation of the previous analysis $(\boldsymbol{x}^a(t_{i-1}))$ to the end of the 24 hour assimilation window:

$$\boldsymbol{x}^{\mathrm{b}}(t_i) = M_{i-1}(\boldsymbol{x}^{\mathrm{a}}(t_{i-1})), \tag{1}$$

where $M$ is the (nonlinear) ISBA-A-gs model. The observation was assimilated at the analysis time, at 09 UTC, at the end of the 24-hour assimilation window. The analysis was calculated from the generic Kalman filter equation:

$$\boldsymbol{x}^{\mathrm{a}}(t_i) = \boldsymbol{x}^{\mathrm{b}}(t_i) + \mathbf{K}_i(\boldsymbol{y}_i^{\mathrm{o}} - \boldsymbol{y}_i), \tag{2}$$

where $\boldsymbol{y}^{\mathrm{o}}$ is the assimilated observation and $\boldsymbol{y}_i = H(\boldsymbol{x}^{\mathrm{b}}(t_i))$ is the model predicted value of the observation at the analysis time. The Kalman gain is defined as:

$$\mathbf{K}_i = \mathbf{B}_i\mathbf{H}_i^{\mathrm{T}}(\mathbf{H}_i\mathbf{B}_i\mathbf{H}_i^{\mathrm{T}} + \mathbf{R}_i)^{-1}, \tag{3}$$

where $\mathbf{H}$ is the Jacobian matrix of the linearized observation operator, $\mathbf{B}$ is the background-error covariance matrix and $\mathbf{R}$ is the observation-error covariance matrix. The observation operator Jacobians were calculated using finite differences for

observation $k$ and model variable $l$:

$$\mathbf{H}_i^{kl} = \frac{H_i^k(M_{i-1}(\boldsymbol{x}(t_{i-1}) + \Delta x_{i-1}^l)) - H_i^k(M_{i-1}(\boldsymbol{x}(t_{i-1})))}{\Delta x_{i-1}^l}, \tag{4}$$

where $\Delta x^l$ is a model perturbation applied to model variable $l$. The WG2 and LAI perturbations were set to $1.0 \times 10^{-4} \times$(w$_{fc}$-w$_{wilt}$) and $1.0 \times 10^{-3} \times$LAI respectively. These were within the range of acceptable perturbation sizes based on the experiments of Draper et al. (2009) and Rüdiger et al. (2010). Equation (4) requires a 24-hour model simulation for each prognostic variable, which implicitly propagates the background-error covariance from the start of the window to the time of the observations at the end of the window. The linear assumptions in deriving the Jacobians are generally acceptable for these perturbation sizes. However, occasionally the linear assumptions can break down, especially during dry periods in summer (Draper et al., 2009; Fairbairn et al., 2015). For this reason we set an upper bound on the soil moisture Jacobians of 1.0. It is worth mentioning that in situations where the model and atmospheric forcing errors are not properly taken into account the SEKF analysis will be suboptimal even if the Jacobians are accurately computed. The Jacobian matrix derived from Eq. (4) is defined as follows:

$$\mathbf{H} = \begin{pmatrix} \frac{\partial \mathrm{WG1}}{\partial \mathrm{WG2}} & \frac{\partial \mathrm{WG1}}{\partial \mathrm{LAI}} \\ \frac{\partial \mathrm{LAI}}{\partial \mathrm{WG2}} & \frac{\partial \mathrm{LAI}}{\partial \mathrm{LAI}} \end{pmatrix}. \tag{5}$$

When assimilating just LAI, only the $\frac{\partial \mathrm{LAI}}{\partial \mathrm{WG2}}$ and $\frac{\partial \mathrm{LAI}}{\partial \mathrm{LAI}}$ terms are included. The $\frac{\partial \mathrm{WG1}}{\partial \mathrm{LAI}}$ is generally small, since the LAI does not substantially influence the surface layer (Barbu et al., 2014). The $\frac{\partial \mathrm{WG1}}{\partial \mathrm{WG2}}$ Jacobian couples WG1 with WG2 (Draper et al., 2009). The $\frac{\partial \mathrm{LAI}}{\partial \mathrm{WG2}}$ couples LAI with WG2 (Barbu et al., 2014). The $\frac{\partial \mathrm{LAI}}{\partial \mathrm{LAI}}$ Jacobian was studied by Rüdiger et al. (2010) and has a strong seasonal dependence. As we will demonstrate in Sect. 3.3, the examination of these Jacobians is essential in order to understand the performance of the SEKF.

SURFEX is implemented using the mosaic approach of Koster and Suarez (1992), where each model grid-box is split into 12 vegetation patches. The SEKF analysis is calculated independently for each patch using the Jacobians for each individual patch but with one mean observation per grid box. The analysis for the gridpoint is calculated by aggregating the analyses over the 12 patches, which are weighted according to their patch fractions (see Barbu et al. (2014) for further details). Taking into account the grid heterogeneity has been the justification for including vegetation patches in the model and in the assimilation scheme. The assimilation scheme uses the hypothesis that the distribution of innovations is proportional to the cover area. The analysis is adapted to plant functional types via the patch fractions and via the Jacobians.

[revised manuscript text omitted]

**2.5   Performance diagnostics**

**2.5.1   System validation**

A system validation was performed by comparing the LAI and WG1 states with the LAI and SSM observations respectively for all the simulations and data assimilation experiments. Note that this was not an independent validation of the performance of the system, for which we would have needed independent observations. The rationale was to check the effectiveness of the SEKF i.e. to examine if it improved the fit between the model simulations and the observations. The fit to the observations was determined by the root mean square difference (RMSD), the correlation coefficient (CC) and the bias.

In addition, Figure S1.5 and Figure S1.6 of the Supplement show the histograms of the innovations (difference between the model-predicted observations and the data) and residuals (difference between the analysis and the data). The SSM innovation pdf agrees very well with Kalman theory, since it closely fits the Gaussian distribution. The LAI innovation pdf is also close to its normal fit, but presents a left tailed distribution. As expected, the standard deviation of residuals is reduced compared to those of innovations. For an optimal filter the innovation time series should be uncorrelated in time. For both SSM and LAI the temporal evolutions of innovations are illustrated in Figures S1.7 and S1.8 of the Supplement, respectively. The SSM temporal sequences of innovations are close to a white noise time series (Figure S1.7), while the LAI innovations (Fig. S1.8) are quite strongly correlated over time, which is not optimal.

[revised manuscript text omitted]

**4.1 Could LAI assimilation be improved?**

In LDAS1, the seasonal variability in the analysis LAI increments was uneven, with large negative increments in late summer/autumn and small positive increments in winter/spring. This occurred because the LAI Jacobian ($\frac{\partial \text{LAI}}{\partial \text{LAI}}$) was frequently equal to zero during winter and therefore the LAI remained at its incorrect minimum value after the analysis update. Moreover, LAI is only assimilated every 10 days so the model LAI would drift back to its underestimated minimum value between cycles. Consequently, the average LAI analysis was negatively biased. These Jacobian values are physically sensible, since the vegetation is dependent on the atmospheric conditions and is often dormant during the winter period. The problem is related to the lack of a model error term in the SEKF.

The lowest LAI values could be corrected with a full EKF and a model error term, but it would be complicated to parametrize the model-error covariance matrix because the LAI minimum is linked to several factors concerning the atmospheric conditions and the vegetation type. A short-term solution to the underestimated LAI minimum was demonstrated in the experiments, which was to set a higher LAI minimum parameter in the model based on observations. However, it would be more sensible in the long-term to resolve the underlying issues with the model physics. A thorough comparison of the ISBA-A-gs simulated LAI with both SPOT-VGT (used in our experiments) and MODIS data over south-west France was performed by Brut et al. (2009). They did notice significant discrepancies between all three data sets, suggesting that there is significant uncertainty in both the model and the observations. However, they also noticed that the modelled LAI of the C3 natural herbaceous (grasslands)/C3 crops had a delayed onset relative to both satellite products (see Figure 4 in Brut et al. (2009)). They found that this was particularly problematic for grasslands in mountainous regions. By comparing the data with in situ measurements, they found that the generic temperature response of photosynthesis used in the model is not appropriate for plants adapted to the cold climatic conditions of the mountainous areas. This problem was also linked to a prolonged LAI minimum in the model relative to the observations. Lafont et al. (2012) found similar issues when comparing the same products over France. Indeed, Figure 4 in our study shows that the NIT LAI minimum was particularly underestimated in the grassland areas of the Massif Central mountains in central France, but not so much in lower regions further north. Finally, these problems could explain the delayed onset and underestimated LAI minimum for both grasslands and C3 crops in Figure 3 in our study.

It should be recognized that errors in the modelled LAI are not just present over grasslands, but also over other vegetation types. Figure 3 shows there are significant discrepancies between the model and the observations for C3 crops and deciduous forests as well. Given that these discrepancies vary substantially between different vegetation types, it is not optimal to assimilate a gridpoint averaged observation. This issue is currently addressed by disaggregating the LAI for each patch individually.

Finally, as already mentioned, LAI is assimilated every 10-days. LAI data availability could be improved using higher spatial and temporal resolution products in order to limit the impact of clouds.

**4.2 Why does SSM assimilation degrade river discharges?**

It is important to point out that it is physically sensible for WG1 to decouple from WG2 during precipitation events. The precipitation forcing leads to a saturation of the surface layer and subsequently WG1 becomes less dependent on WG2. The degradation of drainage and runoff can be caused by limitations in the SEKF, in the land surface model and in the data.

Firstly, as recognized by Draper et al. (2011), an important problem is that the SEKF is not designed to capture the uncertainty in the model and the precipitation forcing, which should increase during precipitation events and therefore compensates for the smaller Jacobians. The SAFRAN precipitation forcing performs well for a mesoscale analysis and has a higher spatial resolution than global satellite products such as ERA-interim (Quitana-Ségui et al., 2008; Vidal et al., 2010). However, by design the precipitation is assumed to be homogeneous over 615 specified climate zones. Errors are therefore introduced from the spatial heterogeneity of the precipitation, particularly in mountainous regions (Quitana-Ségui et al., 2008).

Secondly, the 3-layer ISBA model has strong nonlinearities near the soil moisture thresholds, some of which lead to unrealistic behaviours of the model Jacobians. During dry conditions in summer the SEKF $\frac{\partial \mathrm{WG1}}{\partial \mathrm{WG2}}$ Jacobian can be excessive. This is linked to a rapid increase in transpiration when water is added to WG2 following dry conditions (Draper et al., 2009; Fairbairn et al., 2015). The origin of this nonlinearity is partly related to an unrealistic feature of the surface energy balance. One single surface temperature is used to represent the vegetation and the surface layer, which causes the transpiration to increase too quickly after water is added to WG2 (Draper et al., 2009; Mahfouf, 2014). This problem could be relieved to some extent by introducing the new version of ISBA with a multiple energy balance (MEB, (Boone et al., 2017)) and by using a multi-layer diffusion model (ISBA-DIF, (Decharme et al., 2011)).

Lastly, regarding observations, the current ASCAT product is affected by vegetation (Vreugdenhill et al., 2016) and a seasonal CDF matching is needed in DA systems assimilating ASCAT SSM. This procedure is however sub-optimal. A solution to this problem is to go towards the implementation of an observation operator in order to assimilate the backscattering coefficients directly. In this way, the vegetation information content in the ASCAT signal could be used to analyse vegetation biomass and would also provide information for the analysis of root-zone soil moisture, in addition to the microwave soil moisture signal.

**4.3 Could more sophisticated DA methods improve SSM assimilation?**

The presence of the uncertainties in the model and in the forcing could more easily be addressed with an EnKF than an SEKF because an EnKF can stochastically represent model and precipitation errors (Maggioni et al., 2012; Carrera et al., 2015). Fairbairn et al. (2015) found that an EnKF with a simple stochastic rainfall error estimation demonstrated similar WG2 scores to the SEKF over 12 sites in southwest France (validated using in situ observations). Both methods were affected by nonlinearity problems.

There are DA methods, such as particle filters, designed to handle model nonlinearities. Moradkhani et al. (2012) demonstrated that good results on a hydrological model could be achieved with a particle filter with about 200 members. However, it is substantially more computationally expensive than an EnKF, which typically requires about 20 members to overcome

sampling error problems for LSMs (Maggioni et al., 2012; Carrera et al., 2015; Fairbairn et al., 2015). Therefore we intend to test an EnKF over France using the same validation framework used in this study.

**5 Conclusions**

This study assessed the impact on streamflow simulations of assimilating surface soil moisture (SSM) and leaf area index (LAI) observations into the ISBA-A-gs land surface model (LSM). The drainage and runoff outputs were used to force the MODCOU hydrogeological model and were validated by comparing the simulated streamflow with over 500 river-gauge observations over France during several years. To our knowledge, this is the first article to examine the impact of LAI assimilation on streamflow simulations using a distributed hydrological model. The validation is robust due to to the large number of river gauge observations employed and the long evaluation period (2007-2014). The results from this study could also have ramifications for flood warning accuracy since SIM is used operationally by Meteo-France as a tool for flood forecasting. Furthermore, this study highlights the importance of systematic model/forcing deficiencies on the streamflow simulations. Reasons for not obtaining improvements in discharge simulations are related to model deficiencies, model nonlinearities and the set-up of the assimilation system.

Increasing the LAI minimum parameter resulted in greater evapotranspiration in winter/spring and bias-correcting the radiative forcing increased evapotranspiration during much of the year. Both corrections effectively reduced the positive bias in the drainage/runoff fluxes and substantially improved the Nash efficiency scores. Although DA is not theoretically designed to correct systematic model deficiencies, it was found that assimilating only LAI observations substantially reduced the LAI phase errors in the model. However, this induced a net negative bias in the LAI analysis relative to the observations. Given that drainage and runoff occurs predominantly in late winter and spring, the LAI assimilation had negligible impact on these fluxes.

Assimilating SSM resulted in spurious increases in drainage and runoff, which degraded the SIM discharge Nash efficiency.

An issue in DA experiments was the underlying assumption made by the SEKF that the model is perfect. Allowing for model and atmospheric forcing errors could more easily be addressed with an ensemble Kalman filter (EnKF) method than the SEKF, although both methods are affected by nonlinearity issues. In the future we will test the EnKF using a similar validation employed in this study. Regarding LAI assimilation, the SEKF assimilates the LAI observations by aggregating the different vegetation patches in each gridbox. This approach is not optimal because each vegetation type exhibits unique seasonal variability. Given the high resolution of LAI observations (1 km), work is underway to disaggregate the observations.

While the ISBA LSM is well established and is used operationally at Meteo-France, this study has helped us to identify some limitations that need to be addressed. A new multi-layer diffusion model should improve representation of the coupling between the surface and root-zone soil moisture. Furthermore, a new multiple energy balance version should decouple the bare soil evaporation and the transpiration processes that lead to an unphysical link in ISBA between surface and deep soil moisture. Previous research has demonstrated that the generic temperature response of photosynthesis used in the model is not appropriate for plants adapted to the cold climatic conditions of the mountainous areas. This is consistent with the phase errors and the underestimated grassland LAI minimum in our study. Solving this problem would presumably increase the LAI

minimum in winter, which would be more sensible than simply fitting the LAI minimum to observations. Finally, the LDAS should benefit from further improvement of the satellite-derived LAI and SSM. Using an observation operator for the ASCAT backscattering coefficients would permit accounting for the vegetation information content in the ASCAT signal.

*Acknowledgements.* This work is a contribution to the IMAGINES (grant agreement 311766) project, co-funded by the European Commission within the Copernicus initiative in FP7. The work was also funded by the EUMETSAT H-SAF service. Discussions with Patrick Le Moigne were useful for understanding the SIM hydrological model. Useful feedback was also obtained through discussions with DA scientists at the Met Office. We would like to thank the two anonymous reviewers for their constructive comments. We would also like to thank Dr Jean-Philippe Vidal from IRSTEA for his useful comments and suggestions regarding anthropogenic water management in the SIM hydrological model.

[revised manuscript text omitted]